# Global phenology maps reveal the drivers and effects of seasonal asynchrony

Drew E. Terasaki Hart[1,2,3 ✉], Thảo-Nguyên Bùi[1], Lauren Di Maggio[4] & Ian J. Wang[1]

Terrestrial plant communities show great variation in their annual rhythms of growth, or seasonal phenology[1,2]. The geographical patterns resulting from this variation, known as land surface phenology (LSP)[3], contain valuable information for the study of ecosystem function[4,5], plant ecophysiology[6–8], landscape ecology[9,10] and evolutionary biogeography[11–13]. Yet globally consistent LSP mapping has been hampered by methods that struggle to represent the full range of seasonal phenologies occurring across terrestrial biomes[14], especially the subtle and complex phenologies of many arid and tropical ecosystems[1,15,16]. Here, using a data-driven analysis of satellite imagery to map LSP worldwide, we provide insights into Earth's phenological diversity, documenting both intercontinental convergence between similar climates and regional heterogeneity associated with topoclimate, ecohydrology and vegetation structure. We then map spatial phenological asynchrony and the modes of asynchronous seasonality that control it, identifying hotspots of asynchrony in tropical mountains and Mediterranean climate regions and reporting evidence for the hypothesis that climatically similar sites exhibit greater phenological asynchrony within the tropics. Finally, we find that our global LSP map predicts complex geographical discontinuities in flowering phenology, genetic divergence and even harvest seasonality across a range of taxa, establishing remote sensing as a crucial tool for understanding the ecological and evolutionary consequences of allochrony by allopatry.

Plant communities vary widely in their annual rhythms of growth, the collective result of the adaptation of plant life cycles to the vast range of terrestrial environments[1,2,4,15,17]. The spatiotemporal patterns that this creates, known as land surface phenology (LSP), convey rich ecophysiological information about the relationship between bioclimate and plant function[1,6,7,15] and about the modification of that relationship by human land use[3,18]. Robust characterization of these patterns is therefore a critical step in understanding the seasonal dynamics of Earth's terrestrial ecosystems and the constraints that those dynamics impose on native species and human activity. Yet the tendency of phenological research to focus on scalar phenometrics that assume simple annual growth cycles and discrete growing seasons (for example, start and end of season)[14] has limited our ability to understand global LSP diversity, especially in arid and tropical biomes characterized by subtly varying and multimodal phenologies that remain poorly understood[4,7,15,16]. The historical lack of robust remote-sensing proxies of photosynthesis has compounded this limitation, relegating most previous LSP analyses to traditional vegetation indices that have limited sensitivity to seasonal phenology in evergreen ecosystems[6,19]. New remote sensing indices such as near-infrared reflectance of vegetation (NIR$_V$)[20] and sun-induced (or solar-induced) chlorophyll fluorescence (SIF)[21] serve as stronger and less biome-sensitive predictors of seasonal variation in plant productivity[6,19]. Season-agnostic analysis of these proxies of ecosystem function can offer globally consistent insights into LSP.

As a biological signal of the predominant environmental seasonality controlling the phenologies of many species[1,9,13,17], the geography of LSP offers valuable information for landscape ecology and evolutionary biogeography. Spatial variation in seasonal timing can desynchronize phenologies and therefore decouple ecological dynamics between populations[11]. This spatial phenological asynchrony can cause allochronic reproductive isolation[22]—a phenomenon that we term 'allochrony by allopatry'—which can accelerate genetic divergence and, according to the asynchrony of seasons hypothesis (ASH)[12], even facilitate speciation[23]. The ASH posits that this is most common in the tropics: whereas the phenological cues commonly used by high-latitude species (for example, temperature and daylength seasonality) are synchronized across broad geographical areas, the cues thought to be used by many low-latitude species (for example, seasonal availability of water and cloud-attenuated sunlight[2,10,16,17,24]) can diverge over short geographical distances[1,11,12,25–28]. Crucially, the topoclimatic phenomena purported to drive this divergence could even cause seasonal cycles to be out of sync between places with similar climatological averages, such that nearby sites with a similar habitat could exhibit distinct seasonal patterns in potential phenological controls such as temperature, precipitation, cloud immersion or solar radiation[25–27], a pattern we hereafter refer to as isoclimatic phenological asynchrony. This would increase the likelihood that spatial phenological asynchrony could occur between populations of tropical species despite their characteristically narrow climatic niches[29], strengthening the case for allochrony by allopatry

[1]Department of Environmental Science, Policy, and Management, University of California, Berkeley, CA, USA. [2]The Nature Conservancy, Arlington, VA, USA. [3]CSIRO Environment, Brisbane, Queensland, Australia. [4]Department of Statistics, University of California, Berkeley, CA, USA. ✉e-mail: Drew.TerasakiHart@csiro.au

as a contributor to latitudinal and altitudinal gradients in genetic[30] and species diversity[31]. However, observational and genetic evidence for the ASH is scant and mixed[11,25,32,33], global terrestrial patterns of asynchronous seasonality and phenological asynchrony are mostly unknown[12,33], and the geography, drivers and implications of allochrony by allopatry remain largely unexplored.

We present an innovative analytical framework that uses recent advances in remote sensing to provide a global analysis of the diversity and spatial asynchrony of LSP (Extended Data Fig. 1a). First, using harmonic regression to model LSP as a location's long-term average annual phenology (hereafter, its phenocycle; Extended Data Fig. 2), we estimated a global LSP map from a rigorously quality-filtered (Extended Data Fig. 1b), 20-year (2001 through 2020) time series of 0.05° space-based $NIR_V$ imagery[34] (this and all other input datasets are summarized in Supplementary Table 1). We evaluated this LSP map by comparison to identically modelled results derived from space-based SIF imagery[35], ground-based NDVI imagery[36] and flux-tower estimates of ecosystem productivity[37], then used multivariate analysis to visualize the global spatial and temporal diversity of LSP, identifying patterns of regional complexity and intercontinental convergence that we interpret in light of previous research on phenology, climate and land cover. Next, we calculated a global map of spatial asynchrony of LSP, characterized its hotspots and regional drivers, and examined the evidence for a latitudinal gradient in isoclimatic phenological asynchrony. Finally, using a variety of species datasets, we found that our LSP map predicts allochrony by allopatry and its consequent genetic divergence across a range of taxa inhabiting asynchrony hotspots.

## Phenological diversity

Our global LSP map (Fig. 1) shows strong overall performance worldwide (Extended Data Fig. 3) and reveals ecologically interpretable patterns from regional to intercontinental scales, demonstrating the broad value of a globally consistent, multivariate approach to LSP analysis. When the global set of annual phenocycles in this map is rescaled to a common amplitude and animated, complex patterns of spatially variable timing become starkly apparent (Supplementary Video 1). Decomposition of these patterns into empirical orthogonal functions (EOF) shows that Earth's diverse LSP regimes are well explained by a few modes of spatiotemporal variation. The predominant mode (63.89% of total variation) largely reflects the north–south hemispheric summer–winter dipole, but embedded within it is a clear signal of intercontinental phenological convergence across the five global Mediterranean climate regions and portions of their neighbouring drylands as well as a similarly timed signal in coastal wet-forest regions in Brazil and in Somalia, Kenya and Tanzania (Extended Data Fig. 4a). The most marked non-hemispheric signals embedded in modes two (19.17%), three (8.56%) and four (8.39%) reflect the remaining regions comprising the global tropical and subtropical monsoon systems[38] and a number of agricultural land-use patterns.

Rendering the top three modes as a red–green–blue (RGB) composite (Fig. 1 and Extended Data Fig. 4b,c) reveals the bulk of global LSP diversity (>90% total variability) in great clarity. At the broadest scales, intercontinental convergence is instantly visible as a pattern of similar LSP colour gradients occurring within similar geographical and climatic contexts. One notable example is the convergence between Earth's more strongly seasonal Mediterranean-climate regions (California, coastal Chile and the Mediterranean basin)[39] where woodland and other non-forest areas exhibit phenological maxima in late winter and spring (for example, clusters 8 and 9 in Fig. 1), while the predominantly montane forests in those regions display delayed phenologies that are roughly synchronous with the spring–summer green-up across most temperate, high-latitude regions (such as clusters 1–3 in Fig. 1), a finding that corroborates and extends worldwide the phenological 'double peak' described previously in California[40]. Numerous other examples of intercontinental convergence also emerge, including between the more

climatically moderate Mediterranean climate regions of South Africa and southern and southwestern Australia[39] (Extended Data Fig. 5a,b), between the eastern rainforests of Madagascar and northern Queensland, Australia (Extended Data Fig. 5c,d) and between some regions with similar agricultural crops (for example, maize-growing regions in the USA and Italy; Extended Data Fig. 5e,f).

At smaller scales, our methodology reveals complex patterns of regional phenological heterogeneity that suggest possible environmental controls on LSP (Fig. 1a–d). In some regions, climatic gradients are the likely predominant drivers—for example, in southwestern North America, our LSP map reveals distinct winter/spring LSP peaks in coastal habitats and in high desert that contrast with summer/fall peaks in inland and low-desert habitats (Fig. 1a), mirroring the orographically forced division between winter-monsoon (that is, Mediterranean) and summer-monsoon climates[38,39,41]. However, the LSP patterns in other regions suggest additional drivers—for example, in the Basin and Range region (USA), community composition appears to be a key factor[42], with desert regions with a greater abundance of invasive cheatgrass (colour 1 in Fig. 1b; 20.97% annual vegetation, according to recent estimates[43]) showing an earlier spring onset than less-invaded regions (colour 2; 9.39% annual composition; Tukey's honest significant difference, $P < 0.001$). South Florida (USA) presents an example of LSP patterns that are probably driven by community composition that is tied to topohydrology (Fig. 1c). The distinctions between Everglades sawgrass marsh (showing a phenological peak during the winter dry season, when water levels are lowest), the wooded wetland region to the north and west (showing a quick spring peak that may reflect deciduous cypress leaf-out), and areas of drained, upland and mangrove vegetation (showing broader peaks during the summer wet season), are consistent with regional vegetation maps[44] and $CO_2$ exchange studies[45,46]. Finally, in regions in which water and light are the major controls on plant growth[47], stark LSP discontinuities may indicate differences in ecohydrological dynamics, and therefore in water-balance strategies, between different vegetation structural types. The double peak of Mediterranean forest and non-forest habitats is one example of this (Extended Data Fig. 2a). The Amazon may be another. We observe bimodal phenologies in forests that contrast sharply with unimodal phenologies in natural and anthropogenic non-forest (Fig. 1d) and in some riparian zones (Extended Data Fig. 2e). Previous research suggests that these patterns could reflect closer phenological tracking of optimal light availability in forests of the northern and central Amazon basin, where water stress is a less-frequent constraint on growth[6–8] compared with in seasonally drought-stressed non-forest[1,2,8] and seasonally inundated floodplain forests[8]. However, seasonality of tropical plant productivity reflects a complex integration of environmental controls that is still poorly understood[16], so it remains unclear how generally tropical phenologies track light except when constrained by water, as theory suggests[2,17].

We have thoroughly evaluated the performance of the LSP-fitting procedure used to produce these results. We compared our map against a priori expectations, both regionally (for example, comparing to the double peak previously described in California[40]; Extended Data Fig. 2a) and globally (for example, comparing the unimodality/bimodality in our LSP map (Extended Data Fig. 6a) to that described in previous work[4]). We also compared the full LSP map to a 4.3-year time series of independent SIF data from Orbiting Carbon Observatory-2[35] (Extended Data Fig. 6b; pixel-wise median $R^2$ between fitted phenologies, 0.855), after assessing the seasonality of interpolated portions of that SIF dataset against a second SIF dataset (Extended Data Fig. 6c). We then compared the LSP map to identically modelled phenocycles derived from two global time-series datasets: the normalized difference vegetation index (NDVI) from ground-based phenology cameras in the PhenoCam network[36] and gross primary productivity (GPP) from eddy covariance flux towers in the FLUXNET2015 network[37]. Our LSP map shows strong overall agreement with the seasonal signals in both datasets (Supplementary Tables 2 and 3), although with noticeably lower average agreement in

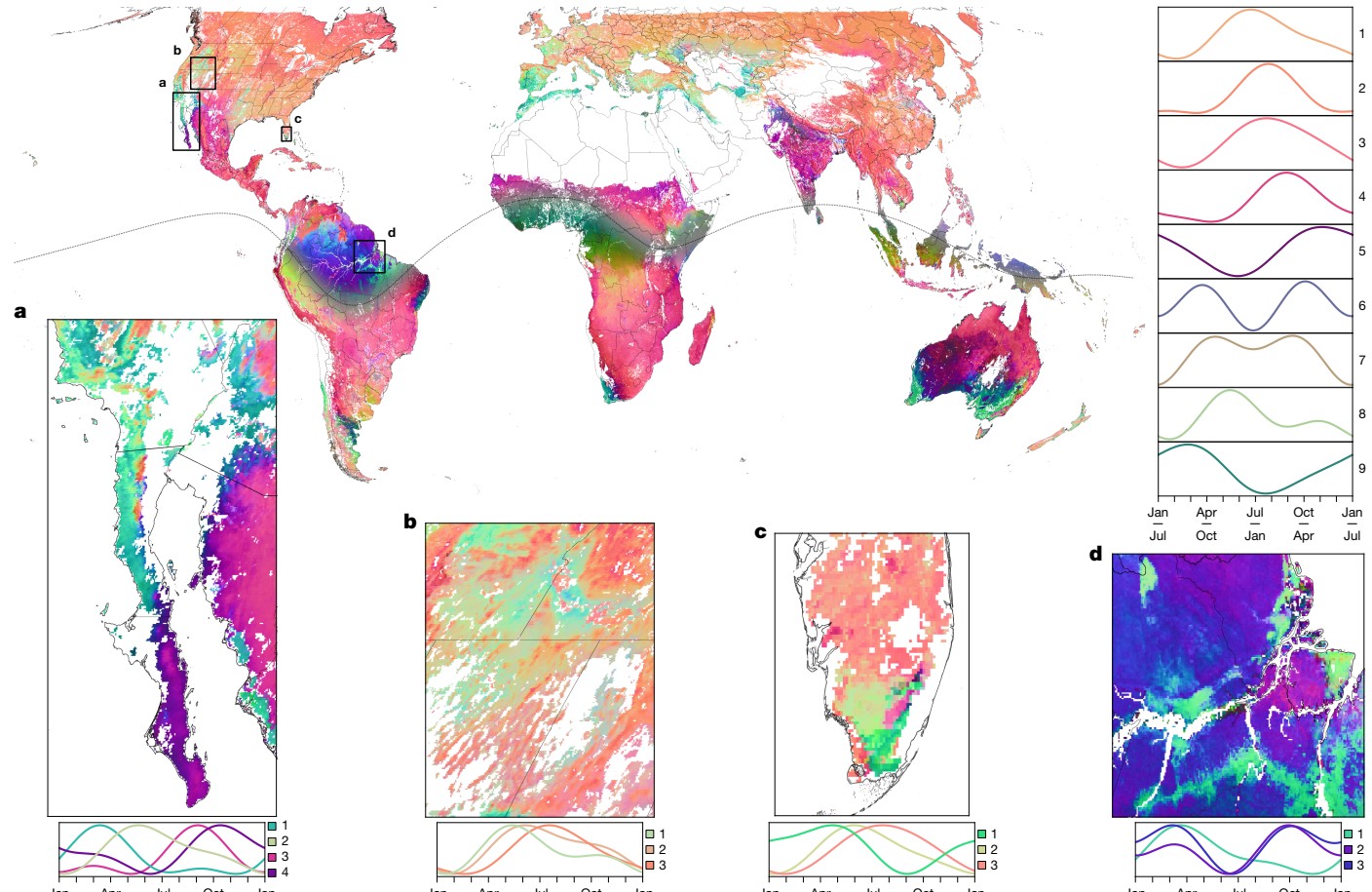

**Fig. 1 | Global LSP mapping reveals intercontinental convergence and complex regional gradients.** The map was coloured by plotting the top three modes of the EOF analysis as an RGB composite. These modes, which explain more than 90% of the phenocycle variation in our global $NIR_V$ time series, were transformed across the intertropical convergence zone (ITCZ; dotted line straddling the equator) before composition, to facilitate interhemispheric comparison. The line plots (top right) depict annual phenocycles (January–January north of the ITCZ; July–July to the south) for nine clusters derived from the global set of fitted phenocycles, coloured by each cluster's median value in the colour composite. Regional maps (**a**–**d**) are paired with phenocycle plots coloured by regionally constrained clustering. Complex gradients appear to reflect patterns of topoclimate, ecohydrology and vegetation structure. **a**, In California and Arizona, USA, and Sonora and the Baja California peninsula, Mexico, a strong gradient aligns with the orographically driven division between Mediterranean winter-monsoon regions (colours 1 and 2) and summer-monsoon regions (3 and 4)[41]. **b**, In the Great Basin, USA, we recover a significant signal of the accelerated spring growth of cheatgrass (1)[42] relative to sagebrush (2) and montane (3) vegetation (one-way analysis of variance (ANOVA), $P < 5 \times 10^{-324}$, with clusters of $n = 4,629$, 904 and 4,891 pixels; two-tailed Tukey's honest significant difference, $P = 5.71 \times 10^{-12}$ in both cases). **c**, In South Florida, USA, we observe starkly contrasting phenologies between Everglades sawgrass marsh (1), wooded wetland (2) and upland, drained and mangrove ecosystems (3). **d**, In the Amazon Delta region, Brazil, we observe unimodal phenologies in non-forest areas (1) that are closely juxtaposed with bimodal phenologies in forest (2 and 3), whether non-forest is naturally occurring (for example, the northwestern patch is Guianan savanna) or anthropogenic (for example, the southern band lies within the arc of deforestation).

semi-arid and seasonally dry biomes than in other ecosystems (Extended Data Fig. 6d,e). We attribute this both to the high interannual variability of productivity in these biomes[48] (especially Australia[15]; Extended Data Fig. 2b), which decreases the likelihood that the temporal patterns of shorter NDVI and GPP time series are characteristic of the long-term average phenologies that we modelled (Extended Data Fig. 6d,e), and to the phenologically divergent land-cover mosaics that can occur there[5,40], which decrease the likelihood that the annual phenology of the vegetation within a camera's field of view or a tower's footprint matches the spatially averaged annual phenology of the mixture of vegetation within a coarser remote sensing pixel.

## Phenological asynchrony

After excluding agricultural pixels to minimize anthropogenic influence[3], we estimated each pixel's spatial phenological asynchrony as the spatial rate of phenological divergence within its surroundings—that is, the slope of the relationship between the geographical and phenological distances between the pixel and all its neighbours (Extended Data Fig. 7a). The phenological asynchrony maps resulting from this calculation show that high asynchrony occurs in regions in which the predominant constraints on plant growth are expected to be availability of light and water, rather than temperature[47], perhaps reflecting the fact that the seasonal timing of these factors is more susceptible to topographic modulation and that their ecophysiological importance varies more as a function of vegetation structure (Fig. 2a). Within that overarching pattern, we find phenological asynchrony hotspots concentrated not only in tropical montane regions, as posited by the ASH, but also in subtropical Mediterranean and semi-arid climate zones—a finding that is consistent across neighbourhood radii (50, 100 and 150 km) both within the $NIR_V$ and SIF datasets (Extended Data Fig. 7b) and between them (Extended Data Fig. 8).

To understand what might generate this pattern, we used a random-forest modelling framework to predict LSP asynchrony as a

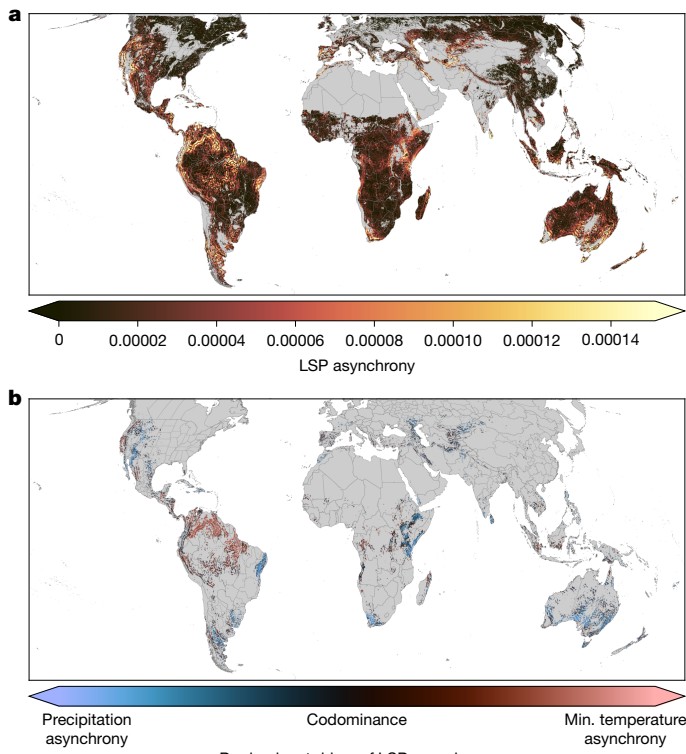

**a**

0    0.00002  0.00004  0.00006  0.00008  0.00010  0.00012  0.00014
LSP asynchrony

**b**

Precipitation            Codominance            Min. temperature
asynchrony                                       asynchrony
Predominant driver of LSP asynchrony

**Fig. 2 | Hotspots of phenological asynchrony are driven by asynchronous seasonality of precipitation and minimum temperature. a,** A global phenological asynchrony map shows the result of applying our asynchrony metric to the MODIS NIR$_V$-derived LSP map, using a 100 km neighbourhood. Brighter colours indicate higher spatial asynchrony of LSP. **b,** The predominance of the two most important drivers of LSP asynchrony, PA (blue) and MTA (pink), varies regionally. Areas that grade towards black show more balanced influence of these two covariates. Predominance was calculated as the normalized difference of pixel-wise absolute SHAP values and plotted within phenological asynchrony hotspots (pixels ≥ 85th percentile). The top two covariates were chosen because their SHAP importance consistently exceeds that of other covariates across models (Extended Data Fig. 9). Min., minimum.

function of its potential environmental drivers. We calculated asynchrony maps for the seasonality of minimum and maximum temperature, precipitation, climate water deficit and cloud cover, all of which revealed strongly structured geographical patterns that were similarly insensitive to neighbourhood size (Extended Data Fig. 7b). We also included four other potential drivers: a measure of topographic complexity, to allow for the latitude–topography interactions expected under the ASH[12]; an index of the spatial variability in vegetation structure, to allow the model to reflect LSP divergence between distinct vegetation types[1,40]; and indices of the frequency of fire and of the extent of land use and non-fire-driven land cover change, to account for potential human contribution to LSP asynchrony patterns. We constructed a random forest for each combination of neighbourhood radius, LSP dataset, and inclusion or exclusion of geographical coordinates (to check the sensitivity to the explicit estimation of spatial process). Despite the local-scale neighbourhood sensitivity of our asynchrony metric (Extended Data Fig. 7b), we found that our overarching modelling results were largely insensitive to all three of these factors (Extended Data Fig. 9a) and showed a strong overall ability to predict patterns of phenological asynchrony ($R^2 = 0.56$ for the 100 km, NIR$_V$-based, coordinate-included model; Extended Data Fig. 9a,b shows $R^2$ values for all models and a map of standardized prediction errors).

Two forms of asynchronous seasonality consistently emerged as the primary drivers of LSP asynchrony: precipitation asynchrony (PA) and minimum temperature asynchrony (MTA). To understand the regional variability of drivers, we calculated the local influence map of each covariate using shapley additive explanations (SHAP) values (Extended Data Fig. 9c), then summarized the maps of the two primary drivers, within LSP asynchrony hotspots (pixels ≥ 85th percentile), as a normalized difference of absolute SHAP values (Fig. 2b). These results suggest that a handful of explanatory mechanisms underlie the major facets of the global pattern of LSP asynchrony. First, PA is the clear driver of asynchrony across the divisions between Mediterranean winter-monsoon regions and neighbouring continental or summer-monsoon climates[38] (Fig. 1a and Extended Data Fig. 2b), whereas PA and MTA are similarly important drivers within Mediterranean climate regions. Second, the drivers of tropical montane LSP asynchrony appear to vary regionally, from PA (for example, the central tropical Andes, the Brazilian Mata Atlântica, the Afromontane, eastern Madagascar and the Australian wet tropics) to MTA (for example, the northern and southern tropical Andes, the Guiana Shield) to codominance in some regions (for example, southern Central America). The unexpected importance of MTA in tropical montane regions may indicate that temperature seasonality exerts phenological control within certain biomes, such as at higher elevations[1], or even that it has a broader but little-recognized role in the control of tropical tree phenologies[16,24,49], but it may also simply indicate omitted variables or complex interactions that are not resolved by our analyses (for example, the interaction of variable insolation with variable thickness of cloud cover[24]). Finally, LSP asynchrony is low across temperate continental climates, where harsh winters synchronize phenologies[47] even when precipitation regimes are spatially variable[50] and PA is high (Extended Data Fig. 7b), and also across many tropical and subtropical regions of low topographic relief, where phenologies are broadly synchronized by year-round warmth and by spatially expansive precipitation regimes unaffected by orographic phenomena[7]. Amazonian forest–savanna ecotones—including in the northwest (that is, the Llanos), the north (that is, Guianan savanna) and the south and southeast (that is, the Cerrado, and the convolved pattern of land clearance in the arc of deforestation)—are a major exception that may relate to the divergence between light-driven and water-driven phenologies[1,8,17].

The logic of the ASH suggests that the phenological asynchrony of tropical montane regions could be caused by spatially variable timing of the topoclimatic phenomena that control seasonal fluctuations in precipitation, cloudiness and available solar radiation—an idea supported by many detailed, regional climatologies[26–28]. This implies that tropical phenological asynchrony could occur not only due to spatial differences in climate but also due to spatial differences in the seasonal timing of similar climates. We refer to such a pattern as isoclimatic phenological asynchrony, and we test for it by examining whether the strength of the relationship between climatic and phenological distances between sites ($\beta_c$), assessed within global high-asynchrony regions, is positively correlated with absolute latitude. Despite the spatial variability in this relationship (Fig. 3a), we find strong overall evidence for a latitudinal gradient (Fig. 3b,c; $P < 0.001$). This lends support to the potential evolutionary importance of allochrony by allopatry: it suggests that allopatric populations of tropical species may experience stronger average allochronic isolation than allopatric populations outside the tropics, owing to the higher likelihood that asynchronous seasonality occurs between sites within even narrow climatic niches (for example, between similar montane rainforests with seasonal precipitation patterns that are similar in shape but are temporally out of phase). If its physiographic basis does indeed persist over long timescales[12], then this phenomenon should be expected to drive genetic differentiation and, therefore, to serve as a mechanism contributing to latitudinal and altitudinal gradients of genetic[30] and species diversity[31], as proposed by the ASH[12]. This could be compounded by other factors believed to make tropical montane species more prone to allopatric isolation and divergence: characteristically narrow topoclimatic

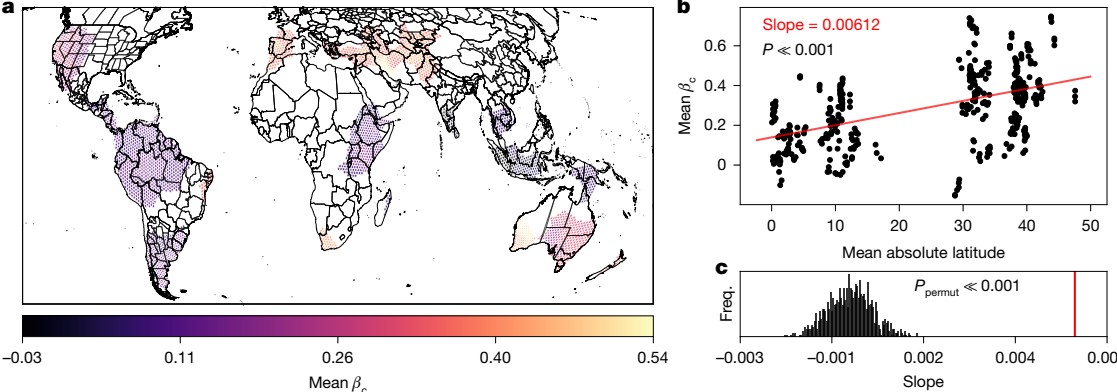

**Fig. 3 | Climatically similar sites exhibit greater phenological divergence within the tropics.** We fit MMRR models in high-asynchrony regions to determine the strength of the relationship between phenological and climatic differences (as estimated by the coefficient $\beta_c$). **a**, This map depicts the average value of $\beta_c$, based on an ensemble of regional MMRR models predicting phenological distance as a function of climatic difference and geographical distance. **b**, The ensemble results (one point per high-asynchrony region;

$n = 429$ regions) reveal a significant, positive relationship between the mean $\beta_c$ and the mean absolute latitude (ordinary least squares regression slope = 0.00612, $P = 9.405 \times 10^{-30}$), indicating a latitudinal gradient in isoclimatic phenological asynchrony. **c**, A two-tailed Monte Carlo analysis of the ensemble results indicates that the slope of the latitudinal gradient is significantly larger in magnitude than slopes derived from 1,000 permuted datasets ($P_{permut} \ll 0.001$). Freq., frequency.

distributions[29], fragmented and isolated climates[31] and low population densities[51].

## Allochrony by allopatry

The evolutionary implications of our results depend on the ability of LSP to serve as a correlate for the climatic and resource seasonality patterns that control the reproductive cycles of a wide range of species[1,9,13,17,24]. In this capacity, remote sensing of proxies for organismal phenology could have an underappreciated role in improving our understanding of spatiotemporal evolutionary dynamics. Using a variety of species-level datasets, we demonstrate that our LSP map is, indeed, a reliable predictor of phenological and genetic signals of reproductive allochrony by allopatry across disparate taxa.

First, using timestamped flower observations from iNaturalist[52], we found that our LSP dataset predicts allochrony by allopatry across a wide range of native plant species. To focus on species most likely to exhibit the sharp phenological discontinuities expected under the ASH—as opposed to the gradual spatial change in bloom date that might occur along altitudinal or latitudinal gradients—we derived a subset of taxa with significantly non-unimodal histograms of range-wide flowering dates (859; 11.8%) from the full number of usable iNaturalist taxa (7,250). As expected, these non-unimodal taxa concentrate in regions where temperature seasonality is not the predominant control on plant growth[47], allowing life histories to spread across much of the calendar year (Extended Data Fig. 10). We dropped 49 taxa that had insufficient data for model fitting, as well as 196 taxa with extremely broad latitudinal distributions that would produce significant but uninteresting results reflecting only the opposite seasonalities of the northern and southern hemispheres. We then used multiple matrix regression with randomization (MMRR)[53] models to test the remaining 614 taxa for a signal of allochrony by allopatry: a significant correlation between flowering date distances and the phenological distances calculated from our LSP map, independent of geographical and environmental distances. Despite the noise inherent in using opportunistic observation dates to represent flowering periods, we found that almost one in five taxa (106; 17.3%) shows evidence of geographical flowering-time variation that is explained by LSP asynchrony (43, or 7.0%, after false-discovery rate correction; Supplementary Table 4). Many of these taxa exhibit patterns of allochrony by allopatry that show marked agreement with stark discontinuities in our LSP map (Fig. 4a).

Next, using data collected from the few published studies of the ASH, we found that our LSP map not only recapitulates the genetic divergence previously attributed to phenological asynchrony in an eastern Brazilian amphibian but also yields convergent results in a sympatric bird. First, we reanalysed data from the only genomic ASH study of which we are aware[25], which reported isolation by PA in a toad (*Rhinella granulosa*) found in eastern Brazil—a region in which we also document strong, precipitation-driven LSP asynchrony (Fig. 2b). Substituting our LSP data for their PA data recovers an identical pattern of genetic isolation by asynchrony (MMRR LSP-distance coefficient = 0.332, $P < 0.001$), visible in the tight agreement between clustering of the sampling locations by their genetic data and clustering by their NIR$_V$ phenocycles (Fig. 4b). Next, using an equivalent analysis, we found similar results (MMRR LSP-distance coefficient = 0.665, $P < 0.001$) and symmetric geographical structure in the only sympatric species included among other available ASH genetic studies, the lesser woodcreeper, *Xiphorhynchus fuscus* (Furnariidae)[32] (Fig. 4b). This suggests that our remote sensing approach can detect little-recognized biogeographical patterns that influence evolutionary dynamics across disparate taxa.

Finally, we found evidence that allochrony by allopatry can also have practical and economic ramifications in other domains, such as agriculture. For example, in contrast to most coffee-producing nations, Colombia is known to have two harvest seasons, fully six months out of sync: some regions harvest from September to December, others from March to June, and still others have a principal harvest in one of these seasons and a minor harvest during the other[54]. A map of classified harvest seasonalities produced by the National Federation of Coffee Growers of Colombia (Fedecafé) reveals a complex geographical pattern, including not only latitudinal structure (the September–December seasonality is predominantly northern), but also orographic structure—that same September–December seasonality extends all the way south along the eastern slope of the Andes. Using sampling points digitized across a previously published version of the Fedecafé map[54], we show that spatial variability in our LSP dataset mirrors the spatial variability in fruiting phenology documented by Fedecafé (Fig. 4c) and that clustering of our LSP dataset significantly matches the official harvest season categories ($P < 0.001$). The pronounced phenological discontinuity across the easternmost range of the Andes causes some sites that are separated by as little as 60 km linear distance to be as out of sync as if they were separated by 60 latitudinal degrees (Extended

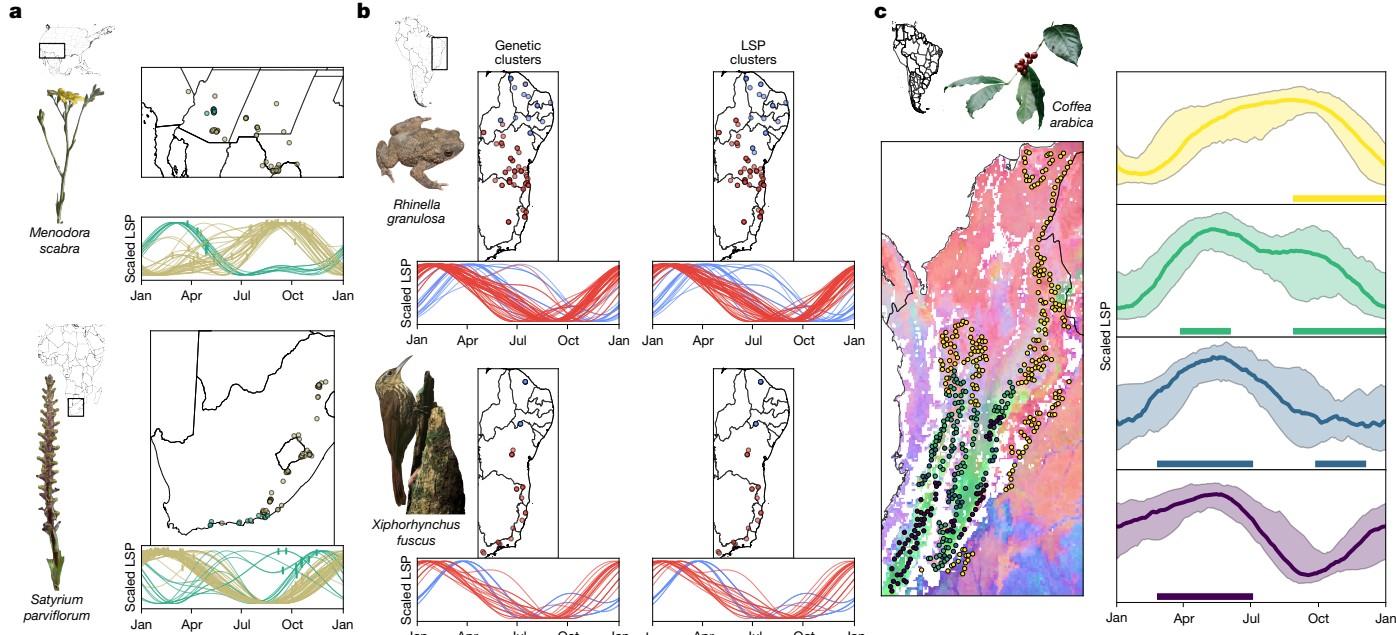

**Fig. 4 | Remote sensing predicts allochrony by allopatry across global hotspots of phenological asynchrony. a**, Example taxa (*Menodora scabra*, in southwestern North America; *Satyrium parviflorum*, in South Africa) display flowering asynchrony based on iNaturalist[52] observations that is predicted by spatial variation in our LSP map. Maps of observation locations and line plots of the phenocycles at those locations are both coloured by *k*-means clustering of the phenocycles (*k* = 2), and the hash marks on the line plots indicate the day of year of each iNaturalist flowering observation. **b**, Data from *R. granulosa*[25] and sympatric *X. fuscus*[32] show congruent results, attributing genetic divergence to allochrony by allopatry across a region of precipitation-driven phenological asynchrony in eastern Brazil. Sampling-site maps and line plots of the phenocycles at those sites, coloured by *k*-means clustering (*k* = 2) of genetic data (left), display close agreement with results coloured by phenocycle clustering (right). **c**, Coffee (*Coffea arabica*) harvest seasons, mapped by Fedecafé[54], exhibit a complex pattern of allochrony by allopatry. Sampling points, coloured by Fedecafé harvest season categories, are mapped over an RGB composite of LSP variability, derived from EOFs (left). This harvest season mapping shows broad agreement with geographical variation in our fitted phenocycles (right). Each harvest season category has a colour-matched plot, depicting the median line and 10th and 90th percentile ribbons across all sampling points and underlined by thick bars indicating the official harvest season months. Species images were derived from iNaturalist photos taken by M. Groeneveld (*M. scabra*; CC BY 4.0), J. Ponder (*S. parviflorum*; CC BY 4.0) and M. Podas (*X. fuscus*; CC BY-SA 4.0); adapted from ref. 64 (CC BY 4.0); or taken by M. Burrows (*C. arabica*).

Data Fig. 2c). This pattern may reflect transmontane differences in the seasonal patterns of precipitation and cloud-attenuated sunlight that result from the orographic blocking of prevailing winds[27,28], a topoclimatic phenomenon that could have a broadly important role in tropical montane biogeography.

## Conclusions

Annual rhythms of climate and resource availability control the phenologies of many plant and animal taxa. These rhythms can differ across geographical space, sometimes substantially. Where this happens, allochrony by allopatry can occur because ecological processes in different locations can be decoupled not only by the physical distance between them but also by the temporal displacement between their asynchronous seasonal and phenological cycles. Our work demonstrates that globally consistent, biome-agnostic remote sensing of LSP, using a minimal modelling framework that avoids the complications arising from concern with discrete growing seasons, thresholded phenometrics and spatially variable amplitudes, can provide a crucial tool for studying this phenomenon. This innovation provides important insights into the global diversity and intricate heterogeneity of terrestrial phenologies—particularly in the tropics, where most LSP algorithms suffer from a tacit 'temperate phenological paradigm'[15,16] that contributes to a persistent phenological knowledge gap[10,13,24]. Our approach could also deepen insights into the phenological shifts happening under climate change[13,16], including phenological anomalies caused by increasingly common extreme weather. Modelling LSP as a long-term average phenocycle has the inherent limitation of excluding these deviations, but also the benefit of providing a multivariate baseline for measuring them, which could help to identify changing patterns of landscape phenology and allochrony by allopatry that can have cascading effects on species interactions and resource availability[10,13,24], animal movement[9], and ecosystem fluxes and phenology–climate feedbacks[4,5].

Our study also offers perspectives and insights across a wide range of domains. For example, although we have focused on terrestrial ecosystems, allochrony by allopatry may have important roles in marine and freshwater ecosystems too. The interactions of currents, stratification, nutrient gradients and atmospheric and coastal influences could create three-dimensional patterns of allochrony by allopatry in marine environments[55], and spatial variation in the seasonal patterns of temperature, hydroclimate and nutrient inputs could cause allochrony by allopatry across the complex geographies of lakes and drainage networks[56]. Our work not only adds to a growing body of evidence that allochrony by allopatry can cause reproductive isolation and genetic divergence[25,32,33] but also suggests avenues for understanding the key life history and landscape parameters that potentiate this. Species with lower dispersal ability, such as the amphibian and understory specialist bird that we analysed here, may have phenologies that are more tightly controlled by local resource availability and may therefore be predisposed[57]. Comparisons between regions could reveal whether divergence is facilitated by isoclimatic phenological asynchrony, which should increase the likelihood of allochrony by allopatry within even narrow climatic niches, or by reduced interannual phenological variability, which should limit long-term gene flow leakage and therefore strengthen reproductive isolation.

Finally, it remains to be determined whether genetic divergence under allochrony by allopatry facilitates speciation and therefore contributes to broad-scale patterns of species diversity, as originally posited by the ASH. Phenological prezygotic isolation is widely recognized as a mechanism of reproductive isolation[22,23], and its instrumental role in some of the best-studied examples of purported sympatric speciation[23,58,59] raises the question of whether 'allochrony by parapatry' might be involved in the speciation of endemics restricted to habitats where our map often shows stark phenological discontinuities across ecotones—including Amazonian floodplains[60], mangroves[61] and other wetlands. Comprehensive phylogeographic work will be needed to determine general patterns, but the strong concordance of hotspots of phenological asynchrony with hotspots of continental biodiversity and endemism[39,62,63], including not only tropical montane regions but also Mediterranean and semi-arid floristic regions, is highly consistent with the notion of allochrony by allopatry as an important macroevolutionary mechanism.

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

## Methods

### Overview of software, data and workflow

We conducted our LSP mapping workflow using Google Earth Engine (GEE) (v.0.1.404 or later)[65] and performed additional analyses using Python[66] with a set of core scientific packages (numpy[67], shapely[68], pandas[69], geopandas[70], rasterio[71], xarray[72], rasterstats[73], dask[74], scipy[75], scikit-learn[76], statsmodels[77] and matplotlib[78]). All of the datasets used in our study are summarized in Supplementary Table 1, and our entire mapping workflow is summarized in Extended Data Fig. 1a.

### LSP datasets

We used GEE to model LSP in two independent time series of remote sensing indices that are strong correlates of seasonal variability in plant productivity—$NIR_V$, an index of the fraction of incident near-infrared light that is reflected by vegetation[20], and SIF, an index of the quantity of incident photons that are absorbed by chlorophyll and re-emitted as fluorescence[21]. We used a 20-year time series of MODIS-derived $NIR_V$ data (daily data from 2001 to 2020, subsampled to every 4 days for computational tractability) as our main dataset (the full workflow diagram is shown in Extended Data Fig. 1a). Following best practices for estimation of patterns at the annual timescale, we chose the MCD43A4 v061 dataset[34], a 16-day temporal composite[79] of nadir bidirectional reflectance distribution function (BRDF)-adjusted reflectance[80]. We used the version of these data that is publicly available in the GEE data catalogue. We did not carry out topographic correction because the scale of our analysis (0.05°; ~5.5 km) is sufficiently coarse that spatial averaging is expected to remove topographic bias[80,81]. We used only pixels with quality values of ≤3 (that is, pixels for which full or magnitude-based BRDF inversions were successfully fitted[82]) for both the red and NIR bands (bands 1 and 2), aggregated to our target analytical resolution of 0.05° (hereafter, target resolution) using the arithmetic mean. We calculated $NIR_V$, as described previously[20], as the product of the NDVI and total NIR reflectance. $NIR_V$ values of ≤0, assumed to be invalid[20], occurred predominantly in high-albedo scenes (for example, treeless snow cover; Extended Data Fig. 2d), where productivity is assumed to be minimal, so they were clamped to the minimum positive $NIR_V$ value observed during a pixel's 20-year time series.

To evaluate our $NIR_V$ maps, we ran some of our main analyses identically but using a global, gridded SIF dataset[35]. This is a roughly 4.3-year (September 2014 to January 2019), 0.05°, spatially contiguous time series dataset, interpolated by artificial neural network (ANN) from the spatially discontinuous SIF data measured along Orbiting Carbon Observatory 2 (OCO-2) orbital swaths. Rigorous internal and external validation of this dataset showed that it accurately captured the global patterns present in the original OCO-2 retrievals and that it explained 81% of the variation in contemporaneous chlorophyll fluorescence imaging spectrometer aerial measurements taken beneath OCO-2 orbits and 72% of the variation in measurements not beneath orbits[35]. We downloaded this dataset from the Distributed Active Archive Center for Biogeochemical Dynamics[83] then ingested it into GEE.

Given that the SIF dataset interpolates across orbital gaps but the paper describing the dataset did not explicitly validate the seasonal phenological patterns of the interpolated data, we assessed the observed seasonality in the interpolated, orbital-gap data against the observed seasonality in another, coarser-resolution SIF dataset collected by the TROPOspheric Monitoring Instrument (TROPOMI)[84,85]. To do so, we extracted SIF time series from the ANN-interpolated dataset at a sample of random points drawn within OCO-2 orbital gaps in three tropical realms (the Neotropics, tropical Africa, and Indo-Pacific and tropical Australia; Extended Data Fig. 6c) then compared those values to contemporaneous time series extracted from the TROPOMI SIF data. We used tropical regions for this assessment because their lack of a pronounced thermal winter creates the greatest possibility that seasonality there exhibits spatially varying patterns that are not

accurately recovered by spatial interpolation from orbital-swath data. If the interpolated dataset adequately captures the true seasonal patterns of SIF within OCO-2 orbital gaps then its time series should explain the bulk of the variation in the TROPOMI time series, and it does ($R^2 = 0.89$; Extended Data Fig. 6c).

### Data filtering

To exclude locations where our harmonic regression-based LSP mapping methodology (see the next section) would return inaccurate results, we used an extensive filtering pipeline that removed invalid land cover, pixels with multiple types of data deficiency and pixels with statistically insignificant LSP regressions. The pixels removed from analysis by each of the filtering steps described below are mapped and summarized in Extended Data Fig. 1b.

For land-cover filtering, we used the GEE data catalog asset for MCD12C1.061[86], a MODIS product estimating annual, global land cover at our target resolution. We used the Annual International Geosphere-Biosphere Programme's (IGBP) classification scheme (land cover type 1). To avoid low-quality data originating from non-target land cover, we excluded data from all pixels with >10% invalid land cover—including urban and built-up land, permanent snow and ice, barren land and water bodies (categories 13, 15, 16 and 0)—for all years within which that classification was assigned. Next, we retained pixels with any other land-cover classifications provided that they never switched between agricultural (categories 12 or 14) and non-agricultural (categories 1 to 11), to avoid fitting phenocycles to the noise resulting from abrupt changes between natural phenologies and those that are deliberately altered by human management (for example, irrigation). We retained pixels where land-cover assignment changed across the time series but was either always agricultural or always non-agricultural because: (1) spurious signals of change between natural land-cover types are common in regions with large, climatically driven interannual variation in plant productivity or where the actual land cover straddles categorical boundaries and challenges classification algorithms (for example, woodland, savanna and semi-arid biomes); (2) actual land-use and land-cover change (LULCC) on the ground is often too subtle to register a change in remotely sensed land-cover maps (for example, selective logging), even when it registers a clear signal in continuous metrics such as $NIR_V$ (Extended Data Fig. 2a); and (3) we only expected other forms of LULCC (for example, deforestation) to affect our modelling results in regions where different land-cover types exhibit different natural phenologies in response to the same broad bioclimatic controls, in which case pixels subject to LULCC should generate model fits that are intermediate to the phenocycles typical of the before and after land-cover types, introducing some noise into our map but neither preventing interpretation of its overarching patterns nor invalidating significant statistical results.

While the LSP of agricultural regions is of interest in many contexts, anthropogenic LSP patterns caused by irrigation and other intensive land management practices[3] could confound our phenological asynchrony analyses, which focus on the climatic drivers and evolutionary implications of longstanding, naturally occurring LSP gradients. Because of this, we used a stricter masking procedure for all datasets used to calculate LSP asynchrony maps and to run asynchrony-related analyses, omitting data from all agricultural pixels (IGBP categories 12 and 14; Extended Data Fig. 1b).

To preclude poorly fitted LSP regressions that could cause spurious results, we removed any target-resolution pixels with data that did not satisfy a set of strict non-missingness criteria. First, we removed any pixels whose LSP time series had >50% missing data, a simple step to remove sites with data dropout because of substantial cloud contamination or MODIS quality control problems. Next, we removed any pixels without at least 10% monthly mean data availability in every month of the year. Finally, owing to a tendency for the harmonic regression procedure (described below) to interpolate spurious second LSP

peaks into extended, seasonally repeating periods of missing data (for example, during high-latitude winters, when Terra and Aqua overpasses occur outside daylight hours for numerous weeks), we removed any pixels for which the binary time series of data availability (0 = missing data, 1 = data available) had a Pielou's evenness[87] of less than 0.8. We calculated Pielou's evenness, $J' = H'/H'_{max}$, using $H'$ (Shannon's diversity index[88]) calculated with 12 values, each value being a monthly average proportion of non-missing 4-day-interval data over the 20-year $NIR_v$ time series. Manual inspection of fitted phenological patterns after applying this series of filtering steps confirmed successful removal of locations that would otherwise produce spurious results. These last two steps removed all locations north of roughly 60° (Extended Data Fig. 1b) because the lack of winter daylight during satellite overpass creates long, seasonally repeating stretches of unavailable data. This is also a known complication for other remote-sensing products (for example, MOD44B.061 Vegetation Continuous Fields[89]), but it does not affect our major findings because the same orbital physics that causes this issue also produces strong, zonally consistent temperature and photoperiod control over annual phenologies at these latitudes and, therefore, limited potential for phenological asynchrony.

Finally, we used the harmonic regression procedure described below not only to calculate characteristic annual LSP patterns but also to estimate the significance of those patterns and filter out pixels with insignificant regression results, using a Monte Carlo framework. To do this, for each pixel in our global $NIR_v$ dataset, we randomly permuted the original LSP time-series image stack, scrambling any true seasonal signal, then ran the harmonic regression and stored an image of the $R^2$ values at all pixels. Next, we calculated from all of the stored $R^2$ images a single summary image of empirical $P$ values indicating, for each pixel, the proportion of permutations for which the permuted time series' $R^2$ values exceeded the $R^2$ value from the unpermuted harmonic regression. We ran this harmonic regression permutation test using 20 permutations at every pixel globally (because of computational limitations), then filtered out any pixels with an empirical $P \geq 0.05$.

## Modelling of LSP

We used harmonic regression to model the long-term average annual LSP pattern (that is, phenocycle) of every pixel in the global, filtered $NIR_v$ and SIF datasets. In our model each pixel's full time series is predicted as a function of time as:

$$y = \beta_0 + \beta_t t + \beta_1 \sin(t_{ann}) + \beta_2 \cos(t_{ann}) + \beta_3 \sin(t_{sem}) + \beta_4 \cos(t_{sem}) + \epsilon,$$

where $y$ is either the SIF or $NIR_v$ time series, $t$ is the linear time component (days from the start of the time series), and $t_{ann}$ and $t_{sem}$ are circular time expressed in annual (ann) and semiannual (sem) frequencies (that is, the day of year expressed in radians, where $2\pi$ radians corresponds to the last day of the year for $t_{ann}$ and to the middle and last days of the year for $t_{sem}$). We then retained all of the resulting coefficient maps except $\beta_t$ (the trend), yielding a stack of five coefficient maps that represents the detrended, long-term, characteristic annual LSP pattern at each pixel globally.

We chose harmonic regression because it is a simple, widely used and clearly interpretable approach to time-series analysis[90], and because it would enable us to characterize the long-term average annual behaviour at all terrestrial locations. Our regression formulation is algebraically equivalent to detrending the full 20-year time series, then running a Fourier transform that includes both annual and semiannual frequency components[91]. We designed a number of the data-filtering approaches described above to ensure against the spurious interpolation into seasonally repeating data gaps that could otherwise be caused by this method. We chose to include both the annual and semiannual frequencies in the harmonic regression to strike a balance between model complexity and overfitting. We expected that complex annual LSP patterns would occur in locations that have

bimodal seasonal precipitation patterns (that is, two rainy seasons)[50] and no winter freeze[47]. Indeed, preliminary analysis revealed numerous regions with stronger bimodal than unimodal annual LSP patterns (that is, regions containing many pixels whose $R^2$ values were higher in semiannual-only harmonic regression models than in annual-only models). The linear combination of annual and semiannual harmonic regression components is complex enough to represent annual LSP curves that are unimodal, evenly bimodal (two equal peaks and troughs) or unevenly bimodal (featuring major and minor peaks and troughs), but not more complex, and therefore avoids overfitting by excluding unfounded higher frequencies[90].

While frequency-specific phase and amplitude estimates could be recovered from the fitted coefficients of our models, their comparative interpretation across such a wide range of phenological patterns would be difficult. Thus, for all downstream analysis and visualization, we instead use Euclidean distances and multivariate statistics calculated directly on the fitted phenocycles, which can be calculated as the multiplication of a pixel's fitted harmonic regression coefficients with the 1-year matrix of daily time values expressed in linear time and in annual and semi-annual cyclical time. Extended Data Fig. 2a–d pairs multivariate visualization (methods described below) with demonstrations of the phenocycle-fitting procedure in various test regions, and Extended Data Fig. 2e shows a similar visualization screenshotted from the GEE app that we created for public exploration of our results (the link to which is provided within the GitHub repository for this project; https://github.com/erthward/phen_asynch, https://doi.org/10.5281/zenodo.15671259)[92].

## Evaluation of LSP mapping

We first evaluated the annual $NIR_v$ LSP map by calculating and inspecting a map of $R^2$ values between the fitted $NIR_v$ and SIF phenocycles at all pixels (Extended Data Fig. 6b). We also checked the distribution of unimodal and bimodal phenologies against prior studies. To do this, we min–max scaled each pixel's phenocycle to the [0, 1] interval and rotated it to start at its minimum value (to avoid problems arising from phenocycle peaks that straddle the start of the calendar year). We then extracted the heights of each phenocycle's peaks, using the 'find_peaks' function in the 'signal' module of the Python package scipy (v.1.13.0)[75], and used the absolute difference of those heights as an indicator of where a pixel lies on a spectrum between perfectly bimodal (0: indicating two peaks of equal height) and unimodal (1: assigned to phenologies having only a single peak). We mapped this index (Extended Data Fig. 6a), then visually compared it to previously published depictions of the global distribution of regions with one versus two growing seasons (see figure 3 of ref. 4).

We also evaluated the fitted phenocycles for both LSP datasets ($NIR_v$, SIF) by comparison with average phenocycles fitted identically to time series of PhenoCam[36] NDVI and FLUXNET2015[37,50] GPP. For the PhenoCam analysis, we used a combination of the R[93] package phenocamapi (v.0.1.5)[94] and custom Python code to download all available (as of 5 March 2025) 3-day summary NDVI datasets from all cameras and regions of interest (ROIs; masked areas of uniform vegetation within a camera's field of view, which are used to generate separate time series datasets). We used NDVI because its phenological signal, which can diverge from that of the green chromatic coordinate in some systems[36], provides a better comparator to our NDVI-derived $NIR_v$ data. We used the 3-day summaries because they have reduced noise, and we analysed the 75th-percentile NDVI summary values to strike a reasonable trade-off between the tendency of higher-percentile values to be less noisy under variable lighting conditions and the risk that very high percentiles can cause outlier influence[95]. We dropped any camera sites that PhenoCam reports as belonging to any of the invalid IGBP land cover classes that we filtered out of our LSP analysis (urban and built-up land, permanent snow and ice, barren land and water bodies) or as being agricultural (because agricultural management

could cause an entirely different phenology within a camera's field of view than the spatially averaged phenological signal reflected in our LSP map), leaving a total of 368 camera sites eligible for analysis. Before fitting a harmonic regression for each site, we removed outliers from each of the site's ROI datasets (using the outlier flag provided by PhenoCam), then combined all datasets by averaging each day's values across all ROIs to approximate the integrated land cover signal in our LSP dataset at that site. We then used the same harmonic regression model used in the LSP-fitting procedure described above to calculate a set of five coefficients describing the detrended, average annual NDVI phenology for a site, and from those coefficients performed matrix multiplication to recover the fitted characteristic annual NDVI phenocycle for each site. Finally, for each site, we calculated the $R^2$ values between the site's characteristic annual NDVI phenocycle and the LSP phenocycle corresponding to the site (that is, the pixel where the camera is located or, if that pixel is masked in our LSP dataset, the nearest valid pixel within a two-pixel-wide box that surrounds it). We summarized this evaluation procedure across all camera sites by producing, for each LSP dataset: (1) a scatter plot of the LSP-NDVI $R^2$ values plotted on the Whittaker biomes[96], to depict bioclimatic patterns in evaluation performance; and (2) a scatterplot comparing LSP-NDVI $R^2$ values to NDVI time series lengths, to depict the relationship between camera data availability and evaluation performance (Extended Data Fig. 6d).

For the FLUXNET2015 comparison, we manually downloaded all datasets available at the time of access (11 October 2021), then, as with PhenoCam, dropped all flux tower sites reporting invalid and agricultural land cover types, yielding 170 valid GPP datasets for analysis. Before fitting a harmonic regression to each dataset, we first removed all datapoints with a daily quality value of <0.7 (that is, with <70% measured or good-quality gap-filled data contributing to their daily aggregated values). We then used the same methods as described for the PhenoCam NDVI comparison above to fit a harmonic regression, predict a characteristic annual time series, calculate $R^2$ values between the annual time series and those from their closest available LSP pixels (up to 2 pixels distant, otherwise a tower's dataset was dropped) and visualize the results (Extended Data Fig. 6e).

## LSP visualization

To visualize the global variability of seasonal LSP that is present in the results of our harmonic regression, we used colour-composite visualization of the results of a dimensionality-reduction analysis to produce a single global map. First, we used Python v.3.7 and the eofs package (v.1.4.0)[97] to run EOF analysis on the covariance matrix of the global set of NIR$_V$ phenocycles. We standardized each pixel's phenocycle before EOF calculation, ensuring that all pixels had equal variances of 1 and therefore allowing the EOF analysis to highlight global variability in the shape and timing of LSP patterns, our topic of interest, irrespective of spatial variation in NIR$_V$ amplitude. Following common practice in EOF analysis, we used the square root of the cosine of the latitude as pixel area weights.

This calculation reduced the global diversity of average annual LSP patterns to four EOFs. Finding that the first three EOFs cumulatively explain >90% of the variation in the dataset (91.62%; Extended Data Fig. 4a), we min–max scaled them, then displayed them using the RGB colour channels, visualizing the bulk majority of global LSP variability within a single map. As they have embedded within them both the unremarkable north–south hemispheric seasonality dipole and hemisphere-independent patterns of interest (for example, monsoon-driven LSP dynamics), we transformed the raw EOF maps before RGB visualization to represent phenological variability in a globally consistent colour scheme. To accomplish this, we used Web-PlotDigitizer[98] to digitize a geospatial vector file of the mean ITCZ in both boreal summer (June, July, August) and boreal winter (December, January, February)[38], then calculated a single, annual mean ITCZ vector

by averaging the boreal summer and winter latitudes at evenly spaced longitudes around the globe. Finally, for each EOF, we constructed a synthetic, transformed map by calculating $w \times EOF + (1 - w) \times (1 - EOF)$, where $w$ varies from 1 in the northern hemisphere to 0 in the southern hemisphere and transitions linearly from 1 to 0 within a 10° latitudinal band surrounding the annual mean ITCZ. We chose to use the ITCZ as the latitudinal boundary across which to transform the EOF maps because it serves as a more natural meteorological Equator than does the geographical Equator[17,38]. To help to interpret the result of this visualization across the region surrounding the ITCZ (Fig. 1), where some colour-warping occurs, we also generated RGB composite maps using untransformed EOF maps and using EOF maps transformed uniformly as 1 − EOF (Extended Data Fig. 4b,c). As this transformation is used only for visual comparison across hemispheres, it has no influence on any of the analytical results reported in our work.

To depict the characteristic phenocycles corresponding to the RGB visualization, we use mini-batch $k$-means clustering (a version of the standard $k$-means clustering algorithm that reduces computational burden by using only a fixed-size random subsample of the full dataset at each iteration) to cluster the standardized, fitted phenocycles within a region into $k$ colours, for $k = 1$:12, then visually inspect a scree plot to determine the optimal value of $k$. Using that chosen value, we assign each pixel to one of $k$ clusters, then plot each cluster centre (after min–max scaling) as its characteristic phenocycle, coloured by the median RGB value across all pixels in the cluster. We used this procedure to produce plots interpreting the predominant phenocycles both globally (Fig. 1 (main)) and within various focal regions (Fig. 1a–d and Extended Data Fig. 5). Before clustering the global map, we rotated the fitted phenocycles of all pixels below the mean ITCZ by 182 days (that is, half a year) to allow similar phenologies in the northern and southern hemispheres to cluster together.

Discovering regional phenological variability in the Great Basin of the United States that appeared to match the cheatgrass-invaded, sagebrush and montane phenologies presented previously[42], we used ancillary data from ref. 43, aggregated to the target resolution of our map, to calculate the average estimated percentage of annual herbaceous cover in each of the three predominant clusters depicted in our analysis (Fig. 1b). To support our interpretation of the three clusters as annual-invaded communities, sagebrush and montane vegetation, which we based on the differences in their estimated average annual herbaceous cover and on a visual comparison to ref. 42, we used ANOVA to test for a significance difference in the estimated percentage of annual herbaceous cover across all three clusters, followed by a Tukey's honest significant difference test to test for significant pairwise differences between the clusters.

To better highlight complex geographical patterns of spatially variable LSP timing, we also produced a video (Supplementary Video 1) animating the min–max scaled average NIR$_V$ phenocycle at each pixel. Scaling each pixel's phenocycle in this way forces all pixels to a common annual amplitude (from zero to one), ignoring spatial differences in intra-annual variability caused by variable ecosystem productivity, and thus highlighting spatial differences in the timing and rates of change of LSP.

## Calculation of phenological asynchrony

We exported the GEE results of our filtered harmonic regression as a global set of tiled, multiband images of regression coefficients. We used GEE's TensorFlow output format and 'kernelSize' argument to generate tiles that overlapped their neighbours by 300 km (double the largest neighbourhood size in our asynchrony calculations), to allow asynchrony to be calculated independently and in parallel.

For each LSP dataset (NIR$_V$, SIF), we calculated our asynchrony metric pixel-wise, for all pixels with at least 30 available neighbours, using an algorithm based on Martin et al.[12] and depicted in Extended Data Fig. 7a:

(1) Calculate the standardized phenocycle for a focal pixel.
(2) Identify all pixels of which the centrepoints are within the chosen neighbourhood radius of the focal pixel (the neighbour pixels).
(3) For each neighbour pixel: (a) calculate its standardized phenocycle; (b) calculate the 365-dimensional Euclidean phenological distance between its phenocycle and the focal pixel's phenocycle; (c) calculate its geographical (geodesic) distance to the focal pixel.
(4) Calculate asynchrony as the slope of the regression of Euclidean phenological neighbour distances on geographical neighbour distances (or as zero, wherever the slope has $P > 0.01$).

We used a regression approach to calculate the asynchrony metric because it explicitly estimates the spatial rate of change in phenology, and therefore well represents the spatial rate of change of seasonal timing that is the subject of the ASH[12]. We standardized phenocycles, nullifying differences in amplitude, before calculating Euclidean distances between them, therefore preserving the timing differences that we are interested in, even between similar-shape but out-of-phase curves (a criterion not met by other common distance metrics, such as dynamic time warping). We ran this calculation in Julia (v.1.4.1)[99] on UC Berkeley's Savio cluster, parallelized by tile, then mosaicked the results into a global map (Fig. 2a).

We produced this global map for each of three neighbourhood radii (50, 100, and 150 km), enabling us to check the sensitivity of our maps and our downstream results to this decision. The values of the resulting maps, expressed as a spatial rate of change in the target variable's units (that is, $\Delta\text{unit}_{\text{target\_variable}}/\Delta m$), scale arbitrarily with a map's neighbourhood radius, but each map provides an internally valid quantitative basis for assessing and comparing asynchrony between sites. To assess the overall level of agreement between the $\text{NIR}_\text{v}$ and SIF asynchrony maps, despite the fine-scale noise expected in a neighbourhood metric, we mapped and scatter plotted pixel-wise comparisons between the two datasets for each of the three neighbourhood radii (Extended Data Fig. 8). Moreover, to evaluate the scale-sensitivity of the LSP asynchrony maps (and of the asynchrony maps that we likewise calculated for the climatic covariates described below), we assessed, for each mapped variable, the $R^2$ values for all three pairwise interneighbourhood map comparisons (Extended Data Fig. 7b).

To visually depict the asynchrony algorithm, we first simulated harmonic-regression output for a low-asynchrony region as a five-layer stack of coefficient values with rasters of low relative-magnitude Gaussian noise added to them and for a high-asynchrony region as a five-layer stack of mean coefficient values with large relative-magnitude, spatially autocorrelated noise added to them using neutral landscape models generated using the nlmpy Python package[100]. We represented each five-layer simulated map as a single-layer map by first calculating each pixel's phenocycle from its simulated vector of harmonic regression coefficients, then calculating the day of the year when its simulated phenocycle attains its peak value. We used this summary map, all pixels' simulated phenocycles and the phenological-distance–geographical-distance regression (the slope of which serves as the asynchrony metric) to graphically depict the asynchrony calculation procedure (Extended Data Fig. 7a).

## Phenological asynchrony model covariates

For the random-forest (RF) model exploring the potential drivers of phenological asynchrony (see below), we produced rasters of physiographic and environmental covariates using workflows combining GEE, Julia, Python and GDAL (v.2.2.3)[101]. First, we applied the same harmonic regression and asynchrony-mapping pipeline described above, skipping the masking steps that were specific to LSP data quality concerns, to the 64-year TerraClimate time series dataset[102] in the GEE catalogue, generating asynchrony maps for the climatic factors potentially driving phenological asynchrony: monthly minimum and maximum temperature, monthly precipitation and monthly

climate water deficit. We supplemented this with an equivalently produced map of asynchrony in cloud cover, using cloud cover fractions calculated from the internal cloud algorithm flag bit (bit 10 of the 1 km reflectance data QA band) of the MODIS Aqua and Terra daily 1 km global surface reflectance datasets (MYD09GA.061[103] and MOD09GA.06[104]) in the GEE catalogue. The $R^2$ values from these harmonic regressions are also mapped in Extended Data Fig. 3, and the asynchrony of the climatic factors is mapped in Extended Data Fig. 7.

To model the potential importance of topographic complexity for driving phenological asynchrony, we downloaded a global map of the vector ruggedness metric[105]. We chose this over other measures of topographic complexity because of its reduced correlation with slope. We downloaded data published previously[106], choosing a map based on Global Multi-resolution Terrain Elevation Data 2010 (GMTED2010) elevation data[107] and median-aggregated at a scale on par with the neighbourhood size of our main LSP asynchrony dataset (100 km).

To allow the model to reflect phenological asynchrony between structurally distinct vegetation communities, we used GEE to create a global map of entropy in vegetation structure within 100 km neighbourhoods (hereafter, the vegetation entropy map). To do this, we used the same 20-year time series of annual MODIS IGBP 0.05° land cover[86] that we used in our LSP data-filtering workflow. We reclassed land cover into categories of forest (IGBP classes 1–5: evergreen or deciduous broadleaf or needleleaf forests and mixed forest), shrubland (IGBP classes 6 and 7: closed and open shrublands), savanna (IGBP classes 8 and 9: woody savannas and savannas), grassland (IGBP class 10) or permanent wetland (IGBP class 11). We then applied the same mask used to calculate LSP asynchrony, so that the information captured by this covariate would reflect the information included in the LSP asynchrony response variable. Next, we reduced the 20-year time series to a single map representing the modal class for each pixel across all years. Finally, we produced the covariate map by calculating the entropy of the vegetation structure classes within each pixel's 100 km radius (the neighbourhood size of our main analysis) as $-\Sigma_i P(c_i) \log_2 P(c_i)$, where $c$ is vegetation structure class and $P(c_i)$ is the proportion of the neighbourhood that is assigned class $i$.

As LSP asynchrony patterns could be influenced by human LULCC, we used GEE to create two other 100-km neighbourhood covariates: the mean proportion of subpixels classified as LULCC, and the mean frequency of fire. We derived the mean LULCC proportion map from a global, harmonized map of Landsat-resolution (30 m) land-cover change and land use in 2019[108]. In GEE, we calculated the proportion of subpixels within each of our target-resolution pixels that were classified as any land-cover change or land-use class, including classes 92–116 and 212–236 (tree cover loss since 2000, with or without regrowth), classes 240–249 (built-up land) and class 252 (cropland). We applied to that LULCC proportion map the same mask used to calculate LSP asynchrony, then calculated the mean within a 100 km radial neighbourhood for every pixel.

As the source data for the LULCC map explicitly excludes fire-driven tree cover loss, we also used GEE to produce a separate covariate map estimating the neighbourhood mean frequency of fire. To do this, for each pixel we counted the number of months with a recorded burn date in the global monthly MODIS Burned Area dataset, MCD64A1.061[109], divided that by the total number of months in the dataset, used the arithmetic mean to aggregate the map from its original 500 m resolution to our target resolution, applied the same mask applied to the map used to calculate LSP asynchrony, then calculated the mean of that fire frequency map within each pixel's 100 km radial neighbourhood.

## Modelling phenological asynchrony drivers

To explore the potential drivers of LSP asynchrony, we constructed an RF model using the R package ranger (v.0.13.1)[110] and incorporating the set of covariates described above, formulated as:

$$\text{LSP.asy}_{\text{neigh}} \sim \text{ppt.asy}_{\text{neigh}} + \text{tmp.min.asy}_{\text{neigh}} + \text{tmp.max.asy}_{\text{neigh}}$$
$$+ \text{def.asy}_{\text{neigh}} + \text{cld.asy}_{\text{neigh}} + \text{vrm.med} + \text{veg.ent}$$
$$+ \text{luc.prp.mea} + \text{brn.frq.mea}[+x+y],$$

where LSP.asy is asynchrony of the LSP dataset used in a given model (either $\text{NIR}_\text{v}$ or SIF), ppt.asy is PA, tmp.min.asy and tmp.max.asy are minimum and maximum temperature asynchrony, def.asy is climate water deficit asynchrony, cld.asy is cloud cover asynchrony, neigh indicates the asynchrony neighbourhood radius used to calculate the asynchrony metrics for a given model (50, 100 or 150 km), veg.ent is vegetation structural entropy, vrm.med is the median vector ruggedness metric, luc.prp.mea is mean proportion of LULCC, brn.frq.mea is mean fire frequency, and $x$ and $y$ are pixel longitude and latitude (within brackets to indicate their inclusion in only half of the suite of models). We chose the RF algorithm owing to its ability to robustly model nonlinear relationships, suited to our expectation that phenological asynchrony would be driven by different and potentially interacting factors in different regions of the globe. We developed a comprehensive and conservative modelling workflow, which we ran once for each combination of LSP dataset ($\text{NIR}_\text{v}$, SIF), neighbourhood radius (50 km, 100 km, 150 km), and coordinate inclusion (geographical coordinates either included or excluded as covariates). We examined the sensitivity of our RF models to the inclusion of geographical coordinates because of the lack of consensus about how to handle spatial data in RF modelling[111,112]. This produced a final set of 12 models (Extended Data Fig. 9a). As we found that salient results were largely insensitive to choice of LSP dataset, neighbourhood radius and coordinate inclusion, we chose the 100 km, $\text{NIR}_\text{v}$-based, coordinates-included model as the main model to summarize and discuss in the main text of this article.

Before producing final results, we used R v.4.0.3 to prepare the modelling data, tune hyperparameters and carry out feature selection. First, we projected the response and covariate rasters to a metric projection (EPSG:3857) to ensure that coordinates were expressed in metres, then stacked them and extracted their values at all valid (that is, non-masked) pixels. Next, we carried out comprehensive hyperparameter tuning[113], assessing model performance as a function of five RF tuning parameters (number of trees per forest: 'ntree' = 150, 200, 250, 300; fraction of observations to use in each tree, for tree decorrelation: 'sample.fraction' = 0.3, 0.55, 0.8; minimum number of observations that can be captured by a node: 'min.node.size' = 1, 3, 5, 10; size of random subset of variables from which to choose each node's split variable: 'mtry' = 1, 3, 5; and whether to sample with replacement: 'replace' = true, false) and as a function of the fraction of the full global dataset used for modelling ('subset.frac' = 0.05, 0.005; drawn as a random subsample, quartile-stratified by the LSP response variable, to reduce the computational demand imposed by the size of the modelling dataset without causing excessive information loss). We included geographical coordinates in all models used for hyperparameter tuning, as we intended to retain them in the main model unless we found that predominant results were highly sensitive to their inclusion. We used as a performance metric the root mean squared error (r.m.s.e.) of the model fitted to a 60% training split of the subsampled global dataset and found that the r.m.s.e. of the predictions made on the 40% test split yielded the same set of optimum-performance hyperparameter choices. Lastly, before running the final set of models, we confirmed that none of our subsetted datasets contained variables with a collinearity of $R^2 \geq 0.75$, and we used the Boruta feature-selection algorithm and R package (Boruta v.7.0.0)[114] to select our final feature set (but found no features that should be dropped).

We constructed the final 12 models using the optimum hyperparameters indicated by our tuning results (ntree = 300, sample.fraction = 0.8, min.node.size = 1, mtry = 5, replace = false and subset.frac = 0.05). To evaluate each model, we calculated two variable importance metrics—ranger's default permutation-based importance metric, which compares the cross-tree average accuracy of out-of-bag sample predictions to the accuracy after permuting covariate values, and the absolute SHAP values[115] summed across all predictions in a model's training dataset, calculated using the R fastshap package (v.0.0.7)[116]—as well as two metrics of overall model performance, $R^2$ and r.m.s.e. To help with spatial model assessment, we used trained models to make LSP asynchrony predictions at all global pixels, then calculated prediction error maps (Extended Data Fig. 9b shows the error map for the main model). Lastly, to aid spatial interpretability of the models, we calculated pixel-wise SHAP values and produced global SHAP maps for each covariate.

Noting low variability across models in the covariates identified as having the highest importance (Extended Data Fig. 9a), we summarized the main model (100-km $\text{NIR}_\text{v}$ asynchrony, coordinates included) in the text and estimated the predominance of the top two covariates in that model, PA (ppt.asy) and MTA (tmp.min.asy), as a normalized difference of absolute SHAP values: $\text{predom} = (|\text{SHAP}_{\text{ppt.asy}}| - |\text{SHAP}_{\text{tmp.min.asy}}|)/(|\text{SHAP}_{\text{ppt.asy}}| + |\text{SHAP}_{\text{tmp.min.asy}}|)$. We plotted a summary map of the normalized difference across global regions of high LSP asynchrony (that is, pixels ≥85th percentile), to show regional variation in the predominance or codominance of these two drivers (Fig. 2b; Extended Data Fig. 9c shows predominance across all covariates except geographical coordinates).

## Isoclimatic phenological asynchrony

To test the hypothesis that phenological asynchrony is less dependent on climatic difference at low latitudes than at higher latitudes, we performed an ensemble analysis. Each sub-analysis in the ensemble first uses clustering to delineate a global set of high-asynchrony regions, then uses matrix regressions to estimate the slope of the relationship between climatic and phenological distance (hereafter, the climate–phenology correlation) within each of those regions. We defined the sub-analyses within the ensemble using unique combinations of low, middle and high values for three hyperparameters to which our final results could exhibit sensitivity, then used Monte Carlo analysis to assess the relationship, across the ensemble, between regions' mean latitudes and the strengths of their climate–phenology correlations.

To delineate high-asynchrony regions, we first converted our $\text{NIR}_\text{v}$ LSP asynchrony map into a map of maximum asynchrony pixels by setting all pixels ≥95th percentile asynchrony value to 1 and masking everything else. We then used the density-based spatial clustering of applications with noise (DBSCAN) algorithm[117], implemented in the Python package sklearn (v.1.0.2)[76], to cluster those high-asynchrony pixels. We chose the DBSCAN algorithm owing to its ability to robustly identify clusters of arbitrary shape around the high-density centres of a point set without forcing all points to have cluster assignments, which was a good match for the noisiness of our asynchrony map. Finally, we used the alpha-complex algorithm (a straight-line edge variant of the alpha-hull algorithm), implemented in Python by the Alpha Shape Toolbox (alphashape, v.1.3.1)[118], to delineate high-asynchrony regions around those clusters. This enabled us to relax the convexity and contiguity assumptions of other hull-determination algorithms and, therefore, to flexibly delineate regions with complex shapes (for example, mountain arcs) without inevitably including all intervening geographical areas, as would occur with convex hulls.

To assess the relationship between the mean latitude of a region and the strength of the climate–phenology correlation within that region, we first standardized and stacked each of the 19 WorldClim bioclimatic variables[119] and standardized our global map of fitted phenocycles. Then, for each delineated region, we executed the following steps:
(1) Draw a set of 1,000 random points within the region that all fall within non-masked NIRv LSP pixels (or draw the maximum number of points possible, if regions are too small for 1,000 points).
(2) Calculate the matrix of pairwise phenological distances ($\text{dist}_{\text{phen}}$) between all points (as 365-dimensional pairwise Euclidean distances between phenocycles).

(3) Calculate the matrix of pairwise climatic distances ($dist_{clim}$) between all points (as 19-dimensional pairwise Euclidean distances between bioclimatic values).

(4) Calculate the matrix of pairwise geographical distances ($dist_{geog}$) between all points (as geodesic distances).

(5) Standardize all three pairwise distance matrix variables (so that coefficients of all regressions are $\beta$ coefficients and, therefore, comparable), then run MMRR[53] using the formula phenology ~ $\beta_c$ climate + $\beta_g$ geography, where $\beta_c$ and $\beta_g$ indicate the strengths of the relationships between climatic and phenological distances and between geographical and phenological distances, respectively.

To hedge against hyperparameter sensitivity, we chose reasonable ranges of low, middle and high values of the key parameters in the clustering and hull-delineation algorithms from which to compose our ensemble. The DBSCAN clustering algorithm relies on two parameters to which our results might be sensitive: 'eps' (epsilon), the maximum geographical distance between two points that can be considered to be in the same neighbourhood; and 'min_samples', the minimum number of samples required within a neighbourhood for a point to be considered as a core point. The alpha-complex algorithm has an additional parameter to which our results might be sensitive: 'alpha', a value controlling how edge members are chosen and, therefore, determining the maximum complexity of a hull's edge. To create the ensemble, we reran the full regionalization and climate–phenology correlation analysis once for each combination of the following parameter values: eps = 2, 3.5, 5; min_samples = 0.3, 0.45, 0.6; and alpha = 0.25, 0.75, 1.25.

As a final step, we summarized the ensemble results across the 27 parameterizations by running the ordinary least squares regression model $\beta_c$ ~ $\gamma_{lat}|\overline{lat}|$, using $\gamma_{lat}$ to quantify the relationship between the absolute value of the mean latitude of each cluster and the strength of its climate–phenology correlation (Fig. 3b). As this regression violates the assumption that samples of the independent variables are IID—each point represents a clustered and delineated high-asynchrony region, and those regions can overlap across distinct parameterizations of the sub-analyses—we used Monte Carlo analysis to generate an empirical $P$ value for $\gamma_{lat}$ in the ensemble linear regression model. We ran 1,000 iterations of the same regression, each time permuting the vector of $|\overline{lat}|$ values, then calculated an empirical $P$ value as the fraction of the 1,000 simulated $\gamma_{lat}$ that are at least as extreme as the observed $\gamma_{lat}$ (Fig. 3c). To provide a spatially explicit geographical interpretation of the results of this analysis, we mapped a summary of the ensemble results as a hexbin map (Fig. 3a), with the colour of each hexbin indicating the mean $\beta_c$ of all high-asynchrony regions (that is, delineated alpha hulls) overlapping the bin's hexagon.

## Allochrony by allopatry: flowering

To explore the ability of remotely sensed LSP to predict geographical variation in flowering phenology, we tested the correlation between $NIR_v$ phenocycles and dates of flowering observations for all available iNaturalist taxa with non-unimodal flowering histograms and without extremely broad latitudinal distributions. First, we used the Python API client pyinaturalist (v.0.19.0)[120] to download from iNaturalist the weekly flowering-observation histogram, and the first ≤5,000 native, non-captive, research-grade flowering observations corresponding to that histogram, for every taxon having ≥50 annotated flowering observation records at the time of download (downloads completed between 5 June 2024, 23:00 UTC and 9 June 2024, 00:00 UTC). This included a total of 7,251 taxa out of the 34,438 iNaturalist taxa with at least one observation (21.1%). We truncated the raw observation datasets to ≤5,000 per taxon to limit strain on the iNaturalist API; preliminary results showed that this decision was inconsequential because none of the 39 taxa affected would ultimately be retained for later analyses.

We further filtered the observation points for each taxon to only those with at least 1 km positional accuracy, then used the alpha-complex algorithm[118], with alpha set to 0.75 (the middle value used in our isoclimatic phenological asynchrony analysis) to fit a conservative geographical boundary (hereafter, observation range) to the set of iNaturalist observations. One taxon dropped out of our analysis at this stage because of the failure to fit an observation range. We then estimated the number of peaks in the flowering-week histogram for each taxon using the following steps:

(1) 'Rotate' the histogram so that the first instance of its minimum value moved into the first position in the vector, to avoid spurious results arising from flowering peaks that straddle the last and first weeks of the calendar year.

(2) Fit a kernel density estimation (KDE) to the histogram, using a bandwidth of 5 weeks, to reduce the noise resulting from temporal variance in observation counts.

(3) Use a simple, neighbour-comparison-based peak-search algorithm (implemented in the find_peaks function in the signal module of the Python package scipy v.1.13.0)[75] to count the number of peaks in the KDE with a height ≥60% of the overall range of values in the histogram.

(4) Calculate the absolute value of the lag-1 temporal autocorrelation in the observed KDE and in KDEs fitted to 100 permuted versions of the rotated flowering histogram.

(5) If the non-permuted KDE has an empirical $P ≤ 0.05$ (that is, if the absolute value of the lag-1 temporal autocorrelation of the non-permuted KDE is greater than that of ≥95% of the permuted KDEs), then it has a significant signal of temporal autocorrelation that probably represents non-random seasonal variability in flowering activity, so assign the counted number of peaks as the observed number of flowering-time peaks for the taxon; otherwise, assign zero as the observed number of statistically significant flowering-time peaks.

Executing this procedure for all available taxa resulted in 6391 taxa (88.2%) with unimodal flowering-time histograms and 859 non-unimodal taxa, including 123 taxa (1.7%) with bimodal histograms, one taxon with a trimodal histogram and 735 taxa (10.1%) with no statistically significant flowering-time peaks. We dropped the unimodal taxa from further analysis because they were unlikely to exhibit the sharp geographical discontinuities in flowering phenology that were our main interest. We retained the 859 non-unimodal taxa to test for significant signals of allochrony by allopatry. We summarized these results by creating a set of hexbins covering all fitted observation ranges and then mapping, for each hexagon, the proportions of taxa with zero and with ≥2 flowering-time peaks and the overall proportion of all non-unimodal taxa (Extended Data Fig. 10). To preclude significant but uninteresting results for taxa broadly distributed across latitudes, and therefore affected by the opposite seasonalities of the northern and southern hemispheres, we dropped any taxa with samples extending beyond both 10° north and south latitudes (196 taxa).

We then looked for evidence of allochrony by allopatry by testing each of the 663 remaining taxa for a correlation between intersite flowering-date distances and intersite LSP distances. To do this, we fitted an MMRR model for each taxon, specified as flowering_date ~ $\beta_{LSP}$ LSP + $\beta_C$ climate + $\beta_G$ geography, where the variables are pairwise distance matrices and $\beta_{LSP}$ and its $P$ value were our output values of interest, indicating the strength and statistical significance of the relationship between LSP and flowering date distances after accounting for environmental and geographical distances. Some non-unimodal taxa may flower opportunistically, perennially or at multiple discrete times of year within the same sites, and should therefore yield insignificant $\beta_{LSP}$ values, but taxa exhibiting the strong geographical discontinuities in flowering time that we would expect under allochrony by allopatry should yield a significant, positive $\beta_{LSP}$ value. To produce the distance covariates for this model, we calculated flowering date distances as the shorter of the two forward-time or backward-time distances between two observations' numerical day-of-year values, LSP distances as the

365-dimensional Euclidean distances between the observation sites' $NIR_v$ phenocycles, climate distances as the 19-dimensional Euclidean distances between the sites' vectors of standardized WorldClim[119] bioclimatic variables and geographical distances as the geodesic distances between sites. We corrected $\beta_{LSP} P$ values to control for the false-discovery rate (FDR) using the 'false_discovery_control' function in the 'stats' module of the Python package scipy (v.1.13.0)[75] with the Benjamini–Hochberg method. Supplementary Table 4 provides results for the 43 taxa that remained significant after FDR control (of 614 taxa successfully tested, after 49 dropped out because of insufficient data for model-fitting), and the full results from all stages of this analysis are archived with the data for this study.

To visualize the results of this analysis for an example taxon, we plotted a temporal comparison between the flowering observation dates and the flowering observation locations' min-max scaled phenocycles as well as a map of the observation locations, coloured according to k-means clustering of the phenocycles ($k = 2$) to highlight the spatial and temporal structure of the geographical discontinuity in phenology. We constructed this visualization (Fig. 4a) for two example taxa with FDR-corrected significance, chosen to demonstrate the correspondence of their patterns of allochrony by allopatry to the regional LSP patterns we had mapped and highlighted earlier in the article (*M. scabra*, in southwestern North America; *S. parviflorum*, in South Africa).

### Allochrony by allopatry: genetics
To test whether remotely sensed LSP predicts the phenologically driven isolation by time[22] that is expected to result from allochrony by allopatry, above and beyond isolation by distance[121] and isolation by environment[122], we fitted genetic MMRR models to a pair of datasets from two of the few published genetic studies of the ASH, substituting LSP distances calculated from our dataset for the authors' previously used measures of asynchronous seasonality, then compared our results to theirs. First, we gathered and prepared the genomic and geographical data from the only genomic test of the ASH of which we are aware, a study of the eastern Brazilian toad *R. granulosa*[25]. We used the R package adegenet (v.2.1.5)[123,124] and data downloaded from the Dryad repository for that study (https://datadryad.org/stash/dataset/doi:10.5061/dryad.pc866t1p4) to calculate a pairwise genetic distance matrix for 80 samples collected from 51 localities, based on the Euclidean distance between allele frequencies at 7,674 independent single-nucleotide polymorphism loci. We calculated geographical- and LSP-distance matrices as described above, using the geographical coordinates of each sample, and prepared a climatic distance matrix using the Euclidean distances between standardized versions of the four WorldClim[119] bioclimatic variables used in the original study: annual mean temperature (BIO1), temperature seasonality (BIO4), annual precipitation (BIO12) and precipitation seasonality (BIO15). Five samples fell within masked pixels in our LSP dataset and thus could not be included in our analysis, yielding a final sample size of 75. We fit an MMRR model specified as genetic $\sim \beta_{LSP}$ LSP $+ \beta_C$ climate $+ \beta_G$ geography, then compared our results to the results presented in table 4 of ref. 25. To visualize our findings we used k-means clustering with Euclidean distances to divide the samples into $k = 2$ clusters, first clustering by $NIR_v$ phenocycles, then a second time clustering by genetic distance vectors. We then prepared side-by-side equivalent plots showing sample localities and their min–max-scaled phenocycles, coloured by either of those clusterings, providing a simple visual indication of the extent to which our LSP map recapitulates the observed genetic structure (Fig. 4b (top)).

To explore whether disparately related, sympatric taxa might exhibit similar patterns of isolation by asynchrony, we repeated the same procedure for the only other sympatric genetic dataset that we could find within previous studies of the ASH: cytochrome B sequencing data for the lesser woodcreeper (*X. fuscus*; Furnariidae)[32]. We first downloaded sample location data from the Zenodo archive for the study

(http://zenodo.org/records/5012226)[125] and the FASTA-formatted sample sequence data from GenBank. We aligned sequences using ClustalW (v.2.1)[126] with the default parameter settings and then used jModelTest2 (v.2.1.10)[127] to compare the fit of 44 models of sequence evolution to the sequence data. We then calculated pairwise genetic distances under the best-fit model (TVM + G), identified with AICc scores, using MEGA X (v.10.1.7)[128]. We then followed the same steps as for the *R. granulosa* data, except that we used all 19 WorldClim variables. Our sample size was reduced to 31 because three sampling sites fell within masked LSP pixels. The results are visualized in Fig. 4b (bottom).

### Allochrony by allopatry: coffee harvest
To test for significant agreement between the harvest season map produced by the National Federation of Coffee Growers of Colombia (Federación Nacional de Cafeteros de Colombia, or Fedecafé) and our LSP map, we constructed a permutation-based test of an index of similarity between the harvest categories in the Fedecafé map and the categories resulting from clustering on $NIR_v$ phenocycles. First, we used WebPlotDigitizer[98] to digitize and save a set of sampling points within each of the four harvest season colours displayed in a previously published Fedecafé map[54]. Next, we used Python to extract $NIR_v$ phenocycles at all unmasked pixels coinciding with those points and then used k-means clustering to cluster all extracted phenocycles into four clusters. We then calculated the Jaccard index[129] of this cluster assignment vis-a-vis the Fedecafé harvest season assignment as:

$$J = n_{both}/(n_{Fedecafé} + n_{LSP} + n_{both}),$$

where $n_{Fedecafé}$ is a count of pairwise point comparisons that have the same assignment only within the Fedecafé map, $n_{LSP}$ is a count of those that have the same assignment only within the LSP clustering and $n_{both}$ is a count of those that have the same assignment in both datasets. Finally, we executed the same operation 1,000 times, each time first permuting the relationship between the sampling points and their phenocycles, generating a set of null $J$ values against which to calculate an empirical $P$ value for the observed $J$ value (as the fraction of the 1,000 simulated $J$ values that are at least as large as the observed $J$). We visualize the overall agreement between the Fedecafé map and ours by plotting sampling points on top of the RGB composite from the LSP EOF analysis (but not transformed across the ITCZ, given the colour-warping this causes within this region) and using colour to match the harvest season assignments of the sampling points to a series of line plots of their median, 10th percentile and 90th percentile phenocycles in our dataset (Fig. 4c).

### Reporting summary
Further information on research design is available in the Nature Portfolio Reporting Summary linked to this article.

## Data availability
Input datasets (Supplementary Table 1) are publicly available and were accessed using the following resources: MODIS MCD43A4 v061 surface reflectance (https://developers.google.com/earth-engine/datasets/catalog/MODIS_061_MCD43A4), OCO-2 SIF (https://daac.ornl.gov/VEGETATION/guides/Global_High_Res_SIF_OCO2.html), TROPOMI SIF (https://doi.org/10.22002/D1.1347), MODIS MCD12C1.061 annual land cover (https://developers.google.com/earth-engine/datasets/catalog/MODIS_061_MCD12C1), PhenoCam NDVI (accessed using R package phenocamapi: https://github.com/PhenoCamNetwork/phenocamapi), FLUXNET GPP (https://fluxnet.org/data/), percentage of annual herbaceous cover in the Great Basin (https://doi.org/10.5066/P9VL3LD5), TerraClimate (https://developers.google.com/earth-engine/datasets/catalog/IDAHO_EPSCOR_TERRACLIMATE), MODIS Aqua and Terra surface reflectance cloud bands

(https://developers.google.com/earth-engine/datasets/catalog/ MODIS_061_MYD09GA, https://developers.google.com/earth-engine/ datasets/catalog/MODIS_061_MOD09GA), EarthEnv topographic complexity (http://www.earthenv.org/topography), Global Land Analysis & Discovery global land cover and land use 2019 (https://glad.umd. edu/dataset/global-land-cover-land-use-v1), MODIS MCD64A1.v061 monthly burned area (https://developers.google.com/earth-engine/ datasets/catalog/MODIS_061_MCD64A1), WorldClim bioclimatic variables (https://www.worldclim.org/data/worldclim21.html), *R. granulosa* single-nucleotide polymorphism data (https://datadryad.org/stash/ dataset/doi:10.5061/dryad.pc866t1p4)[130], *Xiphorhynchus fuscus* cytochrome B sequencing data (http://zenodo.org/records/5012226)[125] and Fedecafé Colombian coffee harvest season map data (digitized from https://doi.org/10.19053/20275137.3200). All data supporting the findings of this study are archived at Zenodo (https://doi.org/10.5281/ zenodo.15654956)[131]. A GEE app provides the ability to explore the LSP modelling method, the global LSP map displayed in Fig. 1 and the global LSP asynchrony map displayed in Fig. 2a; it is demonstrated in Extended Data Fig. 2e and is linked in our GitHub repository (https://github.com/ erthward/phen_asynch; https://doi.org/10.5281/zenodo.15671259)[92].

## Code availability

All custom code and details of the computing environments used to run it are published in this project's GitHub repository (https://github.com/ erthward/phen_asynch, https://doi.org/10.5281/zenodo.15671259)[92].

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

**Acknowledgements** We thank D. Ackerly, L. Anderegg, A. Bishop, D. Buhrman, M. Davis, T. Dawson, D. Ehrenfeld, J. Evans, K. Fesenmyer, H. Flores Moreno, S. Fortney, J. Frederick, N. Graham, A. Hoskins, M. Kelly, M. Kling, N. Knezek, N. Muchhala, P. Papper, S. Rollins, M. Terasaki Hart, A. Turner, M. Tylka, E. Westeen, G. Wogan, P. Wright and M. Yuan for feedback and guidance on this project; C. Paciorek and the staff at Berkeley Research Computing for providing access to the Savio computing cluster and for their ongoing support with troubleshooting; M. Calderón, I. Escalante Meza, B. Garrigós, T. Gode, E. Hollenbeck, B. Lewis, J. Powell, R. Quirós Flores, F. Spooner and the staff of Estación Biológica Las Cruces, Estación Biológica Palo Verde, Cloudbridge Nature Reserve and Estación Biológica Monteverde for assistance during the field season that was the origin of this work; the PhenoCam Network collaborators, FLUXNET collaborators and iNaturalist users who shared their data, as well as iNaturalist user M. Podas, whose *X. fuscus* photograph, under a CC BY-SA 4.0 license (https://creativecommons.org/licenses/by-sa/4.0/) was segmented, flipped horizontally and lightened for use in Fig. 4b. This work is dedicated to J. Hart, whose time ended too soon but whose curiosity lives on. D.E.T.H. was supported by an Emerging Challenges in Tropical Science graduate student fellowship from the Organization for Tropical Studies funded by the Christiane and Christopher Tyson and Rudy and Sally Ruggles Fellowships, by a Tinker Field Research Grant from the UC Berkeley Center for Latin American Studies, by a research equipment grant from IdeaWild, by a Berkeley Fellowship and in part by a Bezos Earth Fund grant to The Nature Conservancy. I.J.W. was supported by a National Science Foundation grant (DEB1845682) and the USDA National Institute of Food and Agriculture (Hatch project 1024618). Some data used in this research were provided by the PhenoCam Network, which has been supported by the National Science Foundation, the Long-Term Agroecosystem Research (LTAR) network, which is supported by the US Department of Agriculture (USDA), the US Department of Energy, the US Geological Survey, the Northeastern States Research Cooperative and the USA National Phenology Network.

**Author contributions** D.E.T.H. conceived the study, gathered and prepared the data, designed and developed the analyses, prepared the results and figures, and wrote the manuscript. L.D.M. and T.-N.B. helped to gather and prepare the data, develop the analyses and figures, and write the paper. I.J.W. helped to conceive the study, design the analyses and write the manuscript.

**Funding** Open access funding provided by CSIRO Library Services.

**Competing interests** The authors declare no competing interests.

**Additional information**
**Correspondence and requests for materials** should be addressed to Drew E. Terasaki Hart.

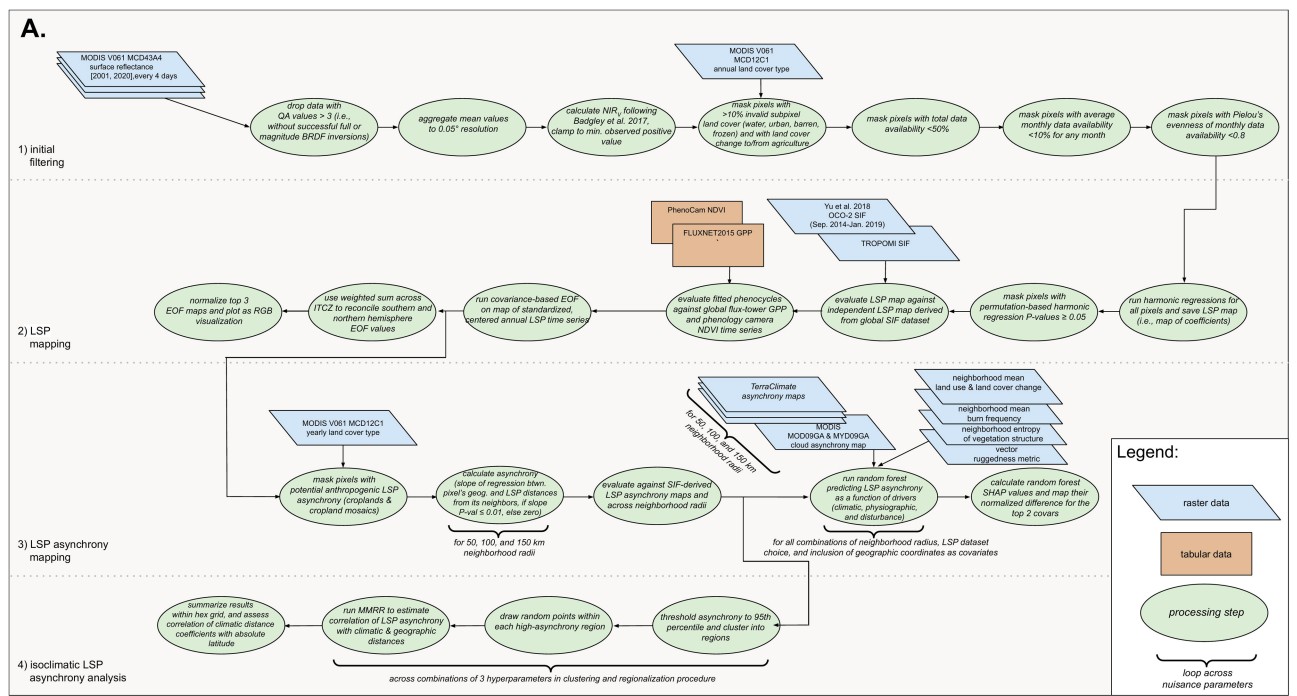

**B.**

| mask | North America | South America | Europe | Africa | Asia | Oceania |
|---|---|---|---|---|---|---|
| **land cover (ag in black)** | 0.261 (0.314) | 0.129 (0.162) | 0.179 (0.391) | 0.426 (0.460) | 0.291 (0.366) | 0.105 (0.131) |
| **monthly data availability** | 0.473 | 0.027 | 0.421 | 0.019 | 0.320 | 0.020 |
| **data availability evenness** | 0.460 | 0.020 | 0.418 | 0.016 | 0.315 | 0.010 |
| **total data availability** | 0.045 | 0.024 | 0.033 | 0.014 | 0.023 | 0.010 |
| **regression significance** | 0.054 | 0.017 | 0.035 | 0.014 | 0.032 | 0.011 |

**Extended Data Fig. 1 | Overview of mapping methods and data masking.**
A. Workflow for all global mapping analyses. B. Maps of data dropped from all analyses (in red) and additionally omitted from asynchrony analyses to avoid anthropogenic asynchrony in agricultural land cover (black) that could confound evolutionary analyses, with colour-matched table reporting cumulative masked proportions of total continental land areas.

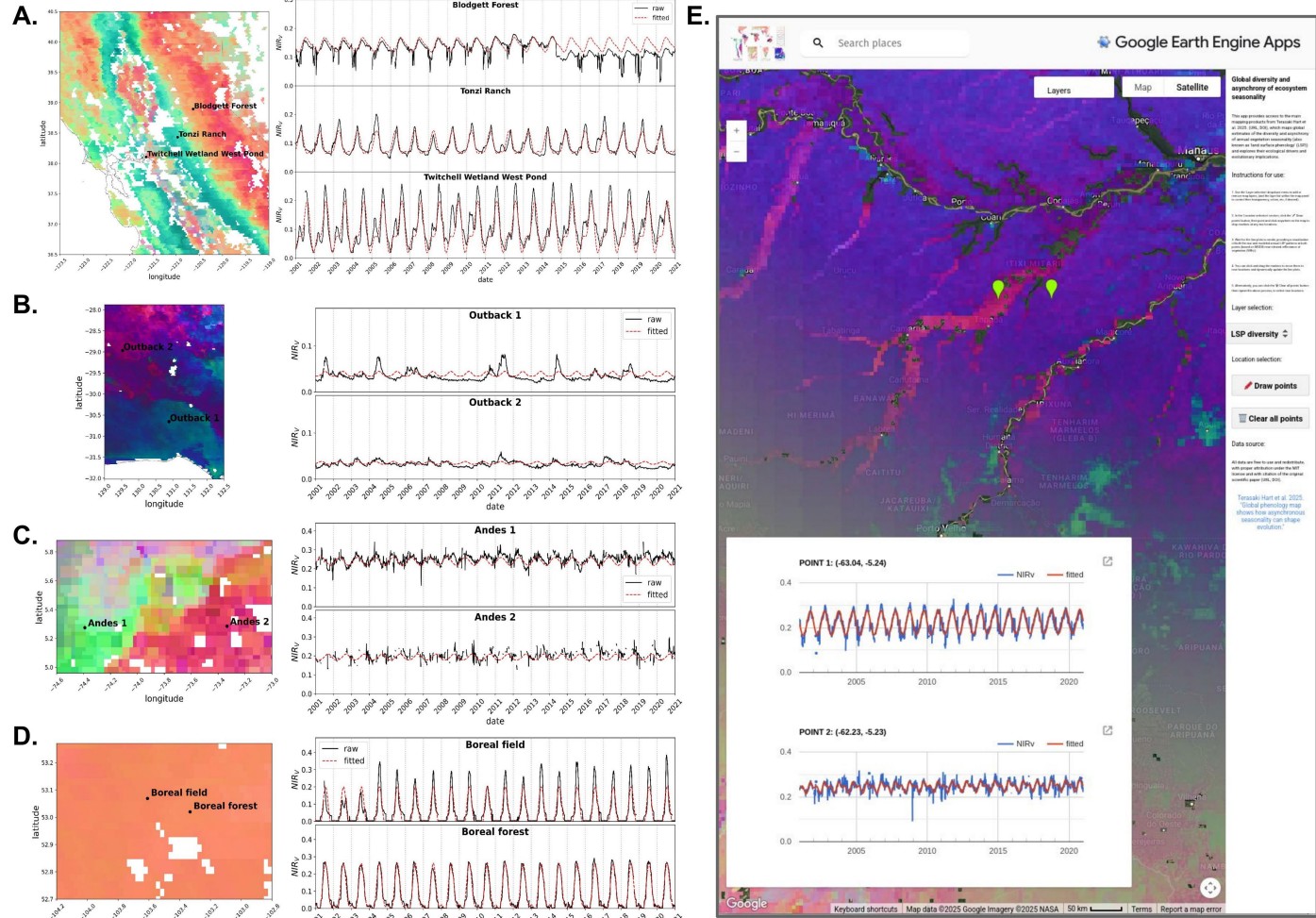

**Extended Data Fig. 2 | LSP model-fitting examples across biomes.**
A. Mediterranean climate: Results of the harmonic regression procedure
are shown at FLUXNET tower sites distributed across three ecosystems
displaying the characteristic 'double peak' LSP pattern across the Californian
Mediterranean climate region[40]: forest (the Blodgett Forest site), oak savanna
(Tonzi Ranch), and wetland (Twitchell Wetland West Pond). The map on the left
shows the same RGB composite as Fig. 1. The plots on the right show each site's
original, 20-year NIR$_v$ time series (solid black line) and fitted phenocycle
(dashed red line). The abrupt reduction in amplitude in mid-2014 at Blodgett
Forest, an experimental forestry plot, likely reflects a land use or land cover
change event, but manual inspection of the MCD12C1 land cover product used
for our data filtering workflow suggests that the on-ground activity was not
intensive enough to register a change in mapped land cover type. B. Arid
climate: Results at two sites in the southern Australian Outback, displayed
identically to A. Our model assigns divergent phenocycles to these two areas:
the Nullarbor Plain, along the coast of South Australia (Outback 1), has a
winter-peaking pattern likely influenced by the Mediterranean monsoonal
climate, whereas the inland desert (Outback 2) has a fall-peaking pattern likely
responding to the summer monsoon[38] – a rough analogue of the LSP gradient
we describe in the southwestern USA and northwestern Mexico (Fig. 1a). The
Outback's rainfall-driven, globally exceptional interannual variability in
productivity[48] results in interannual variability in the timing and size of NIR$_v$
peaks. C. Tropical montane climate: Results at two sites in the Colombian
Andes, displayed identically to A. Tropical montane regions challenge remote

sensing of phenology because of frequent cloud cover and small intra-annual
variability in productivity that yields a low signal:noise ratio. Nevertheless the
regions surrounding these sites exhibit spatially coherent phenocycles that,
despite their proximity, peak nearly six months apart, paralleling the extreme
allochrony by allopatry that we document in the complex geography of
Colombian coffee harvest seasonality (Fig. 4c). D. Boreal climate: Results at
two sites in Saskatchewan, Canada, displayed identically to A. Long periods
of snow cover in treeless land cause annually recurring stretches of invalid
negative NIR$_v$ values, which would lead to low data availability and extensive
data dropout across cleared boreal lands. Backfilling of negative values with
the minimum positive value observed in a pixel's time series – causing the
winter 'flatlining' visible in the 'Boreal field' plot – allows us to retain these
areas and fit reasonable phenocycles, revealing the expected synchrony with
neighbouring forest, where winter values are extremely low but not negative.
E. Readers can recreate and explore the results of our harmonic regression
procedure using a Google Earth Engine app (link available at: https://github.
com/erthward/phen_asynch, https://doi.org/10.5281/zenodo.15671259).
The original NIR$_v$ time series is depicted in blue and the fitted phenocycle in
red for a pair of sites indicated by the map markers. The results depicted here
reveal a marked difference between the pronounced, unimodal annual
phenology observed in the floodplain of the Purus River, an Amazon tributary
(upper line plot), and the bimodal phenology observed in nearby upland forest
(lower line plot), likely reflecting the strong control of annual flooding over the
annual phenologies of floodplain habitats[8].

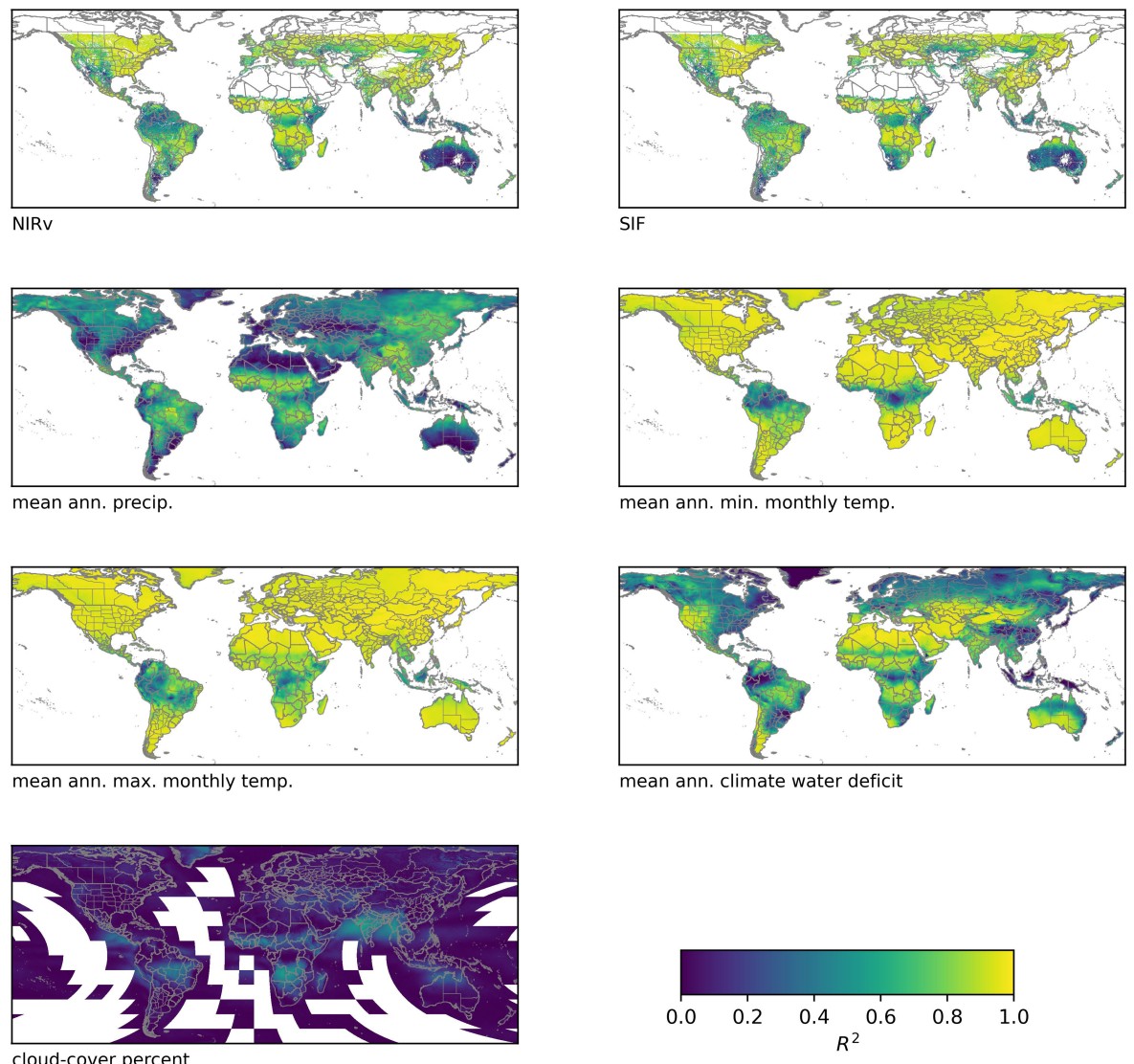

NIRv

SIF

mean ann. precip.

mean ann. min. monthly temp.

mean ann. max. monthly temp.

mean ann. climate water deficit

cloud-cover percent

$R^2$

**Extended Data Fig. 3 | Harmonic regression performance.** Maps of the $R^2$ values from the harmonic regressions estimating annual LSP maps (top row) and climatic seasonality maps (remaining rows).

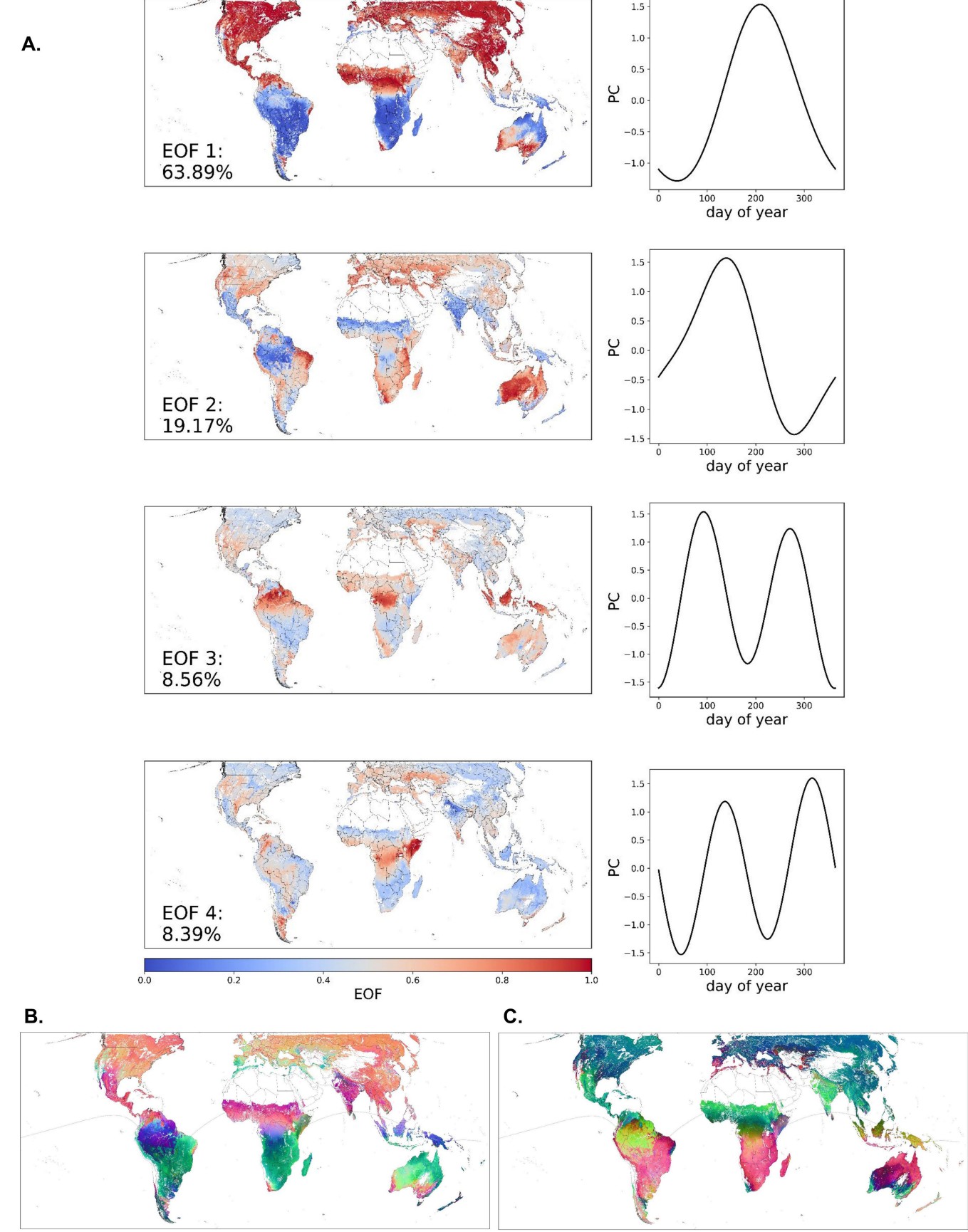

**Extended Data Fig. 4** | See next page for caption.

**Extended Data Fig. 4 | LSP EOF results and visualization.** A. Raw LSP EOF maps: Maps on the left show the top four modes of annual LSP spatiotemporal variability according to an empirical orthogonal function (EOF) analysis, and line plots on the right show the annual temporal variation of the principal components (PCs) corresponding to each EOF. EOF values are standardized and centred on zero, and maps are ordered by decreasing percent total variance explained, from top to bottom. B and C. Non-transformed RGB LSP maps: The non-transformed values of all three top EOF modes are depicted as RGB values (B), and are subtracted from 1.0 and then depicted as RGB values (C). These two representations are the latitudinally-agnostic maps from which we derived the transformation presented in Fig. 1 by computing a weighted sum of these two maps, with weights varying in a piecewise function from 1.0 to 0.0 north to south across the ITCZ (dotted black line straddling the equator).

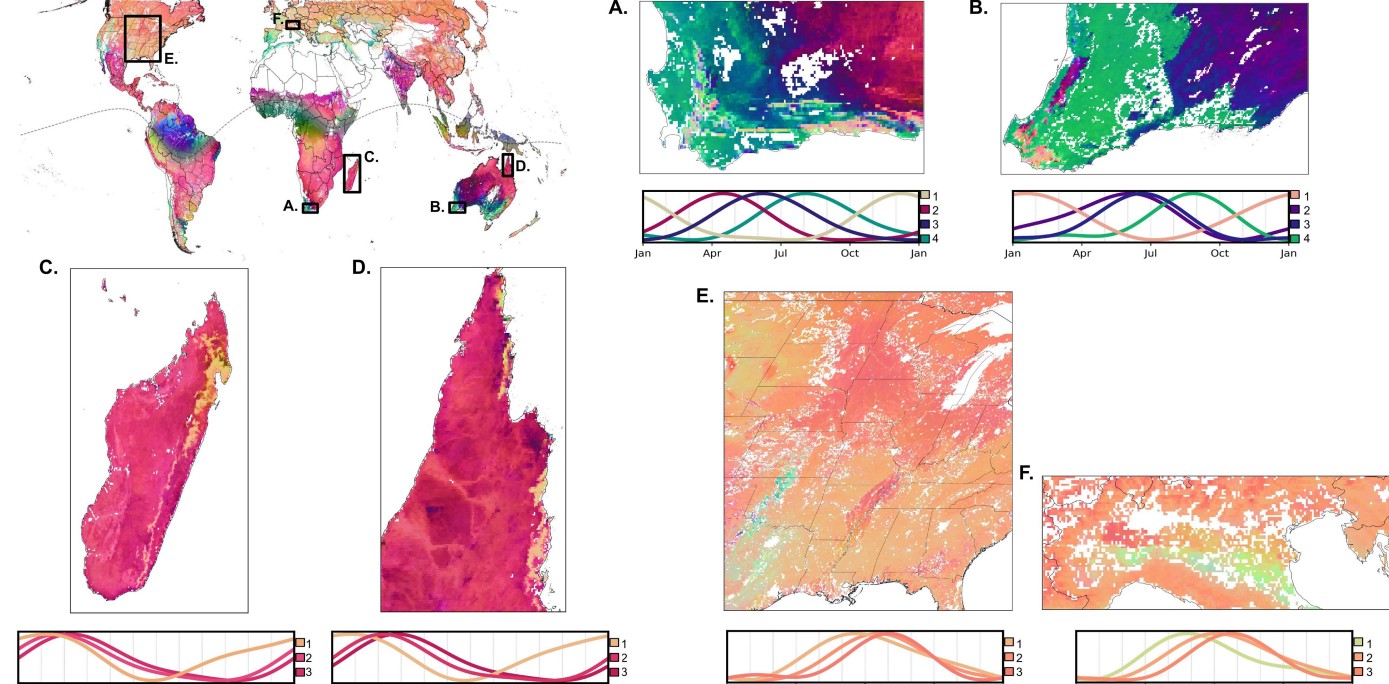

**Extended Data Fig. 5 | Intercontinental convergence in LSP.** We observed a striking convergence between the LSP gradients in Earth's two more climatically moderate Mediterranean climate regions[39]: the Cape Region of South Africa (A) and southern and southwestern Australia (B). In both regions, small areas of moist habitat (colour 1 in both panels) show summer-peaking phenologies, contrasting with the progression of peaks observed across the broader regional aridity gradients (colours 2–4). We also observed convergent phenological gradients across coastal-inland aridity gradients in two southern tropical regions: Madagascar (C) and the Cape York Peninsula of Queensland, Australia (D) show a one-to-two month delay between the summer-peaking phenologies of coastal and montane rainforests (colour 1 in both panels) and the fall-peaking phenologies observed across drier habitats and non-forest (colours 2 and 3). Agricultural mosaics in northern-hemisphere continental climates also display convergence, including the Corn Belt region of the USA (E) and northern Italy (F), where deciduous and montane forest regions (colour 2 in both panels) exhibit characteristic phenologies that peak around July, roughly one month ahead of the characteristically delayed peaks[18] in prominent maize-producing areas (colour 3) but about one-and-a-half months after the peaks in other agricultural areas and in non-forest regions (colour 1).

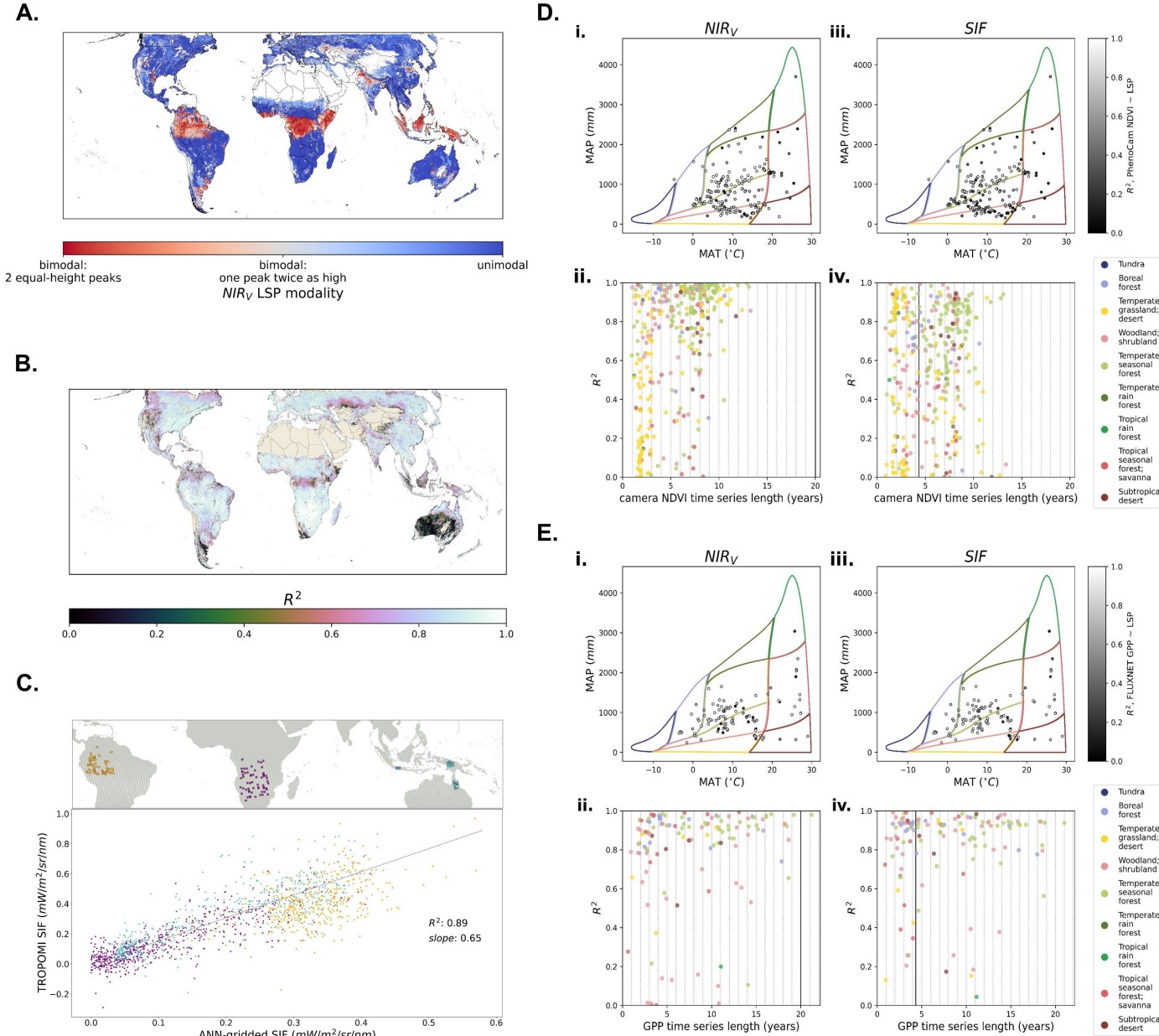

**Extended Data Fig. 6 | LSP map evaluation.** A. LSP modality. The $NIR_V$ phenocycle at each pixel is depicted on a spectrum from strongly bimodal (two peaks of equal height; red) to weakly bimodal (two peaks, one twice as high as the other; grey) to unimodal (a single peak per year; blue). The pattern depicted here is a close match to previously published maps of regions with single versus double growing seasons (e.g., Fig. 3 in Garonna et al.[4]). B. Agreement between $NIR_V$ and SIF phenocycles. This map depicts each pixel's $R^2$ between the phenocycles fitted to its $NIR_V$ and SIF time series. Tan pixels are terrestrial locations that have been masked because of invalid land cover or insufficient data quality. C. Orbital-gap assessment of seasonality in interpolated Orbiting Carbon Observatory 2 (OCO-2) data. Above: Map showing locations of random points in three tropical regions (South America in orange, Africa in purple, Indo-Pacific and Australia in blue), chosen to fall within OCO-2 orbital gaps (n = 180 points; 60 per region). Below: For each sampled point in each of the three regions we plot all contemporaneous estimates from the ANN-gridded OCO-2 SIF dataset used in our asynchrony maps and from an independent TROPOspheric Monitoring Instrument (TROPOMI) SIF dataset (n = 1550 available contemporaneous estimates). An intercept-free OLS regression depicts the significant level of agreement between these two independent sets of measurements (model P-value < 5×10$^{-324}$). D. Evaluation of fitted phenocycles

against NDVI time series at PhenoCam ground phenology cameras. i. and iii. Each phenology camera site (n = 368 sites) is plotted in the environmental space defined by mean annual temperature (MAT) and mean annual precipitation (MAP), with the Whittaker biomes[96] plotted beneath for context. Sites are coloured by the $R^2$ (scaled from 0=black to 1=white) between: 1.) the annual cycle fitted to the site's three-day-summary time series of camera-derived normalized difference vegetation index (NDVI), averaged across all regions of interest within the camera's field of view to approximate the spatial averaging that occurs within a co-located remote sensing pixel; and 2.) the phenocycle fitted to the remotely sensed $NIR_V$ (i.) or SIF (iii.) data at a site (or up to two map pixels away, ~11 km). ii. and iv. Phenology camera evaluation performance ($R^2$) across camera NDVI time series lengths, for both the $NIR_V$ (ii.) and SIF (iv.) datasets, with points coloured by Whittaker biome. Black vertical lines indicate the time series lengths of our LSP datasets (20 years for $NIR_V$; $4\frac{1}{3}$ years for SIF), for comparison. E. Evaluation of fitted phenocycles against GPP time series at FLUXNET eddy covariance flux towers. Visualization is identical to D, but depicts the strength of correlation between daily gross primary productivity (GPP) time series collected at FLUXNET2015 flux towers (n = 170 sites) and LSP phenocycles.

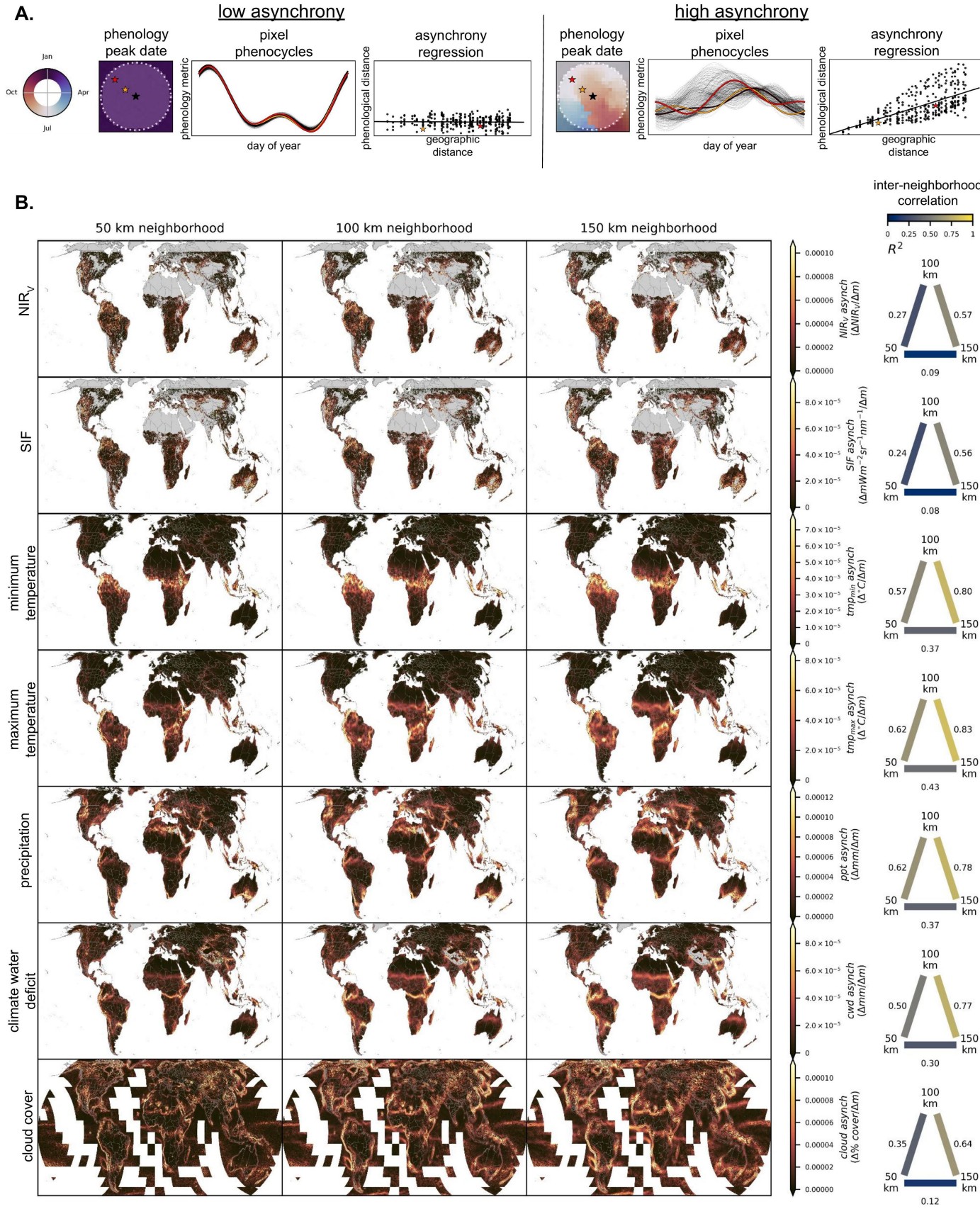

**Extended Data Fig. 7** | See next page for caption.

**Extended Data Fig. 7 | Asynchrony calculation and mapping.** A. Conceptual diagram depicting the stepwise calculation of our spatial phenological asynchrony metric, in regions of both low asynchrony (left) and high (right). Maps depict spatial heterogeneity in day of the year corresponding to peak phenology (circular colorbar provided at far left). Central focal pixel (black star) is the pixel for which asynchrony is calculated, using an analysis based on pairwise comparisons between the focal pixel and each other pixel inside the focal pixel's neighbourhood (white dashed circle). Line plots show phenocycles pertaining to each of the neighbour pixels inside the circular neighbourhood, with the focal pixel shown in bold black. Scatter plots depict the relationship between pairwise geographic distances and pairwise phenological distances for all comparisons between the focal pixel and its neighbours. The slope of the trend line fitted to the scatter plot by simple linear regression is taken as the focal pixel's asynchrony metric. Neighbour pixels at lesser (orange) and greater (red) geographic distances from the focal pixel are tracked across the plots, illustrating how phenological distance increases with geographic distance when asynchrony is high. B. Maps showing the results of calculating the asynchrony metric presented in subpanel A. for all LSP and climatic variables (rows) and across all three neighbourhood radii (columns). Triangle plots (far right) show each variable's map correlations ($R^2$ values) for all three inter-neighbourhood comparisons.

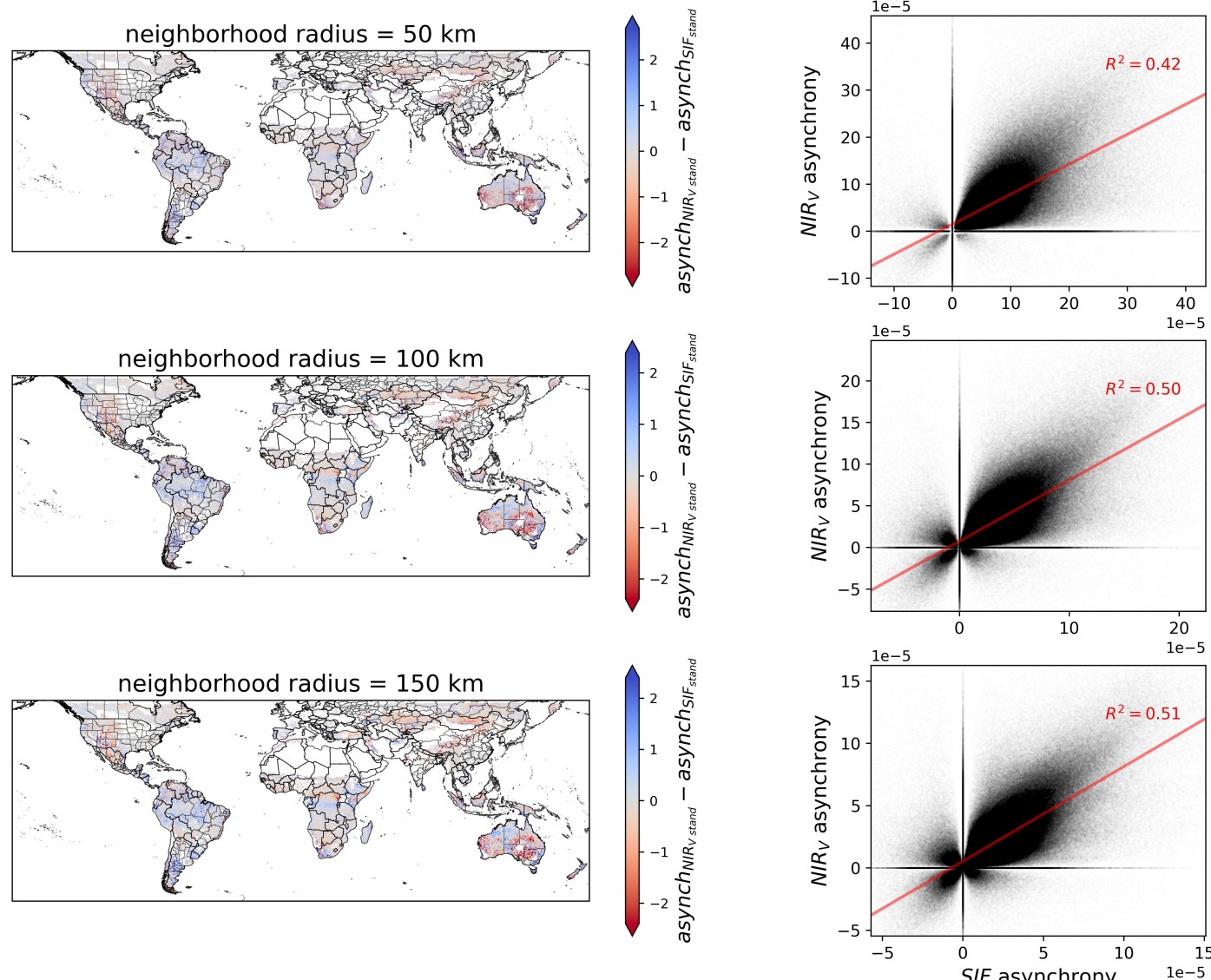

**Extended Data Fig. 8 | NIR$_V$-SIF LSP asynchrony map comparison.** Maps show the pixelwise differences between standardized NIR$_V$- and SIF-derived LSP asynchrony maps, across neighbourhood radii (top row: 50 km; middle: 100 km; bottom: 150 km). Scatter plots at the right depict the correlation between NIR$_V$ and SIF LSP asynchrony map pixel values, with an OLS regression drawn as a red line and its $R^2$ value indicated. Slope P-value $< 5 \times 10^{-324}$ for all three regressions.

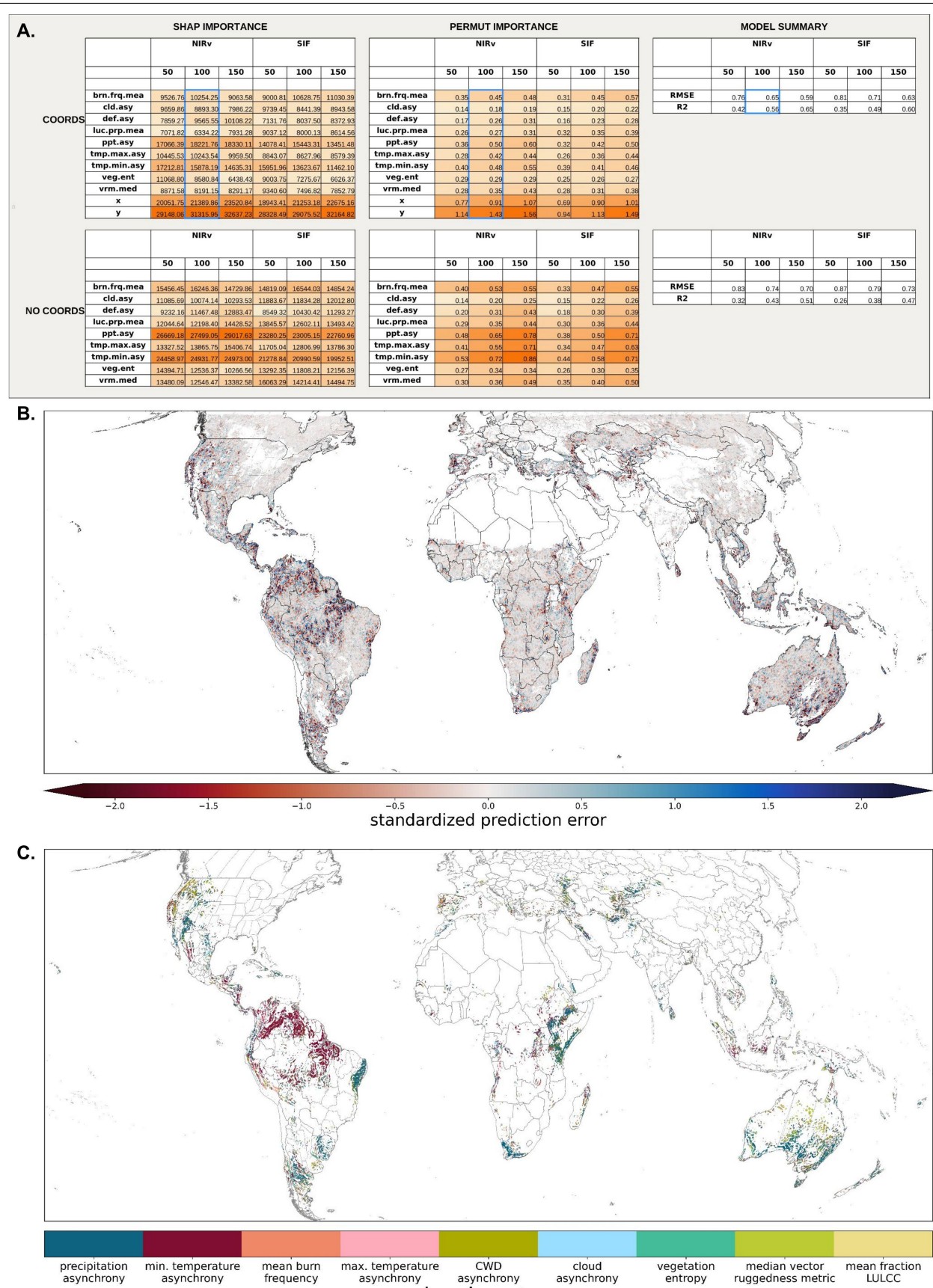

**A.**

**SHAP IMPORTANCE** — COORDS

| | NIRv 50 | NIRv 100 | NIRv 150 | SIF 50 | SIF 100 | SIF 150 |
|---|---|---|---|---|---|---|
| brn.frq.mea | 9526.76 | 10254.25 | 9063.58 | 9000.81 | 10628.75 | 11030.39 |
| cld.asy | 9659.86 | 8893.30 | 7986.22 | 9739.45 | 8441.39 | 8943.58 |
| def.asy | 7859.27 | 9565.55 | 10108.22 | 7131.76 | 8037.50 | 8372.93 |
| luc.prp.mea | 7071.82 | 6334.22 | 7931.28 | 9037.12 | 8000.13 | 8614.56 |
| ppt.asy | 17066.39 | 18221.76 | 18330.11 | 14078.41 | 15443.31 | 13451.48 |
| tmp.max.asy | 10445.53 | 10243.54 | 9959.70 | 8843.07 | 8627.96 | 8579.39 |
| tmp.min.asy | 17212.81 | 15878.19 | 14635.31 | 15951.96 | 13623.67 | 11462.10 |
| veg.ent | 11068.80 | 8580.84 | 6438.43 | 9003.75 | 7275.67 | 6626.37 |
| vrm.med | 8871.58 | 8191.17 | 8291.17 | 9340.60 | 7496.82 | 7852.79 |
| x | 20051.75 | 21389.86 | 23520.84 | 18943.41 | 21253.18 | 22675.16 |
| y | 29148.06 | 31315.95 | 32637.23 | 28328.49 | 29075.52 | 32164.82 |

**PERMUT IMPORTANCE** — COORDS

| | NIRv 50 | NIRv 100 | NIRv 150 | SIF 50 | SIF 100 | SIF 150 |
|---|---|---|---|---|---|---|
| brn.frq.mea | 0.35 | 0.45 | 0.48 | 0.31 | 0.45 | 0.57 |
| cld.asy | 0.14 | 0.18 | 0.19 | 0.15 | 0.20 | 0.22 |
| def.asy | 0.17 | 0.26 | 0.31 | 0.16 | 0.23 | 0.28 |
| luc.prp.mea | 0.26 | 0.31 | 0.31 | 0.32 | 0.35 | 0.39 |
| ppt.asy | 0.36 | 0.50 | 0.60 | 0.32 | 0.42 | 0.50 |
| tmp.max.asy | 0.28 | 0.42 | 0.44 | 0.26 | 0.36 | 0.44 |
| tmp.min.asy | 0.40 | 0.48 | 0.55 | 0.39 | 0.41 | 0.46 |
| veg.ent | 0.29 | 0.29 | 0.29 | 0.25 | 0.26 | 0.27 |
| vrm.med | 0.28 | 0.35 | 0.43 | 0.28 | 0.31 | 0.38 |
| x | 0.77 | 0.91 | 1.07 | 0.69 | 0.90 | 1.01 |
| y | 1.14 | 1.43 | 1.56 | 0.94 | 1.13 | 1.49 |

**MODEL SUMMARY** — COORDS

| | NIRv 50 | NIRv 100 | NIRv 150 | SIF 50 | SIF 100 | SIF 150 |
|---|---|---|---|---|---|---|
| RMSE | 0.76 | 0.65 | 0.59 | 0.81 | 0.71 | 0.63 |
| R2 | 0.42 | 0.56 | 0.65 | 0.35 | 0.49 | 0.60 |

**SHAP IMPORTANCE** — NO COORDS

| | NIRv 50 | NIRv 100 | NIRv 150 | SIF 50 | SIF 100 | SIF 150 |
|---|---|---|---|---|---|---|
| brn.frq.mea | 15456.45 | 16246.36 | 14729.86 | 14819.09 | 16544.03 | 14854.24 |
| cld.asy | 11085.69 | 10074.14 | 10293.53 | 11883.67 | 11834.28 | 12012.80 |
| def.asy | 9232.16 | 11467.48 | 12883.47 | 8549.32 | 10430.42 | 11293.27 |
| luc.prp.mea | 12044.64 | 12198.40 | 14428.52 | 13845.57 | 12602.11 | 13493.42 |
| ppt.asy | 26669.18 | 27499.05 | 29017.63 | 23280.25 | 23005.15 | 22760.96 |
| tmp.max.asy | 13327.52 | 13865.75 | 15406.74 | 11705.04 | 12806.99 | 13786.30 |
| tmp.min.asy | 24458.97 | 24931.77 | 24973.00 | 21278.84 | 20990.59 | 19952.51 |
| veg.ent | 14394.71 | 12536.37 | 10266.56 | 13292.35 | 11808.21 | 12156.39 |
| vrm.med | 13480.09 | 12546.47 | 13382.58 | 16063.29 | 14214.41 | 14494.75 |

**PERMUT IMPORTANCE** — NO COORDS

| | NIRv 50 | NIRv 100 | NIRv 150 | SIF 50 | SIF 100 | SIF 150 |
|---|---|---|---|---|---|---|
| brn.frq.mea | 0.40 | 0.53 | 0.55 | 0.33 | 0.47 | 0.55 |
| cld.asy | 0.14 | 0.20 | 0.25 | 0.15 | 0.22 | 0.26 |
| def.asy | 0.20 | 0.31 | 0.43 | 0.18 | 0.30 | 0.39 |
| luc.prp.mea | 0.29 | 0.35 | 0.44 | 0.30 | 0.36 | 0.44 |
| ppt.asy | 0.48 | 0.65 | 0.78 | 0.38 | 0.50 | 0.71 |
| tmp.max.asy | 0.41 | 0.55 | 0.71 | 0.34 | 0.47 | 0.63 |
| tmp.min.asy | 0.53 | 0.72 | 0.86 | 0.44 | 0.58 | 0.71 |
| veg.ent | 0.27 | 0.34 | 0.34 | 0.26 | 0.30 | 0.35 |
| vrm.med | 0.30 | 0.36 | 0.49 | 0.35 | 0.40 | 0.50 |

**MODEL SUMMARY** — NO COORDS

| | NIRv 50 | NIRv 100 | NIRv 150 | SIF 50 | SIF 100 | SIF 150 |
|---|---|---|---|---|---|---|
| RMSE | 0.83 | 0.74 | 0.70 | 0.87 | 0.79 | 0.73 |
| R2 | 0.32 | 0.43 | 0.51 | 0.26 | 0.38 | 0.47 |

**B.** standardized prediction error

**C.** predominant covar — precipitation asynchrony, min. temperature asynchrony, mean burn frequency, max. temperature asynchrony, CWD asynchrony, cloud asynchrony, vegetation entropy, median vector ruggedness metric, mean fraction LULCC

**Extended Data Fig. 9** | See next page for caption.

**Extended Data Fig. 9 | Modelling of LSP asynchrony drivers.** A. Ensemble results of random forest modelling of LSP asynchrony drivers. Set of colorized tables depicts variable importance, both SHAP-based (left tables) and permutation-based (centre), as well as overall model performance ($R^2$ and root mean squared error; right) for models either including (top tables) or excluding (bottom) geographic coordinates as covariates, and for models using both the $NIR_v$ and SIF phenology metrics and using all three neighbourhood radii (nested columns within tables). Darker orange hues indicate higher relative covariate importance. Abbreviations are: neighbourhood mean burn frequency (*brn.frq.mea*), fractional cloud cover asynchrony (*cld.asy*), asynchrony of monthly climate water deficit (*def.asy*), neighbourhood mean proportion of land use and land cover change sub-pixels (*luc.prp.mea*), asynchrony of monthly precipitation (*ppt.asy*), asynchrony of monthly minimum and maximum temperatures (*tmp.min.asy* and *tmn.max.asy*), and longitude and latitude (*x* and *y*). B. Map showing the standardized LSP asynchrony prediction errors for the main model (the model whose results are outlined in blue in subpanel A: $NIR_v$-based LSP, 100 km neighbourhood radius, with geographic coordinates included as covariates). C. Map depicting predominance in main model of all covariates except geographic coordinates, within global regions of high LSP asynchrony (≥90th percentile). Precipitation asynchrony and minimum temperature asynchrony, already the focus of Fig. 2b, are here depicted in darker hues to allow better discrimination between the remaining covariates.

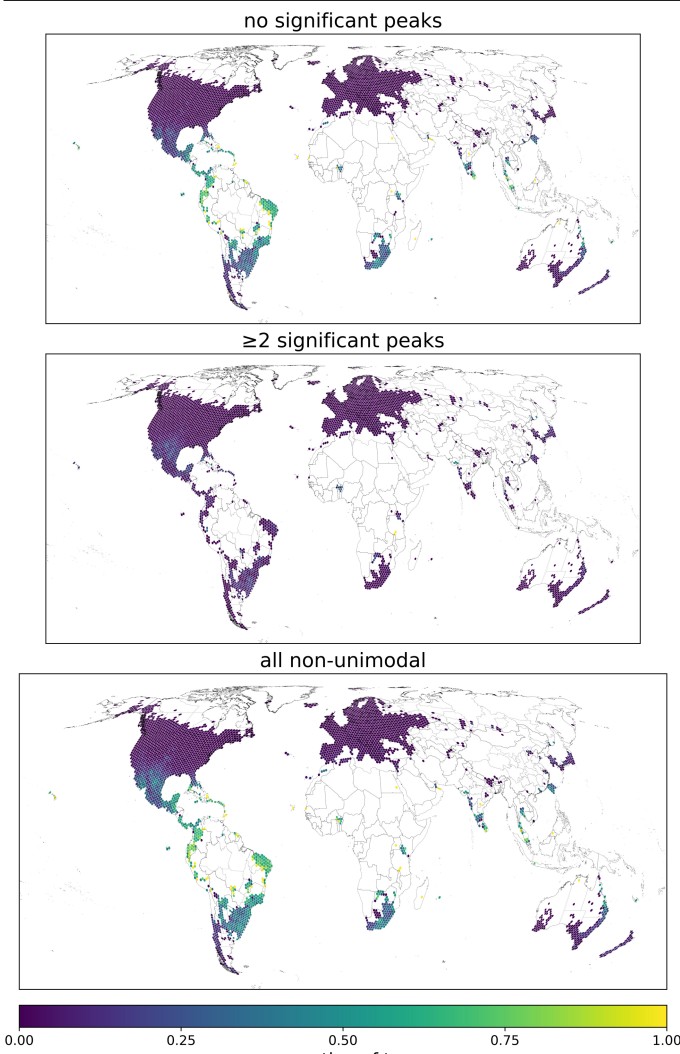

**Extended Data Fig. 10 | Global distribution of non-unimodal flowering taxa from iNaturalist.** Hexbin maps showing the global distribution of iNaturalist taxa with flowering-date histograms that are non-unimodal, calculated as the proportion of all tested taxa whose 'observation ranges' (i.e., alpha hulls fitted to all observation points) overlap each hexbin. Maps show the proportions of taxa with histograms that have no significant temporal flowering peaks (top), taxa with histograms that have two or more significant peaks (middle), and taxa exhibiting any form of non-unimodality (bottom).

# Reporting Summary

## Statistics

For all statistical analyses, confirm that the following items are present in the figure legend, table legend, main text, or Methods section.

| n/a | Confirmed | |
|---|---|---|
| ☐ | ☒ | The exact sample size (*n*) for each experimental group/condition, given as a discrete number and unit of measurement |
| ☐ | ☒ | A statement on whether measurements were taken from distinct samples or whether the same sample was measured repeatedly |
| ☐ | ☒ | The statistical test(s) used AND whether they are one- or two-sided *Only common tests should be described solely by name; describe more complex techniques in the Methods section.* |
| ☐ | ☒ | A description of all covariates tested |
| ☐ | ☒ | A description of any assumptions or corrections, such as tests of normality and adjustment for multiple comparisons |
| ☐ | ☒ | A full description of the statistical parameters including central tendency (e.g. means) or other basic estimates (e.g. regression coefficient) AND variation (e.g. standard deviation) or associated estimates of uncertainty (e.g. confidence intervals) |
| ☐ | ☒ | For null hypothesis testing, the test statistic (e.g. *F*, *t*, *r*) with confidence intervals, effect sizes, degrees of freedom and *P* value noted *Give P values as exact values whenever suitable.* |
| ☒ | ☐ | For Bayesian analysis, information on the choice of priors and Markov chain Monte Carlo settings |
| ☒ | ☐ | For hierarchical and complex designs, identification of the appropriate level for tests and full reporting of outcomes |
| ☒ | ☐ | Estimates of effect sizes (e.g. Cohen's *d*, Pearson's *r*), indicating how they were calculated |

*Our web collection on statistics for biologists contains articles on many of the points above.*

## Software and code

Policy information about availability of computer code

| Data collection | Custom Bash (version 4) and Python (3.7) scripts were used to facilitate download of some SIF data and upload of that data to Google Earth Engine, and custom R (4.0) and Python (3.9) scripts were used to download PhenoCam and iNaturalist data. |
|---|---|

| Data analysis | Data analysis used custom code written in a variety of languages and using a variety of packages on three different computing environments: 1. Google Earth Engine (via version >=0.1.404 of the browser-hosted Javascript API); 2. a local computing environment, including Pop!_OS (22.04 LTS), Bash (5.1.6), GDAL (3.3.1), Python (3.9, with numpy (1.22.4), rasterio (1.2.10), xarray (2022.3.0), rioxarray (0.15.0), pandas (2.2.1), shapely (2.0.3), geopandas (0.14.3), zipfile36 (0.1.3), tensorflow (2.4.1), pyproj (3.6.1), cartopy (0.20.2), geopy (1.13.0), scipy (1.13.0), sklearn (1.0.2), alphashape (1.3.1), rasterstats (0.16.0), statsmodels (0.13.2), seaborn (0.11.2), fuzzywuzzy (0.18.0), Bio (1.79), pyinaturalist (0.19.0), json (2.0.9), nlmpy (no version), matplotlib (3.7.0), h3 (3.7.4), decartes (no version), contextily (1.2.0), xyzservices (2022.4.0), palettable (3.3.0), cmocean (2.0), cmcrameri (1.8), colormap (1.0.4), imageio (2.19.0), and cv2 (4.9.0)), R (4.0.5, with adegent (2.1.5), ape (5.6.2), and phenocamapi (0.1.5)), and MAFFT (7.520); 3. a high-performance computing environment on UC Berkeley's Savio compute cluster, including Scientific Linux (7.9), Bash (4.2.26), GDAL (2.2.3), Python (3.7, with numpy (1.21.5), rasterio (1.1.5), xarray (0.20.2), rioxarray (0.9.1), pandas (1.3.5), geopandas (0.8.1), json (2.0.9), tensorflow (2.3.1), affine (2.3.0), haversine (no version), eofs (1.4.0), scipy (1.4.1), sklearn (0.21.3), and matplotlib (3.1.1)), R (4.0.3, with dplyr (1.0.8), rgdal (1.5.18), sp (1.4.6), sf (0.9.7), raster (3.4.5), maps (3.4.0), rsample (0.1.1), ranger (0.13.1), Boruta (7.0.0), fastshap (0.0.7), vip (0.3.2), pdp (0.7.0), ggplot2 (3.3.5), ggthemes (4.2.4), grid (4.0.3), and RColorBrewer (1.1.2)), and Julia (1.4.1, with Distributed (no version), OrderedCollections (1.4.1), StaticArrays (1.3.5), Glob (1.3.0), TFRecord (0.1.0), JSON (0.21.3), ArchGDAL (0.7.4), Distances (0.10.7), NearestNeighbors (0.4.9), Statistics (no version), StatsBase (0.33.16), GLM (1.6.1), and Colors (0.12.8)).<br><br>All custom code and details of the computing environments used to run it are published in this project's GitHub repository (https://github.com/erthward/phen_asynch). |
|---|---|

For manuscripts utilizing custom algorithms or software that are central to the research but not yet described in published literature, software must be made available to editors and reviewers. We strongly encourage code deposition in a community repository (e.g. GitHub). See the Nature Portfolio guidelines for submitting code & software for further information.

# Data

Policy information about availability of data

All manuscripts must include a data availability statement. This statement should provide the following information, where applicable:
- Accession codes, unique identifiers, or web links for publicly available datasets
- A description of any restrictions on data availability
- For clinical datasets or third party data, please ensure that the statement adheres to our policy

Input datasets (Supplementary Table 1) are publicly available and were accessed using the following resources: MODIS MCD43A4 v061 surface reflectance (https://developers.google.com/earth-engine/datasets/catalog/MODIS_061_MCD43A4), OCO-2 sun-induced chlorophyll fluorescence (https://daac.ornl.gov/VEGETATION/guides/Global_High_Res_SIF_OCO2.html), TROPOMI SIF (https://doi.org/10.22002/D1.1347), MODIS MCD12C1.061 annual land cover (https://developers.google.com/earth-engine/datasets/catalog/MODIS_061_MCD12C1), PhenoCam normalized difference vegetation index (accessed using R package phenocamapi: https://github.com/PhenoCamNetwork/phenocamapi), FLUXNET gross primary productivity (https://fluxnet.org/data/), percent annual herbaceous cover in the Great Basin (https://doi.org/10.5066/P9VL3LD5), TerraClimate (https://developers.google.com/earth-engine/datasets/catalog/IDAHO_EPSCOR_TERRACLIMATE), MODIS Aqua and Terra surface reflectance cloud bands (https://developers.google.com/earth-engine/datasets/catalog/MODIS_061_MYD09GA, https://developers.google.com/earth-engine/datasets/catalog/MODIS_061_MOD09GA), EarthEnv topographic complexity (http://www.earthenv.org/topography), Global Land Analysis & Discovery global land cover and land use 2019 (https://glad.umd.edu/dataset/global-land-cover-land-use-v1), MODIS MCD64A1.v061 monthly burned area (https://developers.google.com/earth-engine/datasets/catalog/MODIS_061_MCD64A1), WorldClim bioclimatic variables (https://www.worldclim.org/data/worldclim21.html), iNaturalist flowering observations (accessed using Python package ipynaturalist, https://github.com/pyinat/pyinaturalist), Rhinella granulosa single nucleotide polymorphism data (https://datadryad.org/stash/dataset/doi:10.5061/dryad.pc866t1p4), Xiphorhynchus fuscus cytochrome B sequence data (http://zenodo.org/records/5012226), and Fedecafé Colombian coffee harvest season map data (digitized from https://doi.org/10.19053/20275137.3200). All data supporting the findings of this study are archived with Zenodo (DOI: https://doi.org/10.5281/zenodo.15654956) 127. A Google Earth Engine App provides the ability to explore the LSP modeling method, the global LSP map displayed in Fig. 1, and the global LSP asynchrony map displayed in Fig. 2A; it is demonstrated in Extended Data Fig. 2E and is linked in our GitHub repository (https://github.com/erthward/phen_asynch).

# Research involving human participants, their data, or biological material

Policy information about studies with human participants or human data. See also policy information about sex, gender (identity/presentation), and sexual orientation and race, ethnicity and racism.

| Reporting on sex and gender | not applicable |
|---|---|
| Reporting on race, ethnicity, or other socially relevant groupings | not applicable |
| Population characteristics | not applicable |
| Recruitment | not applicable |
| Ethics oversight | not applicable |

Note that full information on the approval of the study protocol must also be provided in the manuscript.

# Field-specific reporting

Please select the one below that is the best fit for your research. If you are not sure, read the appropriate sections before making your selection.

☐ Life sciences ☐ Behavioural & social sciences ☒ Ecological, evolutionary & environmental sciences

For a reference copy of the document with all sections, see nature.com/documents/nr-reporting-summary-flat.pdf

# Life sciences study design

All studies must disclose on these points even when the disclosure is negative.

| | |
|---|---|
| Sample size | *Describe how sample size was determined, detailing any statistical methods used to predetermine sample size OR if no sample-size calculation was performed, describe how sample sizes were chosen and provide a rationale for why these sample sizes are sufficient.* |
| Data exclusions | *Describe any data exclusions. If no data were excluded from the analyses, state so OR if data were excluded, describe the exclusions and the rationale behind them, indicating whether exclusion criteria were pre-established.* |
| Replication | *Describe the measures taken to verify the reproducibility of the experimental findings. If all attempts at replication were successful, confirm this OR if there are any findings that were not replicated or cannot be reproduced, note this and describe why.* |
| Randomization | *Describe how samples/organisms/participants were allocated into experimental groups. If allocation was not random, describe how covariates were controlled OR if this is not relevant to your study, explain why.* |
| Blinding | *Describe whether the investigators were blinded to group allocation during data collection and/or analysis. If blinding was not possible, describe why OR explain why blinding was not relevant to your study.* |

# Behavioural & social sciences study design

All studies must disclose on these points even when the disclosure is negative.

| | |
|---|---|
| Study description | *Briefly describe the study type including whether data are quantitative, qualitative, or mixed-methods (e.g. qualitative cross-sectional, quantitative experimental, mixed-methods case study).* |
| Research sample | *State the research sample (e.g. Harvard university undergraduates, villagers in rural India) and provide relevant demographic information (e.g. age, sex) and indicate whether the sample is representative. Provide a rationale for the study sample chosen. For studies involving existing datasets, please describe the dataset and source.* |
| Sampling strategy | *Describe the sampling procedure (e.g. random, snowball, stratified, convenience). Describe the statistical methods that were used to predetermine sample size OR if no sample-size calculation was performed, describe how sample sizes were chosen and provide a rationale for why these sample sizes are sufficient. For qualitative data, please indicate whether data saturation was considered, and what criteria were used to decide that no further sampling was needed.* |
| Data collection | *Provide details about the data collection procedure, including the instruments or devices used to record the data (e.g. pen and paper, computer, eye tracker, video or audio equipment) whether anyone was present besides the participant(s) and the researcher, and whether the researcher was blind to experimental condition and/or the study hypothesis during data collection.* |
| Timing | *Indicate the start and stop dates of data collection. If there is a gap between collection periods, state the dates for each sample cohort.* |
| Data exclusions | *If no data were excluded from the analyses, state so OR if data were excluded, provide the exact number of exclusions and the rationale behind them, indicating whether exclusion criteria were pre-established.* |
| Non-participation | *State how many participants dropped out/declined participation and the reason(s) given OR provide response rate OR state that no participants dropped out/declined participation.* |
| Randomization | *If participants were not allocated into experimental groups, state so OR describe how participants were allocated to groups, and if allocation was not random, describe how covariates were controlled.* |

# Ecological, evolutionary & environmental sciences study design

All studies must disclose on these points even when the disclosure is negative.

| | |
|---|---|
| Study description | We use remote sensing archives of MODIS near infrared reflectance of vegetation and sun-induced chlorophyll fluorescence to assess global patterns of land surface phenological diversity and spatial phenological asynchrony, to explore potential climatic and physiographic drivers of that asynchrony, and to test whether that asynchrony is more correlated with between-site climatic differences at higher latitudes than in the tropics. We then used iNaturalist flowering observation data, previously published genomic and genetic data, and a map of Colombian coffee harvest seasonality to demonstrate the utility of land surface phenology for |

predicting species-level spatial 'allochrony by allopatry', i.e., phenological asynchrony and genetic divergence as a function of the difference in seasonal timing between populations.

| | |
|---|---|
| Research sample | The global land surface phenology datasets we present are derived from the MODIS land surface reflectance archive and (for validation of the MODIS data) a neural-net interpolated map of sun-induced chlorophyll fluorescence from the Orbiting Carbon Observatory 2. The samples in our global random forest analyses predicting the spatial asynchrony of land surface phenology are random samples chosen to represent our full global phenological dataset and quartile-stratified by the response variable. iNaturalist samples are opportunistic, voluntarily reported and annotated records of flowering, and genomic and genetic samples were determined by the designs of the studies from which they are derived. |
| Sampling strategy | In our random forest analyses, sample sizes (expressed as a fraction of the size of the full global dataset) was tuned as a hyperparameter during our model-development process (details described in methods). In our ensemble of permutation-based matrix regressions, sample sizes were up to 1000 points drawn within each high-asynchrony region (reduced due to dropout of any randomly drawn points that did not fall within valid pixels in the land surface phenology dataset). The samples in our ensemble of permutation-based matrix regressions are random samples of up to 1000 points drawn within the high-asynchrony regions defined by the regionalization algorithm described in the methods. Samples in the iNaturalist flowering phenological analysis included all available samples (as of the noted data of download) for all taxa passing a series of eligibility requirements. Samples for the genomic and genetic analyses derive from the only previously published genomic study of the Asynchrony of Seasons Hypothesis (ASH) and the only sympatric species among the few previously published genetic studies of the ASH. Samples for the Colombian coffee analysis were drawn across the entire area of a previously published map of coffee harvest seasonality. |
| Data collection | All input data comes from publicly available data archives and is documented in previous peer-reviewed publications. |
| Timing and spatial scale | Our main phenological dataset (MODIS near infrared reflectance of vegetation) provides data for every fourth day during the period from 2001/01/01 to 2020/12/31, providing us enough data to estimate the characteristic annual land surface phenology pattern for a location while remaining computable on Google Earth Engine. Our validation dataset (Orbiting Carbon Observatory 2-derived sun-induced chlorophyll fluorescence) provides twice-monthly data covering the period from 2014/09/01 to 2018/12/31, which was the longest available such archive at an acceptably high resolution at the time this study began. |
| Data exclusions | Pixels were excluded from the land surface phenology datasets if they contained >10% coverage of invalid land cover (e.g., permanent snow and ice, barren, or water bodies, where no true phenological pattern would be expected); if they were subject to land cover change that could contaminate the characteristic phenology being fitted (i.e., transitioned to/from agriculture); if they were missing >50% of potential data overall, >90% of data in any month (across years), or had a Pielou's data availability evenness of <0.8 (calculation described in methods), to prevent fitting of spurious phenological peaks during long, seasonally recurring periods of low data coverage, such as we observed in very high latitudes; or if permutation testing indicated that the R^2 of their fitted phenological harmonic regression was not significant (details provided in methods). To avoid sensitivity of downstream analyses to anthropogenic phenology patterns, all agricultural pixels were excluded from asynchrony maps and asynchrony-based analyses. |
| Reproducibility | Our results are not based on experiments, so experimental reproducibility could not be verified. However, we went to lengths to evaluate our findings against a second, independent remote sensing dataset, to assess the accuracy of that evaluation dataset itself, to tune the hyperparameters of our random forest models, and to cross-check our random forest modeling results across combinations of key' modeling decisions (choice between two response-variable datasets; choice of neighborhood within which to calculate pixels spatial asynchrony values; and choice of whether or not to include geographic coordinates as model features). |
| Randomization | Our study did not allocate units to groups. However, we used random subsampling when constructing predictive models on large, dense spatial datasets (details described in methods). |
| Blinding | Blinding was not relevant to our study, as sampling units were not assigned to treatments and controls. |

Did the study involve field work?   ☐ Yes   ☒ No

# Field work, collection and transport

| | |
|---|---|
| Field conditions | *Describe the study conditions for field work, providing relevant parameters (e.g. temperature, rainfall).* |
| Location | *State the location of the sampling or experiment, providing relevant parameters (e.g. latitude and longitude, elevation, water depth).* |
| Access & import/export | *Describe the efforts you have made to access habitats and to collect and import/export your samples in a responsible manner and in compliance with local, national and international laws, noting any permits that were obtained (give the name of the issuing authority, the date of issue, and any identifying information).* |
| Disturbance | *Describe any disturbance caused by the study and how it was minimized.* |

# Reporting for specific materials, systems and methods

We require information from authors about some types of materials, experimental systems and methods used in many studies. Here, indicate whether each material, system or method listed is relevant to your study. If you are not sure if a list item applies to your research, read the appropriate section before selecting a response.

## Materials & experimental systems

| n/a | Involved in the study |
|---|---|
| ☒ ☐ | Antibodies |
| ☒ ☐ | Eukaryotic cell lines |
| ☒ ☐ | Palaeontology and archaeology |
| ☒ ☐ | Animals and other organisms |
| ☒ ☐ | Clinical data |
| ☒ ☐ | Dual use research of concern |
| ☒ ☐ | Plants |

## Methods

| n/a | Involved in the study |
|---|---|
| ☒ ☐ | ChIP-seq |
| ☒ ☐ | Flow cytometry |
| ☒ ☐ | MRI-based neuroimaging |

# Antibodies

| | |
|---|---|
| Antibodies used | *Describe all antibodies used in the study; as applicable, provide supplier name, catalog number, clone name, and lot number.* |
| Validation | *Describe the validation of each primary antibody for the species and application, noting any validation statements on the manufacturer's website, relevant citations, antibody profiles in online databases, or data provided in the manuscript.* |

# Eukaryotic cell lines

Policy information about cell lines and Sex and Gender in Research

| | |
|---|---|
| Cell line source(s) | *State the source of each cell line used and the sex of all primary cell lines and cells derived from human participants or vertebrate models.* |
| Authentication | *Describe the authentication procedures for each cell line used OR declare that none of the cell lines used were authenticated.* |
| Mycoplasma contamination | *Confirm that all cell lines tested negative for mycoplasma contamination OR describe the results of the testing for mycoplasma contamination OR declare that the cell lines were not tested for mycoplasma contamination.* |
| Commonly misidentified lines (See ICLAC register) | *Name any commonly misidentified cell lines used in the study and provide a rationale for their use.* |

# Palaeontology and Archaeology

| | |
|---|---|
| Specimen provenance | *Provide provenance information for specimens and describe permits that were obtained for the work (including the name of the issuing authority, the date of issue, and any identifying information). Permits should encompass collection and, where applicable, export.* |
| Specimen deposition | *Indicate where the specimens have been deposited to permit free access by other researchers.* |
| Dating methods | *If new dates are provided, describe how they were obtained (e.g. collection, storage, sample pretreatment and measurement), where they were obtained (i.e. lab name), the calibration program and the protocol for quality assurance OR state that no new dates are provided.* |

☐ Tick this box to confirm that the raw and calibrated dates are available in the paper or in Supplementary Information.

| | |
|---|---|
| Ethics oversight | *Identify the organization(s) that approved or provided guidance on the study protocol, OR state that no ethical approval or guidance was required and explain why not.* |

Note that full information on the approval of the study protocol must also be provided in the manuscript.

# Animals and other research organisms

Policy information about studies involving animals; ARRIVE guidelines recommended for reporting animal research, and Sex and Gender in Research

| | |
|---|---|
| Laboratory animals | *For laboratory animals, report species, strain and age OR state that the study did not involve laboratory animals.* |
| Wild animals | *Provide details on animals observed in or captured in the field; report species and age where possible. Describe how animals were caught and transported and what happened to captive animals after the study (if killed, explain why and describe method; if released, say where and when) OR state that the study did not involve wild animals.* |
| Reporting on sex | *Indicate if findings apply to only one sex; describe whether sex was considered in study design, methods used for assigning sex. Provide data disaggregated for sex where this information has been collected in the source data as appropriate; provide overall* |

*numbers in this Reporting Summary. Please state if this information has not been collected. Report sex-based analyses where performed, justify reasons for lack of sex-based analysis.*

Field-collected samples | *For laboratory work with field-collected samples, describe all relevant parameters such as housing, maintenance, temperature, photoperiod and end-of-experiment protocol OR state that the study did not involve samples collected from the field.*

Ethics oversight | *Identify the organization(s) that approved or provided guidance on the study protocol, OR state that no ethical approval or guidance was required and explain why not.*

Note that full information on the approval of the study protocol must also be provided in the manuscript.

# Clinical data

Policy information about clinical studies

All manuscripts should comply with the ICMJE guidelines for publication of clinical research and a completed CONSORT checklist must be included with all submissions.

Clinical trial registration | *Provide the trial registration number from ClinicalTrials.gov or an equivalent agency.*

Study protocol | *Note where the full trial protocol can be accessed OR if not available, explain why.*

Data collection | *Describe the settings and locales of data collection, noting the time periods of recruitment and data collection.*

Outcomes | *Describe how you pre-defined primary and secondary outcome measures and how you assessed these measures.*

# Dual use research of concern

Policy information about dual use research of concern

## Hazards

Could the accidental, deliberate or reckless misuse of agents or technologies generated in the work, or the application of information presented in the manuscript, pose a threat to:

No | Yes
☐ | ☐ Public health
☐ | ☐ National security
☐ | ☐ Crops and/or livestock
☐ | ☐ Ecosystems
☐ | ☐ Any other significant area

## Experiments of concern

Does the work involve any of these experiments of concern:

No | Yes
☐ | ☐ Demonstrate how to render a vaccine ineffective
☐ | ☐ Confer resistance to therapeutically useful antibiotics or antiviral agents
☐ | ☐ Enhance the virulence of a pathogen or render a nonpathogen virulent
☐ | ☐ Increase transmissibility of a pathogen
☐ | ☐ Alter the host range of a pathogen
☐ | ☐ Enable evasion of diagnostic/detection modalities
☐ | ☐ Enable the weaponization of a biological agent or toxin
☐ | ☐ Any other potentially harmful combination of experiments and agents

## Plants

Seed stocks
> not applicable

Novel plant genotypes
> not applicable

Authentication
> not applicable

## ChIP-seq

### Data deposition

☐ Confirm that both raw and final processed data have been deposited in a public database such as GEO.

☐ Confirm that you have deposited or provided access to graph files (e.g. BED files) for the called peaks.

Data access links
*May remain private before publication.*
> *For "Initial submission" or "Revised version" documents, provide reviewer access links. For your "Final submission" document, provide a link to the deposited data.*

Files in database submission
> *Provide a list of all files available in the database submission.*

Genome browser session
(e.g. UCSC)
> *Provide a link to an anonymized genome browser session for "Initial submission" and "Revised version" documents only, to enable peer review. Write "no longer applicable" for "Final submission" documents.*

### Methodology

Replicates
> *Describe the experimental replicates, specifying number, type and replicate agreement.*

Sequencing depth
> *Describe the sequencing depth for each experiment, providing the total number of reads, uniquely mapped reads, length of reads and whether they were paired- or single-end.*

Antibodies
> *Describe the antibodies used for the ChIP-seq experiments; as applicable, provide supplier name, catalog number, clone name, and lot number.*

Peak calling parameters
> *Specify the command line program and parameters used for read mapping and peak calling, including the ChIP, control and index files used.*

Data quality
> *Describe the methods used to ensure data quality in full detail, including how many peaks are at FDR 5% and above 5-fold enrichment.*

Software
> *Describe the software used to collect and analyze the ChIP-seq data. For custom code that has been deposited into a community repository, provide accession details.*

## Flow Cytometry

### Plots

Confirm that:

☐ The axis labels state the marker and fluorochrome used (e.g. CD4-FITC).

☐ The axis scales are clearly visible. Include numbers along axes only for bottom left plot of group (a 'group' is an analysis of identical markers).

☐ All plots are contour plots with outliers or pseudocolor plots.

☐ A numerical value for number of cells or percentage (with statistics) is provided.

### Methodology

Sample preparation
> *Describe the sample preparation, detailing the biological source of the cells and any tissue processing steps used.*

Instrument
> *Identify the instrument used for data collection, specifying make and model number.*

Software
> *Describe the software used to collect and analyze the flow cytometry data. For custom code that has been deposited into a community repository, provide accession details.*

| Cell population abundance | *Describe the abundance of the relevant cell populations within post-sort fractions, providing details on the purity of the samples and how it was determined.* |
|---|---|
| Gating strategy | *Describe the gating strategy used for all relevant experiments, specifying the preliminary FSC/SSC gates of the starting cell population, indicating where boundaries between "positive" and "negative" staining cell populations are defined.* |

☐ Tick this box to confirm that a figure exemplifying the gating strategy is provided in the Supplementary Information.

# Magnetic resonance imaging

## Experimental design

| Design type | *Indicate task or resting state; event-related or block design.* |
|---|---|
| Design specifications | *Specify the number of blocks, trials or experimental units per session and/or subject, and specify the length of each trial or block (if trials are blocked) and interval between trials.* |
| Behavioral performance measures | *State number and/or type of variables recorded (e.g. correct button press, response time) and what statistics were used to establish that the subjects were performing the task as expected (e.g. mean, range, and/or standard deviation across subjects).* |

## Acquisition

| Imaging type(s) | *Specify: functional, structural, diffusion, perfusion.* |
|---|---|
| Field strength | *Specify in Tesla* |
| Sequence & imaging parameters | *Specify the pulse sequence type (gradient echo, spin echo, etc.), imaging type (EPI, spiral, etc.), field of view, matrix size, slice thickness, orientation and TE/TR/flip angle.* |
| Area of acquisition | *State whether a whole brain scan was used OR define the area of acquisition, describing how the region was determined.* |

Diffusion MRI     ☐ Used     ☐ Not used

## Preprocessing

| Preprocessing software | *Provide detail on software version and revision number and on specific parameters (model/functions, brain extraction, segmentation, smoothing kernel size, etc.).* |
|---|---|
| Normalization | *If data were normalized/standardized, describe the approach(es): specify linear or non-linear and define image types used for transformation OR indicate that data were not normalized and explain rationale for lack of normalization.* |
| Normalization template | *Describe the template used for normalization/transformation, specifying subject space or group standardized space (e.g. original Talairach, MNI305, ICBM152) OR indicate that the data were not normalized.* |
| Noise and artifact removal | *Describe your procedure(s) for artifact and structured noise removal, specifying motion parameters, tissue signals and physiological signals (heart rate, respiration).* |
| Volume censoring | *Define your software and/or method and criteria for volume censoring, and state the extent of such censoring.* |

## Statistical modeling & inference

| Model type and settings | *Specify type (mass univariate, multivariate, RSA, predictive, etc.) and describe essential details of the model at the first and second levels (e.g. fixed, random or mixed effects; drift or auto-correlation).* |
|---|---|
| Effect(s) tested | *Define precise effect in terms of the task or stimulus conditions instead of psychological concepts and indicate whether ANOVA or factorial designs were used.* |

Specify type of analysis:     ☐ Whole brain     ☐ ROI-based     ☐ Both

| Statistic type for inference (See Eklund et al. 2016) | *Specify voxel-wise or cluster-wise and report all relevant parameters for cluster-wise methods.* |
|---|---|
| Correction | *Describe the type of correction and how it is obtained for multiple comparisons (e.g. FWE, FDR, permutation or Monte Carlo).* |

## Models & analysis

| n/a | Involved in the study |
| --- | --- |
| ☐ ☐ | Functional and/or effective connectivity |
| ☐ ☐ | Graph analysis |
| ☐ ☐ | Multivariate modeling or predictive analysis |

**Functional and/or effective connectivity**

*Report the measures of dependence used and the model details (e.g. Pearson correlation, partial correlation, mutual information).*

**Graph analysis**

*Report the dependent variable and connectivity measure, specifying weighted graph or binarized graph, subject- or group-level, and the global and/or node summaries used (e.g. clustering coefficient, efficiency, etc.).*

**Multivariate modeling and predictive analysis**

*Specify independent variables, features extraction and dimension reduction, model, training and evaluation metrics.*

