## [Peer Review file · Nature]

Global phenology maps reveal how seasonal asynchrony can shape evolution

Corresponding Author: Dr Drew Terasaki Hart

Version 0:

Reviewer comments:

Referee #1

(Remarks to the Author)

The authors offer an interesting and mostly well-written manuscript in which they seek big and deep ecological patterns in a large amount of remote sensing data using lots of computations. The goal is ambitious and the findings are provocative, but the devil is in the details. Unfortunately, the ms has many critical bits of information missing and curious choices ill-explained. The authors will, hopefully, be able to provide sufficient clarifications, explanations, and justifications in a revision.

Zeroth, the current versions of all MODIS products used in the ms are V061, not V006. Pulling datasets from a service provider like GEE instead of the official data archive (e.g., LPDAAC) risks working with products that are out of date. Note, too, the 6.1 products have been available for several years.

First, it is curious to read in the title and the abstract the promise of a global perspective, only to see in the many maps lots of high northern latitude forests and tundra excluded from the analysis as well as large portions of Central Asia. It is particularly curious to see most of Kazakhstan not mapped since the first article to describe and promote the idea of “land surface phenology” focused on the changes in LSPs in Kazakhstan in the wake of the collapse of the Soviet Union: de Beurs, K.M. and Henebry, G.M., 2004. Land surface phenology, climatic variation, and institutional change: Analyzing agricultural land cover change in Kazakhstan. *Remote Sensing of Environment*, 89(4), pp.497-509, <https://doi.org/10.1016/j.rse.2003.11.006>.

Second, although the “10-year archive” is repeatedly mentioned, I could not find an explicit listing of which ten years their “archive” spanned. Given that the MODIS products used in the analysis span more than two decades, it is odd that the authors restricted their focus to just 10 years, especially given the strong influence of climate oscillation modes on LSPs.

Third, the authors place inordinate trust in the MODIS land cover product (MCD12Q1 V6) as a means to identify land cover change. [A more appropriate product for their scale of analysis is MCD12C1 V61], especially the land cover type percentage at 0.05 degrees from which you can directly get both the 1st and 2nd more common LCTs, not just the most common.] The trickiest phenologies (both organismal and land surface) occur in arid to semi-arid grasslands, savannas, and shrublands. Interannual variation in land cover types in MCD12Q1 can arise from interannual variation in weather, particularly in continental climates. Relying on an 8 out of 10 rule (l 352) will exclude vast swaths of drier biomes that are not changing land cover type but merely responding to the climatic forcings of temperature and precipitation.

Fourth, it is important to be very clear about which pixels were excluded and why, but the section starting at l 341 “Data filtering for LSP calculation” gives insufficient details to replicate their filtering. Perhaps a more detailed explanation, particularly of the use of products’ QA/QC bits could be provided in the SOM. In addition, it would be helpful to include in the SOM truly global maps of which pixels were and which were excluded and why, along with tables of areas included and excluded by continent. It was surprising to read at l 425 that “urban and built-up lands, and water bodies” had been filtered out of the remote sensing data, since no mention was made in the data filtering section. Given that the NBAR data (MCD43A4 V061: <https://lpdaac.usgs.gov/products/mcd43a4v061/>) filters out areas that lack characteristic BRDFs, such as cities and open water bodies, the exclusion makes sense. Yet, at ll 505-507, more filtering via land cover types occurs:

croplands, urban and built-up lands, or cropland/natural vegetation mosaics. It is critical for the readers to get a consistent and coherent account of the data processing pipelines at a sufficient level of detail to enable replication. And the appropriate place for that detail is in SOM, in addition to sharing the pipeline code at github to advance open science.

Fifth, harmonic regression can explain the appearances, but it also imposes strong fingerprints in the resulting products. I was surprised, however, to see that the authors did not include any phase components in their fitting (l 389).

Sixth, the authors misuse the term "validation"; they are just comparing values. Validation in the remote sensing world implies much more effort, see: https://modis-land.gsfc.nasa.gov/MODLAND_val.html; <https://doi.org/10.1016/j.rse.2019.111490>; <https://doi.org/10.1080/01431161.2012.674230>; <https://doi.org/10.1016/j.rse.2021.112686>. On a related but different note: why did not authors try using the MCD12Q2V061 product to conduct their diversity-convergence-asynchrony analyses?

Seventh, was the EOF/PCA calculated using the covariance matrix of the image stack or the correlation matrix?

Eighth, the asynchrony analyses are predicated on the LSP derivations. Given all the questions and concerns raised above, I don't have confidence in the final results or their interpretation.

Ninth, many of the graphics are difficult to read due to small fonts and/or color schemes that exclude the colorblind and/or poor color contrasts. Many of the maps in the SOM lack a legend.

(Remarks on code availability)

Since neither the main text nor the SOM offer sufficient clarity to replicate results, I felt no urge to consult the code at this stage.

Referee #2

(Remarks to the Author)

The manuscript presents a novel global NIRv-based phenology map dataset and investigates the spatial variability and asynchrony of phenology in relation to topoclimate, microtopography, and community composition. The study identifies tropical montane and Mediterranean climate regions as hotspots and highlights precipitation and minimum temperature as key drivers of spatial asynchrony. The topic is of broad interdisciplinary interest, and the statistical approaches and analysis methods employed are robust. The manuscript is well written. The investigation of spatial variability and asynchrony adds valuable insights into the understanding of phenological patterns at a global scale. However, the interpretation and implication of the results in comparison to existing knowledge are vague, and the detailed evaluation of the new dataset's spatial and temporal characteristics against previous datasets across various climate zones and vegetation types need further to be enhanced.

Characterizing spatiotemporal patterns of phenology has been one of the main focuses of the field in the past two decades, leveraging various vegetation index based satellite products, especially at mid-high latitudes. The authors have developed a novel dataset based on NIRv, presenting several advantages over traditional metrics. However, a comprehensive evaluation of the strengths and limitations of this new dataset is needed. While the dataset provides fresh insights, its interpretation and implications require further refinement and articulation. For example, the influence of both environmental controls and species composition on plant productivity/phenology has been documented. It is important to highlight the novel and unique contributions of this study insights derived from the NIRv dataset compared to existing knowledge, e.g., the complexity of phenology in tropical regions, which differs from other regions, presents an interesting area of exploration. The identification of patterns within tropical regions, addressing uncertainties from previous datasets, potentially constitutes a significant contribution. Nonetheless, these aspects need more emphasis in the manuscript.

The authors built a harmonic regression model to fit stand-level photosynthesis from NIRv satellite products, labeling the resulting time series as "phenology." However, this actually represents the seasonality of photosynthesis capacity rather than the conventional definition of phenology, which typically refers to the timing of specific phenological events such as the onset or cessation of seasons. Given this distinction, I recommend that the authors reconsider the terminology used for their dataset to distinguish it from existing phenology datasets widely utilized in the field. This clarification will help to avoid potential confusion and accurately reflect the unique nature of their dataset's focus on photosynthesis dynamics.

Why did the authors choose this harmonic regression model instead of other models? How well does this model behave compared to other models across varying climate types?

Several important methodological details are missing, including the years (start year and end year) of NIRv data used, the specifics of the harmonic regression model fitting process (When fitting the harmonic regression model, it is unclear whether the model utilized the time series data from all years (10 years) collectively, employed the long-term mean, or if it was fitted separately for each individual year), and whether and how does this harmonic regression model represent inter-annual variation (temporal variability) in NIRv (e.g., due to inter-annual variation in T, precip, ENSO etc)?

With climate warming, growing season becomes longer, and the phenological patterns are expected to change over time. whether this temporal evolution adequately captured within the harmonic regression model? How does this change affect temporal changes in the coefficient of the annual (ann) and semiannual (sem) frequencies? elucidating these aspects helps understand the model's ability to characterize phenological dynamics under climate change.

The calculation of phenology asynchrony should be supplemented with the reporting of p-values to assess the significance of trends. Moreover, while the slope represents a continuous variable, employing classification methods offers a more straightforward approach to discerning differences or similarities in phenology asynchrony. The rationale of using slope as phenology asynchrony instead of utilizing time series classification algorithms to show the differences or similarities in

phenology asynchrony requires clearer articulation.

It looks like the spatial patterns of phenology asynchrony aligns well with climate patterns. It would be useful to explain what spatial asynchrony represents in the broader picture, and what new information can we get from the changes in spatial asynchrony.

Minor:

Consider displaying actual NIRv time series data alongside fitted curves for selected sites would enhance the visualization of results.

Data availability has missing links?

The authors excluded pixels whose LSP time series had >50% missing data. This threshold may be considered lenient.

What's the confidence level in the fitted curve when pixels exhibit a 45% absence of LSP time series data but are still retained in the analysis? Stricter criteria or a more thorough justification for the chosen threshold may enhance confidence in the dataset's reliability and the accuracy of the resulting analyses.

Fig.1 color legend is hard to interpret. Not visually straightforward to see intercontinental convergence due to a multitude of colors. Classify them into several main categories and show the NIRv curve for each of them?

Referee #3

(Remarks to the Author)

This paper analyses the factors controlling the asynchrony of land surface phenology (LSP). The authors use multiple remote sensing datasets and flux towers to extract phenology and asynchrony, and then use machine learning and SHAP values to predict covariates of asynchrony. The authors map spatial asynchrony in LSP and propose hotspots in tropical montane and Mediterranean climates. The authors then use their results to support the hypothesis that 'allochory through allopatry' contributes to global biodiversity patterns.

The article is well written in the introduction, but I found the methods section difficult to follow. I have two main concerns that I suggest the authors address

I have doubts that the resolution of MODIS and the SIF product is sufficient to answer the research question, as the asynchrony hypothesis should operate more at the individual scale, whereas MODIS can only detect landscape processes. In my opinion, this is a key aspect that the authors should address and, if necessary, use ground data to support their claims with satellite observations.

The second concern relates to the presentation and structure of the article, in particular the methods section. In the methods, there's a lot of detail about technical aspects, but the whole workflow is unclear. There are variables that are not written out and clearly described (for example LSP.asyneig, a key variable that is not defined). I would suggest to include a flowchart in the supplementary material and to describe the different steps more clearly.

Minor points

"Finally, we removed any pixels whose Pielou's evenness⁸⁵ was less than 0.8; we calculated Pielou's evenness, $J' = H'/H'_{max}$, by calculating H' (i.e., Shannon's diversity index⁸⁶) using 12 values, each value being a monthly average proportion of non-missing daily data over the 10-year NIRV archive. Manual inspection of fitted phenological patterns after applying this series of filtering steps confirmed successful removal of locations otherwise producing spurious results."

It is unclear why this was done, as in principle the authors are filtering time series with high variability, but in some rainfall limited systems this variability is a property of the system. I would encourage the authors to clarify this.

In the methods, I found the Monte Carlo analysis for the confidence intervals very interesting. A rather robust analysis.

"that allow stronger estimation of photosynthesis, across global deciduous and evergreen terrestrial biomes, than do the greenness indices previously used for such purposes"

I would be more cautious about this statement. NIRv is more of a proxy for green fPAR and fluorescence is more of a proxy for APAR. This is why these datasets relate better to photosynthesis than others, but they are not a direct estimate of photosynthesis. NIRv is also a greenness index (based on the same bands as NDVI) that better accounts for soil effects on greenness. So please clarify this part of the text.

"To estimate the seasonality of stand-level photosynthesis"

Fitting a regression to NIRv and SIF is not estimating seasonality at stand level. We are talking about satellite data, which are not necessarily representative of the stand, in addition to the above comment on photosynthesis.

"Validation" against FLUXNET data. First of all, I would suggest to apply the quality check also to GPP data, e.g. by filtering the daily GPP data with $fqc > 0.70$, as done in several articles. This will ensure that the daily GPP comes from measured NEE data and not from gap filled data. Secondly, I think this validation can be quite misleading as it is not given that the flux footprint tower is representative of the pixel information. I think it is not really necessary and does not provide any further insight.

Version 1:

Reviewer comments:

Referee #1

(Remarks to the Author)

The authors have done an admirable job of addressing my concerns. I have just a few more minor suggestions to improve the precision and clarity of the ms.

First, it is appropriate to touch upon the effects of land cover/land use change on land surface phenologies in the introduction rather than waiting until later in the ms. Climate change is just one forcing on LSP. Consider this relevant study: Zhang, X., Liu, L., & Henebry, G. M. (2019). Impacts of land cover and land use change on long-term trend of land surface phenology: a case study in agricultural ecosystems. *Environmental Research Letters*, 14(4), 044020.

Second, solar-induced fluorescence rather than sun-induced.

Third, you cite the MOD09/MYD09 products as the source for the NIRv data at line 83 but at line 495 indicate that you used the NBAR product MCD43A4.061 to calculate the NIRv.

Fourth, at lines 266-267 you state "In this capacity, remote sensing of phenology could play an underappreciated role in improving our understanding of spatiotemporal evolutionary dynamics." I think it would clarify the sentence to insert a delimiting phrase—perhaps either "land surface" or "proxies for organismal"—before phenology.

Fifth, at line 338, consider saying "terrestrial phenologies" rather than the singular as you are considering specifics associated with particular places.

Sixth, you need a citation to GMTED2010. Look here: <https://pubs.usgs.gov/publication/ofr20111073>

Finally, it is appropriate to acknowledge explicitly your gratitude to the anonymous reviewers who have donated their attention and expertise to improve the presentation of your research.

Referee #2

(Remarks to the Author)

The authors did a good job and have addressed most comments, but further improvement and clarification are needed in several areas:

First, this study primarily investigates spatial variations in long-term average seasonality from a spatial perspective, examining the impact of ecological and evolutionary influences across geographic gradients. This approach differs from most phenology studies, which typically assess interannual variation or long-term phenological trends in response to climate change. Because of this, the authors should:

- Explicitly clarify this unique focus by defining "LSP" (long-term mean annual phenology) early in the introduction. Currently, the definition is vague, and readers may struggle to understand the specific variables or indices employed in this study.
- Address the disconnect between the first and second paragraphs in the introduction. The first paragraph (lines 41-47) discusses phenology in the context of ecosystem science, carbon-water cycles, and earth system modeling. This is traditional phenology concept, focusing on the temporal, interannual dimension of phenology, which is not really the focus of this study. This paragraph does not establish a foundation for the subsequent analysis. In contrast, the second paragraph, which addresses landscape ecology and evolutionary biogeography, is more aligned with the study's spatial emphasis. I recommend revising the introduction to better align with the specific content and objectives of this study.
- It would be beneficial to discuss how climate change is altering landscape patterns of allochry by allopatry and the associated uncertainties that this introduces to the current results. While this is not the direct focus of the study, it presents a promising direction for future research. For instance, the harmonic model used in this study does not account for extreme events like summer droughts or spring frosts that affect NIRv time series, yet such events are occurring with increasing frequency. Discussing these limitations could enhance the contextual relevance of the study's findings under ongoing climate variability and change.

Second, I still find Figure 1 challenging to interpret and understand, and this concern has been echoed by another reviewer. At present, the key message of Figure 1 is unclear. Each figure should stand alone as a clear, self-explanatory element within the manuscript. In contrast, Figure 2 conveys useful information, but it primarily serves as a zoomed-in view of Figure 1 rather than an independent figure. The spatial map in Figure 2 that shows the location of each sub-figure is redundant with Figure 1. I recommend combining Figures 1 and 2 to present a global map with selected zoom-in views to improve clarity and streamline the presentation.

Further, Figure 2 could benefit from an improved layout for the subfigures. Additionally, the statement in the caption that "Regional heterogeneity of land surface phenology (LSP) reflects the influence of topoclimate, ecohydrology, and vegetation structure" is not directly supported by the figure itself, as it does not depict any specific patterns related to topoclimate, ecohydrology, or vegetation structure. Need to clarify if this conclusion is derived through visual interpretation of spatial patterns, a literature review, or an analysis of datasets on topoclimate, ecohydrology, and vegetation structure. If these factors were not directly analyzed, this interpretation should be moved from the figure caption to the results or discussion sections to avoid suggesting that it is an explicit result of the figure itself.

Third, the authors used USANPN SOS and spring indices in the analysis to demonstrate the performance of LSP fitting (line 165). However, USA-NPN spring indices are not direct observations; rather, they are simplified model estimates of phenological events based solely on growing degree days. In other words, these indices primarily reflect the

temperature/climate regime rather than the biological seasonality of the region. This raises questions about the appropriateness of including these indices in the analysis. Additionally, the observed lack of alignment between the USA-NPN and LSP fitting results (e.g., Figure S15) is unsurprising, with an R^2 value of only 0.13. This does not support a claim of "significant correlation to NPN first-leaf dates" (line 169) and falls short of explaining the SOS to the same extent as the spring indices ($R^2 = 0.36$). Even if a similar ability to explain SOS were observed, the implications of such an alignment would remain unclear. The rationale for this analysis appears insufficiently justified, and the results may detract from the overall narrative of the study. Datasets like these derived from citizen science, which may lack rigorous quality control, raise concerns regarding data robustness. For this purpose, the PhenoCam Network, which offers more direct and standardized phenological observations for each plant functional type, would likely be a more suitable alternative.

Fourth, what is the added insight from using NIRv and SIF-based phenology approaches compared to traditional vegetation indices (Line 55)? While the authors cite numerous studies to show consistency with previous findings, it is unclear what novel contributions this framework brings beyond existing understanding. The novelty of this approach needs to be clearly articulated, highlighting what new perspectives or advantages are gained by using NIRv and SIF that are not achievable through traditional vegetation indices.

Fifth, Fig. S1 workflow: the authors currently calculate aggregated mean values before applying the MODIS land cover filter. My understanding is that it would be more accurate to filter for land cover first and then proceed with aggregation. In the current order, pixels from non-targeted land cover types, such as urban or water, could skew the aggregated values. For instance, if such pixels represent a significant proportion (e.g., 40%) while forest 60%, they could substantially alter the final aggregated value.

Another question related to this is about land cover change. e.g., in Figure S2, an abrupt change is visible for the Blodgett Forest site in 2015. Could this shift be due to land cover change? It is essential to clarify how pixels with changing land cover types are treated. While the authors mention handling transitions to and from agricultural land, it is unclear how other types of land cover change are managed in the workflow.

Minor:

Line 150 In the Amazon, light and water availability exhibit a strong inverse relationship: during the wet season, cloud cover reduces light availability, though water is abundant due to frequent rainfall; conversely, in the dry season, sunlight is plentiful, but water availability is limited. Both forests and non-forest ecosystems in the Amazon are regulated by a combination of light and water. It is difficult to isolate and represent their individual effects independently.

Fig. S5 why does Australia show such low R^2 ?

Fig. S16, the white-to-black color legend is challenging to interpret. Consider switching to a contrasting color scheme, such as red-to-blue, to improve visual clarity and make patterns more discernible.

Referee #3

(Remarks to the Author)

The authors have created a global map of land surface phenology (LSP) using MODIS imagery that reveals LSP patterns. The map shows that similar climates have similar patterns, but local factors such as topography and vegetation can lead to differences in LSP diversity, and machine learning can be used to explain these patterns. This research has important implications for understanding how geographic separation can lead to differences in ecological and evolutionary processes. The paper has been significantly improved in terms of clarity and reproducibility compared to the first submission. The authors have also updated the datasets, for example, using the most recent MODIS collection, extending the time period, and improving the gap-filling of NIRv to extend the spatial scope of the analysis. The authors have done a tremendous amount of work to address my concerns and introduce a new analysis using ground data, which is very valuable. I still have one important point. I am still not convinced that the asynchrony hypothesis can be analyzed with MODIS, especially with the currently selected drivers. I really appreciate the effort of the authors to include the species level analysis, that is great. The new Figure 5 and the section "Remote sensing predicts species-level phenological asynchrony and genetic divergence" are very informative. However, at the MODIS scale, there are aspects related to landscape management, disturbance, deforestation, land use that can be quite important in determining the LSP diversity signal. The authors use spatial structural entropy as a predictor, but I would suggest going deeper into the potential drivers of LSP by analysing for example disturbance patterns within the region (using for example the Global Forest Watch dataset), MODIS fire or burned area datasets, land cover change and other structural metrics.

Minor points

Lines 44-50. In general I agree with the statement, but it is unclear how LSP diversity can actually improve the capability of Earth system models (running at grid cell or column level assuming one or a mixture of PFTs). In my opinion, this argument should be better explained. Once we understand LSP, how can this be used to improve models? Should it only be used to benchmark modelled LSP or can we gain insights into landscape diversity and how this interacts with climate? At the moment it's quite unclear what the benefit of understanding LSP is for models. In fact, I would remove this point as for me the analysis is more interesting in terms of landscape ecology and understanding ecological processes rather than supporting modelling.

Line 54 "Multivariate analysis of new physiologically based proxies" Proxies or what exactly? SIF and NIRv are proxies for two very different things.

Methods:

Line 510: Why are longer SIF datasets not used?

Figure S2: I think it would be important to show example time series for some specific areas, e.g. few pixels in the tropics, few in the Mediterranean (although I think Tonzi Ranch can be classified as Mediterranean), some boreal areas, so areas where harmonic regression may have problems due to data gaps.

Version 2:

Reviewer comments:

Referee #2

(Remarks to the Author)

The authors have done a substantial amount of work to address the previous reviewer's comments, and the manuscript has been significantly improved. I only have one major concern in this round.

In the revised manuscript, the authors state that they used NDVI derived from ground-based phenology cameras in the PhenoCam network and added a new Fig. S18. However, this appears to be a misunderstanding. The PhenoCam network does not provide NDVI; rather, it provides green chromatic coordinate (Gcc), which is derived from RGB imagery. Gcc is calculated as the ratio of the green digital number to the sum of the red, green, and blue digital numbers ($G_{cc} = G / [R + G + B]$). While Gcc is widely used as a proxy for canopy greenness and phenological transitions, it is fundamentally different from NDVI, which requires near-infrared (NIR) data that is not available from standard PhenoCam cameras.

It is surprising that the authors incorporated and analyzed this dataset—adding new figures in the process—yet confused Gcc with NDVI. This suggests a lack of clarity regarding the variables used. This point should be corrected in the manuscript.

In addition, I could not find a clear data description section in the Methods that introduces and details the datasets used in the study. I recommend including a dedicated subsection that explicitly describes the source, nature, and processing of all datasets, with citations, to improve transparency and reproducibility.

Response to reviewers: Terasaki Hart et al.: Global phenology maps reveal the diversity, convergence, and asynchrony of ecosystem function

Below, we provide our detailed responses (in green) to each of the reviewer comments on our original submission (in black).

Referee #1:

(Remarks to the Author):

The authors offer an interesting and mostly well-written manuscript in which they seek big and deep ecological patterns in a large amount of remote sensing data using lots of computations. The goal is ambitious and the findings are provocative, but the devil is in the details.

Unfortunately, the ms has many critical bits of information missing and curious choices ill-explained. The authors will, hopefully, be able to provide sufficient clarifications, explanations, and justifications in a revision.

Thank you for your enthusiasm and interest. In our revised manuscript, we have tried to carefully include all of the methodological details necessary to fully recreate our work, and we have a publicly-accessible repository with all of the code needed to perform all of our analyses. We have also sought to better articulate the rationale for the decisions we made throughout the manuscript (details below).

Zeroth, the current versions of all MODIS products used in the ms are V061, not V006. Pulling datasets from a service provider like GEE instead of the official data archive (e.g., LPDAAC) risks working with products that are out of date. Note, too, the 6.1 products have been available for several years.

This is a phenomenal point. Thank you. We began this work in 2018, some years before the release of V061. After completing the GEE-based portion of our pipeline we then worked on the remaining stages of the analysis, but we neglected to return to the GEE code and edit it to update our results to use MODIS V061. We are very grateful that you pointed out this issue. We have updated all of our analyses to use the V061 dataset now. This did not qualitatively change our results, but we agree that using the most recent data available is important, and we believe it makes our analysis and findings more robust.

First, it is curious to read in the title and the abstract the promise of a global perspective, only to see in the many maps lots of high northern latitude forests and tundra excluded from the analysis as well as large portions of Central Asia. It is particularly curious to see most of Kazakhstan not mapped since the first article to describe and promote the idea of “land surface phenology” focused on the changes in LSPs in Kazakhstan in the wake of the collapse of the Soviet Union: de Beurs, K.M. and Henebry, G.M., 2004. Land surface phenology, climatic variation, and institutional change: Analyzing agricultural land cover change in Kazakhstan.

Remote Sensing of Environment, 89(4), pp.497-509, <https://doi.org/10.1016/j.rse.2003.11.006>.

Thank you for pointing this out. Dropout of these high latitude regions resulted from a combination of invalid values in the NIR_v dataset and data filtering based on minimum monthly data availability and monthly data evenness. Following Badgley et al. 2018, we excluded all NIR_v values ≤ 0 , an artefact that those authors attribute to inadequate MODIS filtering. This caused long gaps of missing data during high-latitude winters because pixels with extensive and persistent snow cover returned prolonged series of negative values. Those gaps, in turn, caused the pixels to drop out of our analysis completely because of the minimum requirements we imposed regarding monthly data availability and data availability evenness, requirements designed to prevent interpolation of unrealistic harmonic regression artefacts into the gaps. By backfilling these errant data points with the minimum positive value observed across a pixel's time series, we now retain them in our analysis, allowing us to increase the spatial coverage of our analysis within high latitude regions.

Very high latitudes still drop out of our analysis, driven not by long spans of negative values but simply by long, seasonally repeating spans of unavailable data, because of lack of daylight during winter MODIS overpass times (an issue that also presents complications for the derivation of other global remote sensing products, e.g., MOD44B.061 Vegetation Continuous Fields; Townshend et al. 2022). Since these regions are not central to the subsequent analyses in the manuscript, this omission does not impact the findings we report, and outside of these very high latitude areas your suggestions have enabled us to substantially increase the spatial scope of our analyses. We hope that any previous lack of clarity about these data masking procedures is resolved by our edits (L505-508) and improved explanations (L574-580) in the methods section, the newly included workflow diagram (Fig. S1), and the mask maps that you requested (Fig. S4).

Second, although the “10-year archive” is repeatedly mentioned, I could not find an explicit listing of which ten years their “archive” spanned. Given that the MODIS products used in the analysis span more than two decades, it is odd that the authors restricted their focus to just 10 years, especially given the strong influence of climate oscillation modes on LSPs.

We thank you (and reviewer 2) for pointing out this omission. We have now made the span of years clear, in both the main text (L82-84) and the methods (L493-495). More importantly, we fully agree with your concerns about the potential for longer-term climate oscillations to bias our characteristic annual LSP patterns, so we have now extensively optimized our code, which allowed us to run our analysis for a 20-year period (2001 through 2020), thinning the data to every four days, and which also enabled the other improvements we have made (such as the backfilling of invalid NIR_v values that we explained in our response to the prior comment).

Third, the authors place inordinate trust in the MODIS land cover product (MCD12Q1 V6) as a means to identify land cover change. [A more appropriate product for their scale of analysis is MCD12C1 V61), especially the land cover type percentage at 0.05 degrees from which you can directly get both the 1st and 2nd more common LCTs, not just the most common.] The trickiest

phenologies (both organismal and land surface) occur in arid to semi-arid grasslands, savannas, and shrublands. Interannual variation in land cover types in MCD12Q1 can arise from interannual variation in weather, particularly in continental climates. Relying on an 8 out of 10 rule (l 352) will exclude vast swaths of drier biomes that are not changing land cover type but merely responding to the climatic forcings of temperature and precipitation.

This was an extremely helpful critique, as it forced us to circle back and carefully review our method for dealing with inappropriate land cover types. We have now updated our masking procedure to use MODIS V061 instead of V006. We have also switched to using the more appropriate, pre-aggregated MCD12C1 product that you suggested. For the sake of simplicity, we only include pixels with valid majority land cover types (i.e., not barren, frozen, urban, or permanent water).

With regard to temporal consistency in land cover, we are glad that you pointed out how we had inadvertently omitted drier regions because of interannual climatic variability. Exploratory analysis revealed not only that this was indeed occurring but also that our approach was causing dropout in regions that simply have higher classification uncertainties, regardless of climate forcing, because of intermediate vegetation structure (e.g., miombo and other woodland areas). We have altered our filtering so as to only drop pixels that register a change between agricultural and non-agricultural land cover types (i.e., to or from classes 12 or 14 from classes outside those two), allowing us to more accurately omit the land cover change events that were our original cause of concern without causing much collateral data loss. We have updated the methods to reflect these changes (L539-552).

Fourth, it is important to be very clear about which pixels were excluded and why, but the section starting at l 341 “Data filtering for LSP calculation” gives insufficient details to replicate their filtering. Perhaps a more detailed explanation, particularly of the use of products’ QA/QC bits could be provided in the SOM. In addition, it would be helpful to include in the SOM truly global maps of which pixels were and which were excluded and why, along with tables of areas included and excluded by continent. It was surprising to read at l 425 that “urban and built-up lands, and water bodies” had been filtered out of the remote sensing data, since no mention was made in the data filtering section. Given that the NBAR data (MCD43A4 V061: <https://lpdaac.usgs.gov/products/mcd43a4v061/>) filters out areas that lack characteristic BRDFs, such as cities and open water bodies, the exclusion makes sense. Yet, at ll 505-507, more filtering via land cover types occurs: croplands, urban and built-up lands, or cropland/natural vegetation mosaics. It is critical for the readers to get a consistent and coherent account of the data processing pipelines at a sufficient level of detail to enable replication. And the appropriate place for that detail is in SOM, in addition to sharing the pipeline code at github to advance open science.

This, too, was critically helpful feedback. We have now revisited and clarified these aspects of our methods, not only in our code and in the text of the SOM but also in the mapping workflow diagram (Fig. S1), the global masking maps (Fig. S4), and the table of continentally stratified inclusion/exclusion areas that you requested (Table S1).

Fifth, harmonic regression can explain the appearances, but it also imposes strong fingerprints in the resulting products. I was surprised, however, to see that the authors did not include any phase components in their fitting (I 389).

We agree that a harmonic regression could impose a strong fingerprint on the output. This was the reason that we implemented masking procedures to ensure that we're only using pixels with a minimum of 10% average data availability in all months and a minimum of 0.8 Pielou's evenness of data availability across all months. We found that having at least that much data coverage provided regular anchor points that constrained the fitted phenocycles to reasonable patterns. The final step, masking out pixels that do not pass a strict permutation test, serves as an additional filter to catch any remaining pixels with questionable phenocycles, and the fact that this step drops so few additional pixels (see Fig. S4) gives us confidence that our overall masking procedure is solidly achieving its aims.

Additionally, we agree that phase is an important component to model in the NIR_v and SIF time series. Our models do in fact capture it: Because of the algebraic manipulation used to represent the two-frequency (i.e., annual and semi-annual) discrete Fourier transform as a trigonometric linear combination, and thus as a harmonic regression (based on Shumway and Stoffer 2017 Eqns. 4.1 and 4.2, which we now cite), the annual and semiannual phase values could be recovered from our coefficients as $\phi_{ann} = \tan^{-1}(-\beta_1/\beta_2)$ and $\phi_{sem} = \tan^{-1}(-\beta_3/\beta_4)$. A corollary approach has been used in other peer-reviewed literature (e.g., Wilson et al. 2018). However, we do not make use of the fitted phase values in our analyses, because we needed to design information-rich methods that could be used to visually and analytically assess phenological inter-site differences, across the global range of observed phenocycles, and irrespective of variation in amplitude, phase, and degree of bimodality. To prevent similar concern amongst readers we have now revised our explanation of the LSP-fitting method and our interpretation of its results, including citing the source of inspiration for its mathematical formulation (L593-629).

Sixth, the authors misuse the term "validation"; they are just comparing values. Validation in the remote sensing world implies much more effort, see:

https://modis-land.gsfc.nasa.gov/MODLAND_val.html;

<https://doi.org/10.1016/j.rse.2019.111490>; <https://doi.org/10.1080/01431161.2012.674230>;

<https://doi.org/10.1016/j.rse.2021.112686>. On a related but different note: why did not authors try using the MCD12Q2V061 product to conduct their diversity-convergence-asynchrony analyses?

You are entirely right to point out that our "validation" analyses do not satisfy the proper definition of "validation" within the remote sensing domain. We should not have used this term so loosely. We have adjusted this language wherever we had misused it in the manuscript, instead referring to these analyses as evaluations or assessments.

Your suggestion to try using the MCD12Q2 product is a very reasonable one, and one that we did indeed consider earlier on. However, we determined that the MCD12Q2 product, while

useful in many places and for many purposes, did not provide adequate coverage and sensitivity to enable the global, cross-biome analyses we envisioned (as has also been noted by other work aimed at improving LSP-mapping in biomes with less marked growing seasons, e.g., Xie et al. 2023). Because it's based on EVI MCD12Q2 is less likely to be sensitive to seasonal dynamics in tropical and evergreen environments (Ganguly et al. 2010), and because it relies on the extraction of scalar phenometrics that assume discrete growing seasons it entirely omits many of the tropical regions we were interested in. Thus, we opted to construct a purpose-built analytical workflow that enables calculation of pairwise comparisons between the scaled phenocycles at any two locations, allowing us to focus on spatial variation in average phenological timing.

Seventh, was the EOF/PCA calculated using the covariance matrix of the image stack or the correlation matrix?

We are grateful that you brought this to our attention. We neglected to include enough detail on this analysis in our methods section. The software we used (Python's `eofs` package; Dawson 2016) uses singular value decomposition of a spatiotemporal anomaly matrix, a method that the package authors demonstrate is mathematically equivalent to calculating EOFs using the covariance matrix. We standardized each time series in our dataset and the `eofs` package then centered them (to derive an anomaly matrix). This allows our analysis to focus on geographic variation in the patterns and timing of seasonal LSP cycles, irrespective of differences in their fitted amplitudes. We now clarify all of this detail, in both the methods section and the workflow diagram, and we also now cite the package's peer-reviewed publication (L699-707).

We also note that, in reviewing this step of our analyses, we realized that we had accidentally been normalizing our time series before EOF calculation, rather than standardizing. The potential problem with normalization is that pixels' time series could still have unequal variances, which could cause the EOF results to reflect, in part, some of the geographic variability in the variance – a nuisance result, in our case. We have now corrected this, and as a result have seen minor shifts in the EOF results and their RGB visualization that have no noticeable effect on our overarching findings and their interpretation.

Eighth, the asynchrony analyses are predicated on the LSP derivations. Given all the questions and concerns raised above, I don't have confidence in the final results or their interpretation.

Given the list of critiques raised above, we entirely understand your lack of confidence in the asynchrony analysis and its interpretation. We believe we have been able to address all of your critiques, and we are grateful for all of your detailed input, as we feel this process has considerably strengthened our overall workflow. We are eager to hear your thoughts regarding the remainder of our manuscript.

Ninth, many of the graphics are difficult to read due to small fonts and/or color schemes that exclude the colorblind and/or poor color contrasts. Many of the maps in the SOM lack a legend.

On the basis of your critiques, we have increased font sizes and added a map legend that was missing. We are of course happy to make any further specific changes, at the reviewers' or editors' request.

We also note that we had actually already carefully chosen colorblind-friendly color palettes wherever possible, using a protanomaly/protanopia simulator to confirm that we had made reasonable choices. Nevertheless, we remain open to any specific suggestions on how to improve the colorblind friendliness of specific figures, as this is something we are always keen to do better. We also fully acknowledge and regret that the RGB maps are inherently not colorblind-friendly. We could not conceive of any other clear way to combine the information from the top EOFs into a single summary map while retaining colorblind-friendliness, and we found this method otherwise acceptable, in light of the routine application of false-color visualization in remote sensing research. We hope that including the separate maps of each of the EOFs, plotted using a colorblind-friendly divergent palette (Fig. S6), still ensures access to these results for colorblind readers.

Referee #1 (Remarks on code availability)

Since neither the main text nor the SOM offer sufficient clarity to replicate results, I felt no urge to consult the code at this stage.

This is entirely reasonable. The code remains publicly accessible and we would be keen to receive any feedback, should you decide to consult it at a later date.

Referee #2:

(Remarks to the Author)

The manuscript presents a novel global NIRv-based phenology map dataset and investigates the spatial variability and asynchrony of phenology in relation to topoclimate, microtopography, and community composition. The study identifies tropical montane and Mediterranean climate regions as hotspots and highlights precipitation and minimum temperature as key drivers of spatial asynchrony. The topic is of broad interdisciplinary interest, and the statistical approaches and analysis methods employed are robust. The manuscript is well written. The investigation of spatial variability and asynchrony adds valuable insights into the understanding of phenological patterns at a global scale. However, the interpretation and implication of the results in comparison to existing knowledge are vague, and the detailed evaluation of the new dataset's spatial and temporal characteristics against previous datasets across various climate zones and vegetation types need further to be enhanced.

We thank you for your interest in our study. We also agree that we should have added more and clearer contextual interpretation of our datasets and results. We have made an extensive effort

to revise the manuscript to provide more comprehensive interpretation of our results in light of prior work (e.g., L147-154; L226-231; L231-239) and to better articulate its many and broad implications (e.g., the 'Conclusion' section). We are eager to hear your assessment of this updated version.

Characterizing spatiotemporal patterns of phenology has been one of the main focuses of the field in the past two decades, leveraging various vegetation index based satellite products, especially at mid-high latitudes. The authors have developed a novel dataset based on NIRv, presenting several advantages over traditional metrics. However, a comprehensive evaluation of the strengths and limitations of this new dataset is needed. While the dataset provides fresh insights, its interpretation and implications require further refinement and articulation. For example, the influence of both environmental controls and species composition on plant productivity/phenology has been documented. It is important to highlight the novel and unique contributions of this study insights derived from the NIRv dataset compared to existing knowledge, e.g., the complexity of phenology in tropical regions, which differs from other regions, presents an interesting area of exploration. The identification of patterns within tropical regions, addressing uncertainties from previous datasets, potentially constitutes a significant contribution. Nonetheless, these aspects need more emphasis in the manuscript.

We have now attempted to better articulate the strengths, limitations, and unique contributions of our work. We have substantially revised the discussion of examples of complex, regional LSP patterns, with the goal of clarifying the broader insights that these exemplify. We have also significantly expanded the analyses and improved the conclusions, focusing on: 1) how our approach enables better comparative understanding of between-site differences in phenological pattern and timing, across biomes and irrespective of amplitude; 2) how our methodological innovations and findings shed light on the complex phenological diversity we observe in tropical ecosystems; and 3) how future work improving and extending our concepts and methods can address the limitations of this initial analysis and generate crucial insight into big questions across a range of fields.

The authors built a harmonic regression model to fit stand-level photosynthesis from NIRv satellite products, labeling the resulting time series as "phenology." However, this actually represents the seasonality of photosynthesis capacity rather than the conventional definition of phenology, which typically refers to the timing of specific phenological events such as the onset or cessation of seasons. Given this distinction, I recommend that the authors reconsider the terminology used for their dataset to distinguish it from existing phenology datasets widely utilized in the field. This clarification will help to avoid potential confusion and accurately reflect the unique nature of their dataset's focus on photosynthesis dynamics.

This is a thoughtful and thought-provoking critique that prompted a lot of conceptual reflection. We agree that the term 'phenology' often refers to the timing of specific biological events, but this is not always the case, especially in the context of the term 'land surface phenology'. While land surface phenology does often study fixed events (e.g., leaf-out), it has also often been defined as simply dealing with temporal variation more generally – e.g., "the seasonal pattern of

variation in vegetated land surfaces observed from remote sensing” (White and Nemani, 2006). The earliest use of the term to which we could find reference explicitly clarifies that land surface phenology “is not a traditional phenology associated with specific events in a plant’s life history; rather, land surface phenology describes the seasonality of reflectance characteristics that are associated with stages of vegetation development.” (Henebry and Su 1995, p 143; quoted in Henebry and de Beurs, 2013).

In this light, we see our work as fitting well within the broader corpus of self-described ‘land surface phenology’ research. We did nonetheless consider potential alternative terms, but they always struck us as less accurate – e.g., ‘land surface seasonality’ is a term that appears in the literature but often with an implicit (or explicit; Alemu and Henebry, 2013) focus on climatic/abiotic phenomena, and ‘vegetation dynamics’ is used in some contexts but seems too general and vague to refer to the specific remote sensing signal that we are modeling. Thus, we have opted to retain the term ‘land surface phenology’. We feel that labeling our work this way may help to highlight the advantages that derive from a continuous-time methodological approach that can overcome what we see as some of the limitations of this field’s emphasis on the estimation of scalar phenometrics. Indeed, there is some precedent for the insights that can be derived from such an approach (e.g., Boyce et al. 2017). Nevertheless, we remain open to considering specific terminological recommendations.

Why did the authors choose this harmonic regression model instead of other models? How well does this model behave compared to other models across varying climate types?

This is a fair question, and we have edited the methods section to better address it. In short, we chose harmonic regression out of preference for modeling minimalism: this approach is simple, justifiable, broadly understood, and easily interpreted. It allows us to naively fit the same model at all locations, with minimal parameterization, and then develop confidence in the results using downstream assessment (e.g., our permutation-based significance testing and our flux-tower and now NPN evaluations) and exploration (e.g., our regional interpretation of the results, using regional RGB visualizations and time series clustering as well as references to regionally focused literature and data products). Hence, we determined that it was optimal to choose this methodology based on its characteristics, interpretability, and fit for our goals, rather than using formal model comparisons. We have explained this rationale in the methods section of our revised manuscript (L608-624) and have described the method’s strong overall performance within the results (L155-180).

Several important methodological details are missing, including the years (start year and end year) of NIRv data used...

We apologize for this omission. We have now clarified, in the main text and the methods, that our analysis covers the period from 2001 through 2020 – newly increased to 20 years in response to points raised by both you and reviewer 1.

... the specifics of the harmonic regression model fitting process (When fitting the harmonic regression model, it is unclear whether the model utilized the time series data from all years (10 years) collectively, employed the long-term mean, or if it was fitted separately for each individual year)...

Thank you for pointing this out; we had not realized this was unclear. We have now clarified, in both the main text (L82-84) and methods (L493-495), that each pixel's model was fitted to a single time series across the entire 20-year study period.

... and whether and how does this harmonic regression model represent inter-annual variation (temporal variability) in NIR_v (e.g., due to inter-annual variation in T, precip, ENSO etc)?

This is a great question, and one that we should have addressed more directly. We do not include any model components that explicitly represent inter-annual variability. Rather, we expected locations with higher interannual variability to be reflected by lower R² values, something we do indeed see in places known to have high interannual variability of productivity (e.g., the drier, inland regions of Australia) within our harmonic regression R²s (Fig. S5), NIR_v - SIF comparison (Fig. S12) and flux tower evaluation (Fig. S16). We could imagine methods that would assess annual divergences from our long-term average phenocycles and then compare that divergence to the indices of climate oscillation modes (e.g., ENSO). Something like this would be of broad interest, and could be particularly fascinating to consider in the context of the evolutionary implications of allochryny by allopatry, as we now mention in the conclusion (L355-359).

With climate warming, growing season becomes longer, and the phenological patterns are expected to change over time. whether this temporal evolution adequately captured within the harmonic regression model? How does this change affect temporal changes in the coefficient of the annual (ann) and semiannual (sem) frequencies? elucidating these aspects helps understand the model's ability to characterize phenological dynamics under climate change.

This is indeed true. Our model detrends the NIR_v time series with respect to the mean (by way of dropping the β_t coefficient after model fitting) but does not allow estimation of a trend in the shape of the average annual LSP curve (i.e., the other coefficients do not vary as a function of time). Indeed, the analysis we constructed herein, focused on the ecological and evolutionary influence of geographic variation in long-term average phenological patterns, purposefully averages over any such trends. That said, we could certainly imagine extensions of our work that would enable this sort of analysis, providing insight into the ways in which climate change is changing landscape patterns of allochryny by allopatry. Indeed, this would be a useful alternative to the frequent focus of the LSP literature on the analysis of temporal trends in univariate phenometrics (e.g., start and end of season). We now mention, in the conclusion, that our research provides insights that could be useful for future work to understand Anthropocene phenological shifts and their consequences (L337-342).

The calculation of phenology asynchrony should be supplemented with the reporting of p-values to assess the significance of trends.

This is an excellent point, and was an oversight in our previous draft. We had not felt compelled to report on the P-values because they were all highly significant. Upon reviewing and rerunning our workflow we realized that this was a result of forcing the asynchrony regressions through the origin, a conceptually reasonable choice but not an empirically reasonable one. We have reworked the asynchrony-calculation workflow to allow intercepts to be estimated and to fix asynchrony values at zero wherever the regression slope's P-value is > 0.01 . This results in asynchrony maps where all non-zero values are supported by significant P-values, and that now exhibit the noisiness that we had initially expected from such a neighborhood metric. We have updated the methods to clarify this (L774-775), but the main text needed no update because our overarching findings remained unchanged.

Moreover, while the slope represents a continuous variable, employing classification methods offers a more straightforward approach to discerning differences or similarities in phenology asynchrony. The rationale of using slope as phenology asynchrony instead of utilizing time series classification algorithms to show the differences or similarities in phenology asynchrony requires clearer articulation.

This is an interesting thought. We agree that this would be a relatively straightforward approach for calculating asynchrony if our interest were simply to characterize the LSP variability within a location's (i.e., a pixel's) neighborhood. However, the spatiotemporal dynamics proposed by the ASH do not imply only that seasonality, and thus phenology, should be more variable in some regions than in others, but instead explicitly state that seasonality/phenology should turn over more quickly as a function of geographic distance. This is the reason that we devised an approach that explicitly models this spatial rate of phenological change, as we have now clarified in the methods section (L776-778). Clustering methods could still be useful for this, but only if such methods ultimately relied on some form of regression as a means of characterizing the spatial rate of change. We do, however, use clustering in a number of places, to make our high-dimensional LSP results more tractable for visualization and interpretation.

It looks like the spatial patterns of phenology asynchrony aligns well with climate patterns. It would be useful to explain what spatial asynchrony represents in the broader picture, and what new information can we get from the changes in spatial asynchrony.

This insightful comment helped us to see the ways in which we were not clearly highlighting some of the valuable findings that arise from our work. It is true that many of the patterns in the spatial asynchrony map are driven by climate (e.g., the bands of asynchrony separating winter-monsoon and summer-monsoon climate regions in North America's desert southwest), which we mention in the paper. However, we also discuss numerous examples of LSP variation that does not appear to be driven by differences in climate – for example, the sharp differences in LSP patterns that occur between different vegetation structures and communities within Mediterranean climate regions, the Everglades, and the Amazon. This LSP variability causes

signals of high spatial asynchrony that are not correlated with spatial climatic differences, but that may in fact reflect the different ecohydrological dynamics experienced by different microtopographic positions subjected to the same climate, or by differential ecophysiological responses of different vegetation types to the same climate. We have edited the main text to better articulate these patterns and their likely drivers (L126-154).

Minor:

Consider displaying actual NIRv time series data alongside fitted curves for selected sites would enhance the visualization of results.

This was a great suggestion. However, this actually required a separate figure, given that the line plots in the figures in the previous draft only depicted a single, 'average' year, whereas the curve was initially fitted to 10 (now 20) years of data. We have now created that figure (Fig. S2), and we have also provided a figure that shows a snapshot of the GEE App that we created as a data viewer (Fig. S3), where users can click on any two sites and generate a comparative visualization of their fitted LSP patterns. (That data viewer's link, <https://lyrical-ring-231401.projects.earthengine.app/view/globalphenologicaldiversityandasynchronyterasakihart2024>, is provided within the project's GitHub repo rather than within the manuscript, to prevent problems arising from future changes to GEE that could affect our code or the current account where we are hosting the app, and thus could break a static link.)

Data availability has missing links?

We purposely left these links missing in case we needed to adjust and/or rerun our analysis during review. We intend to archive our final dataset with a minted DOI, then update these links within the manuscript before publication. In the meantime, we have created the GEE app data viewer that we mentioned in our response to the prior comment, but we would also be happy to provide you with direct access to any of the datasets, should that be helpful.

The authors excluded pixels whose LSP time series had >50% missing data. This threshold may be considered lenient. What's the confidence level in the fitted curve when pixels exhibit a 45% absence of LSP time series data but are still retained in the analysis? Stricter criteria or a more thorough justification for the chosen threshold may enhance confidence in the dataset's reliability and the accuracy of the resulting analyses.

This is a great point. We had previously observed that this filter had little influence on the footprint of the final masking map, given that pixels with substantial missing data across the full time series were also typically masked out by the other filters designed to avoid data that would cause unreasonable modeling artefacts – namely, the minimum monthly data availability filter, a minimum data availability evenness filter, and a permutation-based significance test filter. Our new masking-map figure (Fig. S4) shows that this is indeed the case, because this filter only affects a substantial number of pixels within two perennially cloudy regions, the tropical Andes and western Gabon, where the monthly data availability and data availability evenness masks

ensure that any retained pixels have reasonable data availability throughout the calendar year, and where significance testing insures against spurious fits.

Fig.1 color legend is hard to interpret. Not visually straightforward to see intercontinental convergence due to a multitude of colors. Classify them into several main categories and show the NIRv curve for each of them?

We agree with this overall sentiment, and we had previously experimented with numerous approaches for aiding interpretability, including something along the lines of what you have suggested here. The main issue arises from the fact that north-south hemispheric differences complicate the presentation of representative curves for a given color, and the complex shapes of some phenocycles complicates any attempt to 'rotate' northern and southern hemispheric phenologies to a common temporal reference point. We have edited the text to reference a series of supplemental figures (Fig. S6 and Figs. S8-10) that are intended to support the interpretation of intercontinental convergence and better depict some key examples. We also hope that the addition of Fig. S2 will further clarify the color patterns in Fig. 1 that we refer to in the text with respect to the example of the three more strongly seasonal Mediterranean regions. We hope that these changes improve the interpretability and visualization of intercontinental convergence, as you suggested, but we remain open to further suggestions.

Referee #3:

(Remarks to the Author)

This paper analyses the factors controlling the asynchrony of land surface phenology (LSP). The authors use multiple remote sensing datasets and flux towers to extract phenology and asynchrony, and then use machine learning and SHAP values to predict covariates of asynchrony. The authors map spatial asynchrony in LSP and propose hotspots in tropical montane and Mediterranean climates. The authors then use their results to support the hypothesis that 'allochry through allopatry' contributes to global biodiversity patterns. The article is well written in the introduction, but I found the methods section difficult to follow. I have two main concerns that I suggest the authors address

I have doubts that the resolution of MODIS and the SIF product is sufficient to answer the research question, as the asynchrony hypothesis should operate more at the individual scale, whereas MODIS can only detect landscape processes. In my opinion, this is a key aspect that the authors should address and, if necessary, use ground data to support their claims with satellite observations.

We agree that the use of ground data is a crucial step for demonstrating the relevance of our remote sensing analysis to the study of phenological and evolutionary dynamics in specific taxa, and we thank you for encouraging us to seek corroboration from ground data to justify the scale of our study. We see allochry by allopatry, and thus the mechanism underlying the

asynchrony of seasons hypothesis (ASH), as a phenomenon that should manifest at the scale of populations distributed across landscapes and, thus, as a phenomenon that is well matched to the landscape scale of MODIS. Indeed, populations of many species are commonly separated by distances equivalent to many times the ~5.5 km spatial resolution of our maps. That said, we agree this would only be a supposition in the absence of corroborating evidence.

We have now updated the manuscript with a series of species-level analyses, demonstrating that our LSP maps predict patterns of allochry by allopatry across a range of geographies, taxa, and ecological phenomena. The main findings include: 1.) as many as one in five iNaturalist taxa with non-unimodal flowering histograms, which are concentrated in phenological asynchrony hotspots, show stark spatial discontinuities in reproductive phenology that are predicted by our LSP map, as would be expected under the ASH (L271-290; Fig. 5A); 2.) both a toad and a bird in eastern Brazil, whose data we pulled from the few existing genetic studies of the ASH, show concordant patterns of population-level genetic divergence that are predicted by our LSP map, consistent with allochry by allopatry driven by spatial asynchrony in precipitation regimes (L291-307; Fig. 5B); and 3.) our LSP map even predicts the complex, latitudinally and orographically structured pattern of coffee harvest seasons across Colombia, demonstrating the broad but largely unrecognized ecological importance of allochry by allopatry (L308-326; Fig. 5C). Taken together, we feel that these results, which we have now included in our revised manuscript, demonstrate that the scale of our remote sensing work is adequate not just for identifying regions in which allochry by allopatry is likely to occur but also for testing species-specific ecological and evolutionary hypotheses. As we now better explain in the conclusion, we feel that this points the way toward a rich variety of new areas of research.

The second concern relates to the presentation and structure of the article, in particular the methods section. In the methods, there's a lot of detail about technical aspects, but the whole workflow is unclear. There are variables that are not written out and clearly described (for example LSP.asyneig, a key variable that is not defined). I would suggest to include a flowchart in the supplementary material and to describe the different steps more clearly.

We loved the suggestion of a workflow diagram and we have now added one (Fig. S1). We have also more clearly defined variable names and other key aspects of the overall workflow. We hope you agree that these changes have made the manuscript significantly more readable and cogent, but we nonetheless remain open to any additional suggestions.

Minor points

“Finally, we removed any pixels whose Pielou's evenness⁸⁵ was less than 0.8; we calculated Pielou's evenness, $J' = H'/H'max$, by calculating H' (i.e., Shannon's diversity index⁸⁶) using 12 values, each value being a monthly average proportion of non-missing daily data over the 10-year NIRV archive. Manual inspection of fitted phenological patterns after applying this series of filtering steps confirmed successful removal of locations otherwise producing spurious results.”

It is unclear why this was done, as in principle the authors are filtering time series with high variability, but in some rainfall limited systems this variability is a property of the system. I would encourage the authors to clarify this.

This is an excellent point. We believe this is simply a misunderstanding of what we did, which made us realize that we needed to write that sentence more clearly. We did not filter pixels below 0.8 evenness in their data *values*, which we agree would drop places with high rainfall-driven variability. Rather, we dropped pixels with less than 0.8 evenness in their data *availability*, assessed on a binary time series in which 0s indicated missing data. We did this to ensure that we were not fitting regressions at locations with repeating seasonal data gaps, into which a harmonic regression could interpolate spurious secondary peaks. We have now clarified our explanation of this step within the methods (L566-574).

In the methods, I found the Monte Carlo analysis for the confidence intervals very interesting. A rather robust analysis.

Thank you! We felt this was an important step.

“that allow stronger estimation of photosynthesis, across global deciduous and evergreen terrestrial biomes, than do the greenness indices previously used for such purposes”
I would be more cautious about this statement. NIRv is more of a proxy for green fPAR and fluorescence is more of a proxy for APAR. This is why these datasets relate better to photosynthesis than others, but they are not a direct estimate of photosynthesis. NIRv is also a greenness index (based on the same bands as NDVI) that better accounts for soil effects on greenness. So please clarify this part of the text.

This is a great point. We have now edited this statement to better interpret the nature of NIRv and SIF: “[...] that serve as stronger proxies of photosynthetic capacity, across global deciduous and evergreen terrestrial biomes, than the indices more traditionally used for such purposes” (L593-596). We hope you find the statement to be clearer and more accurate.

“To estimate the seasonality of stand-level photosynthesis”

Fitting a regression to NIRv and SIF is not estimating seasonality at stand level. We are talking about satellite data, which are not necessarily representative of the stand, in addition to the above comment on photosynthesis.

Thank you for raising this point. We agree completely. We have rewritten the sentence to provide a more accurate description of our methods and how they can be interpreted. It now reads, “We ran a harmonic regression to model the annual land surface phenology of every pixel in the global, filtered NIRV and SIF datasets...” (L596-597).

"Validation" against FLUXNET data. First of all, I would suggest to apply the quality check also to GPP data, e.g. by filtering the daily GPP data with $fqc > 0.70$, as done in several articles. This will ensure that the daily GPP comes from measured NEE data and not from gap filled data. Secondly, I think this validation can be quite misleading as it is not given that the flux footprint tower is representative of the pixel information. I think it is not really necessary and does not provide any further insight.

We appreciate your concerns regarding the limitations of the flux tower assessment. We agree that it is not a given that the flux tower footprint is representative of the pixel information, especially in regions with phenologically divergent habitat mosaics, as we discuss in the text. That said, we still feel that a flux tower-based assessment adds some value, given the common use of NIR_v and SIF data as estimators of GPP values and patterns. This analysis mirrors that use, and, we think, speaks to that body of related literature. We do find that it provides some insight, given both the generally high level of correlation between LSP and GPP phenocycles and the trend of reduced correlation in regions marked by high interannual and spatial variability in phenology. For this reason, we have chosen to retain the assessment. That said, we agree that it is not a 'validation', so have changed that terminology accordingly. We have also incorporated your suggested filtering of daily mean QC values < 0.7 before rerunning this evaluation on our updated LSP datasets (L665-667).

We have also now developed a complementary evaluation that relies on ground data from phenology observation stations within the US National Phenology Network (L168-171, L679-696; Fig. S15). Like the flux-tower assessment, this also has its limitations: the footprint of the phenology observation sites is not necessarily representative of the footprints of our remote sensing pixels, and the calculation of start and end phenometrics is notoriously scale-dependent, complicating cross-scale comparisons such as this (Park et al. 2021). That said, we feel that this assessment provides a complementary perspective on the overall validity of our findings, both because of the ground-based observations it derives from and because of the independent, thermal-based phenology dataset (SI-x) against which it provides a comparable correlation with those ground observations.

References:

- Alemu, W. G., and Henebry, G. M. (2013). Land surface phenologies and seasonalities using cool earthlight in mid-latitude croplands." *Environmental Research Letters* 8.4: 045002.
- Badgley, G., Field, C. B., Berry, J. A. (2017). Canopy near-infrared reflectance and terrestrial photosynthesis. *Science Advances*, 3, e1602244.
- Boyce, D. G., Petrie, B., Frank, K. T., Worm, B., Leggett, W. C. (2017). Environmental structuring of marine plankton phenology. *Nat. Ecol. Evol.* 1, 1484–1494.
- Dawson, A. (2016). eofs: a library for EOF analysis of meteorological, oceanographic, and climate data. *J. Open Res. Softw.* 4, p.e14.
- Ganguly, S., Friedl, M. A., Tan, B., Zhang, X., ... Verma, M. (2010). Land surface phenology from MODIS: Characterization of the Collection 5 global land cover dynamics product. *Remote sensing of environment*, 114(8), 1805-1816.

- Henebry, G., and de Beurs, K. M. (2013). "Remote sensing of land surface phenology: A prospectus." *Phenology: An integrative environmental science*. Dordrecht: Springer Netherlands. 385-411.
- Henebry, G. M. and Su, H. (1995) "Observing spatial structure in the Flint Hills using AVHRR maximum biweekly NDVI composites." In: *Proceedings of the 14th North American Prairie Conference*. Kansas State University Press, Manhattan.
- Martin, P., Bonier, F., Moore, I., Tewksbury, J. (2009). Latitudinal variation in the asynchrony of seasons: implications for higher rates of population differentiation and speciation in the tropics. *Ideas Ecol Evol* 2 (2009).
- Park, D. S., Newman, E. A., Breckheimer, I. K. (2021). Scale gaps in landscape phenology: challenges and opportunities. *Trends in Ecology & Evolution*, 36(8), 709-721.
- Shumway, R. H., Stoffer, D. S.. (2017). *Time Series Analysis and Its Applications*. Springer, New York, ed. 4th.
- Townshend, J., Hansen, M., Carroll, M., DiMiceli, C., Sohlberg, R., Huang, C. (2022). MODIS Collection 6.1 (C61) VCF Product User Guide. University of Maryland. https://lpdaac.usgs.gov/documents/1494/MOD44B_User_Guide_V61.pdf.
- White, M. A., and Nemani, R. R. (2006). Real-time monitoring and short-term forecasting of land surface phenology. *Remote Sensing of Environment* 104.1: 43-49.
- Wilson, B. T., Knight, J. F., McRoberts, R. E.. (2018). Harmonic regression of Landsat time series for modeling attributes from national forest inventory data. *ISPRS Journal of Photogrammetry and Remote Sensing*, 137, 29-46.
- Xie, Q., Moore, C. E., Cleverly, J., Hall, C. C., Ding, Y., Ma, X., ... Huete, A. (2023). Land surface phenology indicators retrieved across diverse ecosystems using a modified threshold algorithm. *Ecological Indicators*, 147, 110000.

Response to reviewers, revision 2: Terasaki Hart et al.: Global phenology maps reveal the diversity, convergence, and asynchrony of ecosystem function

Below, we provide our detailed responses (in green) to each of the reviewer comments on our original submission (in black).

Referee #1:

The authors have done an admirable job of addressing my concerns. I have just a few more minor suggestions to improve the precision and clarity of the ms.

Thank you!

First, it is appropriate to touch upon the effects of land cover/land use change on land surface phenologies in the introduction rather than waiting until later in the ms. Climate change is just one forcing on LSP. Consider this relevant study: Zhang, X., Liu, L., & Henebry, G. M. (2019). Impacts of land cover and land use change on long-term trend of land surface phenology: a case study in agricultural ecosystems. *Environmental Research Letters*, 14(4), 044020.

This makes a lot of sense. We have worked this point into our rewrite of the first paragraph of the introduction (L42).

Second, solar-induced fluorescence rather than sun-induced.

We are glad that you pointed this out. As both synonyms are used regularly in the literature, we have added a parenthetical clarification (L52-53).

Third, you cite the MOD09/MYD09 products as the source for the NIRv data at line 83 but at line 495 indicate that you used the NBAR product MCD43A4.061 to calculate the NIRv.

Thank you for catching this! We now correctly cite MCD43A4.061 in both locations (L85 and L512).

Fourth, at lines 266-267 you state “In this capacity, remote sensing of phenology could play an underappreciated role in improving our understanding of spatiotemporal evolutionary dynamics.” I think it would clarify the sentence to insert a delimiting phrase—perhaps either “land surface” or “proxies for organismal”—before phenology.

Agreed! This adds important clarity. We have changed this to read “remote sensing of proxies for organismal phenology could play...” (L265-266).

Fifth, at line 338, consider saying “terrestrial phenologies” rather than the singular as you are considering specifics associated with particular places.

This is an excellent point. We have incorporated this change (L336).

Sixth, you need a citation to GMTED2010. Look here: <https://pubs.usgs.gov/>

Thank you for pointing this out! We have now added the citation (L841).

Finally, it is appropriate to acknowledge explicitly your gratitude to the anonymous reviewers who have donated their attention and expertise to improve the presentation of your research. You are absolutely correct. We sincerely apologize for neglecting to include this in the earlier draft acknowledgments, and we have now done so. Your thoughtful, detailed suggestions have dramatically strengthened the manuscript. We are sincerely grateful for the considerable time all of the reviewers have taken to provide such thorough and constructive review.

Referee #2:

The authors did a good job and have addressed most comments, but further improvement and clarification are needed in several areas:

We are glad that we were able to address most of your concerns in the first round of review. As you review our new round of responses we hope you will agree that we have now managed to address your outstanding concerns, including points that we had not fully resolved previously.

First, this study primarily investigates spatial variations in long-term average seasonality from a spatial perspective, examining the impact of ecological and evolutionary influences across geographic gradients. This approach differs from most phenology studies, which typically assess interannual variation or long-term phenological trends in response to climate change.

Because of this, the authors should:

- Explicitly clarify this unique focus by defining "LSP" (long-term mean annual phenology) early in the introduction. Currently, the definition is vague, and readers may struggle to understand the specific variables or indices employed in this study.

Thank you, this is an excellent point. We have more explicitly defined this within the introduction (L82-83).

- Address the disconnect between the first and second paragraphs in the introduction. The first paragraph (lines 41-47) discusses phenology in the context of ecosystem science, carbon-water cycles, and earth system modeling. This is traditional phenology concept, focusing on the temporal, interannual dimension of phenology, which is not really the focus of this study. This paragraph does not establish a foundation for the subsequent analysis. In contrast, the second paragraph, which addresses landscape ecology and evolutionary biogeography, is more aligned with the study's spatial emphasis. I recommend revising the introduction to better align with the specific content and objectives of this study.

Both you and reviewer 3 have raised points along these lines, and we agree completely with your critique. We have revised the first paragraph of the introduction, removing the tangential references to carbon and water cycles and ESMs and honing the focus on land surface phenology and its relevance to landscape ecology (L38-56). We hope you will agree that this does a much better job of framing the scope and contextualizing the results of our work.

- It would be beneficial to discuss how climate change is altering landscape patterns of allochryony by allopatry and the associated uncertainties that this introduces to the current results. While this is not the direct focus of the study, it presents a promising direction for future research. For instance, the harmonic model used in this study does not account for extreme events like summer droughts or spring frosts that affect NIRv time series, yet such events are occurring with increasing frequency. Discussing these limitations could enhance the contextual relevance of the study's findings under ongoing climate variability and change.

Because our models are focused on long-term averages rather than trends or extremes, we had not given this much attention. However, you are right that this is important, not only to clarify the limitations of our results and their interpretation but also to articulate an important future research direction, as you suggested. We now address this within the conclusion (L338-345).

Second, I still find Figure 1 challenging to interpret and understand, and this concern has been echoed by another reviewer. At present, the key message of Figure 1 is unclear. Each figure

should stand alone as a clear, self-explanatory element within the manuscript. In contrast, Figure 2 conveys useful information, but it primarily serves as a zoomed-in view of Figure 1 rather than an independent figure. The spatial map in Figure 2 that shows the location of each sub-figure is redundant with Figure 1. I recommend combining Figures 1 and 2 to present a global map with selected zoom-in views to improve clarity and streamline the presentation.

Further, Figure 2 could benefit from an improved layout for the subfigures.

We appreciate this suggestion — it served as crucial guidance. We have now collapsed Figures 1 and 2 into a single figure. This new Figure 1 now shows the global map along with the zoomed-in regional maps. Based on your suggestion in the last round of review, we have now also paired the global map with depictions of 9 globally predominant phenocycles, based on K-means clustering of the global set of fitted phenocycles, which we recently managed to accomplish by first rotating the phenocycles of locations south of the ITCZ to match the seasonality of those north of it. We believe this new figure better harmonizes the global- and regional-scale results, balancing depictions of the complex and nuanced spatial patterns of LSP with interpretation of the global trends that emerge from those patterns. We have updated the methods (L743-753) to reflect these changes, retaining the clarification that these visualization methods aid interpretation but do not influence our analytical results.

Additionally, the statement in the caption that "Regional heterogeneity of land surface phenology (LSP) reflects the influence of topoclimate, ecohydrology, and vegetation structure" is not directly supported by the figure itself, as it does not depict any specific patterns related to topoclimate, ecohydrology, or vegetation structure. Need to clarify if this conclusion is derived through visual interpretation of spatial patterns, a literature review, or an analysis of datasets on topoclimate, ecohydrology, and vegetation structure. If these factors were not directly analyzed, this interpretation should be moved from the figure caption to the results or discussion sections to avoid suggesting that it is an explicit result of the figure itself.

This is an excellent point. This statement is based on a combination of all three of the factors that you mentioned here — mostly on the visual interpretation of our mapped patterns in light of prior regional research, but also on a statistical analysis in the case of the cheatgrass example. We have rewritten the caption to clarify this.

Third, the authors used USANPN SOS and spring indices in the analysis to demonstrate the performance of LSP fitting (line 165). However, USA-NPN spring indices are not direct observations; rather, they are simplified model estimates of phenological events based solely on growing degree days. In other words, these indices primarily reflect the temperature/climate regime rather than the biological seasonality of the region. This raises questions about the appropriateness of including these indices in the analysis. Additionally, the observed lack of alignment between the USA-NPN and LSP fitting results (e.g., Figure S15) is unsurprising, with an R^2 value of only 0.13. This does not support a claim of "significant correlation to NPN first-leaf dates" (line 169) and falls short of explaining the SOS to the same extent as the spring indices ($R^2 = 0.36$). Even if a similar ability to explain SOS were observed, the implications of such an alignment would remain unclear. The rationale for this analysis appears insufficiently justified, and the results may detract from the overall narrative of the study. Datasets like these derived from citizen science, which may lack rigorous quality control, raise concerns regarding data robustness. For this purpose, the PhenoCam Network, which offers more direct and standardized phenological observations for each plant functional type, would likely be a more suitable alternative.

We agree that this intercomparison was not especially useful and indeed detracted from the overall thrust of our work. You make an excellent point regarding the robustness and suitability of the USA-NPN dataset. As recommended, we have now replaced it with a systematic comparison of our LSP results with PhenoCam time series (Fig. S18; L165-178; L672-700). The results of this evaluation are similar to the FLUXNET results, both in terms of their overall strong agreement and their lower average agreement in semi-arid and seasonally dry biomes and in shorter time series.

Fourth, what is the added insight from using NIRv and SIF-based phenology approaches compared to traditional vegetation indices (Line 55)? While the authors cite numerous studies to show consistency with previous findings, it is unclear what novel contributions this framework brings beyond existing understanding. The novelty of this approach needs to be clearly articulated, highlighting what new perspectives or advantages are gained by using NIRv and SIF that are not achievable through traditional vegetation indices.

This is a good point. Our rewrites during round 1 of review removed some key citations related to this. We have reintroduced those citations and better articulated the added value of these newer remote sensing metrics that have a tighter, cross-biome correlation with seasonal variation in plant productivity, and thus with ecosystem function (L49-54).

Fifth, Fig. S1 workflow: the authors currently calculate aggregated mean values before applying the MODIS land cover filter. My understanding is that it would be more accurate to filter for land cover first and then proceed with aggregation. In the current order, pixels from non-targeted land cover types, such as urban or water, could skew the aggregated values. For instance, if such pixels represent a significant proportion (e.g., 40%) while forest 60%, they could substantially alter the final aggregated value.

This is a totally valid concern, and we have now reworked our analysis pipeline to better address it. We have not attempted to implement land cover filtering at the level of the native MODIS resolution (500 m; product MCD12Q1), as that would have required undoing the work that we did in round 1 of review in order to satisfy reviewer 1's express request to use the 5 km aggregated land cover product provided by the MODIS program for uses such as ours (MCD12C1). However, whereas we previously used a simple majority rule for sub-pixel valid land cover – i.e., 5 km pixels would be retained as long as >50% of their sub-pixels had valid land cover – we have now changed this to a much stricter 90% rule. We have updated the methods (L557-560) and the workflow diagram (Fig. S1) to reflect this change.

We did not observe any major qualitative changes in our results following the change to the 90% threshold. Pixel dropout occurred in the expected places – fringes surrounding patches of water, barren land, developed land – and caused the total loss of 7.16% of the pixels in our global LSP diversity map (Figure 1) and 7.34% of the pixels in our global LSP asynchrony map (now Figure 2A). This caused a shift in the third and fourth LSP EOFs and in their resultant RGB visualization. We also replaced one of the iNaturalist species visualized in former Figure 5 (now Figure 4) because it lost too much observation site coverage in the LSP map. We have replaced that species (*Xanthisma spinulosum*) with another in the same region (*Menodora scabra*) and have updated the main text (L270-289), the methods (L1041-1052), and Table S7 to reflect the other impacts of our now-reduced global LSP dataset on the number of iNaturalist taxa assessed, but our overall findings and their interpretation remain as before.

Another question related to this is about land cover change. e.g., in Figure S2, an abrupt change is visible for the Blodgett Forest site in 2015. Could this shift be due to land cover change? It is essential to clarify how pixels with changing land cover types are treated. While the authors mention handling transitions to and from agricultural land, it is unclear how other types of land cover change are managed in the workflow.

Yes, this shift in the Blodgett Forest time series is likely a result of management-driven land cover change. The site is described as “ponderosa pine plantation, mixed-evergreen coniferous forest”, according to the AmeriFlux website (<https://ameriflux.lbl.gov/doi/AmeriFlux/US-Blodgett/>). It is located on an experimental research station, and thus is subject to controlled burns, thinning, and logging. Manual inspection of our land cover dataset (MCD12C1) in Google Earth Engine shows that the flux tower site is classed as evergreen needleleaf for all years in our 2001-2020 time period except the first year, when it is classed as woody savanna. This suggests that the mid-2014 signal visible in the NIR_v time series in Fig. S2 was the result of a relatively minor land use or land cover change event. We have now noted this within the figure’s caption.

As you mentioned, we do omit pixels that register transitions to and from agricultural land. This is because agricultural management practices often purposefully control phenology (e.g., irrigation) and thus shift the timing of LSP. Beyond that, we do not filter our dataset for land cover change for three reasons. First, as Reviewer 1 pointed out in round 1 of review, our previous approach to filtering land cover change was causing us to inadvertently drop broad swathes of biomes whose interannual variation in productivity and intermediate vegetation structure cause spurious interannual variation in land cover classification (e.g., woodland and savanna). Second, as the Blodgett Forest example shows well, many forms of land cover change (e.g., selective logging) are substantial enough to register a shift in the time series of a continuous metric such as NIR_v yet too subtle to register a change in an annual, classified land cover time series such as MCD12C1, and thus would not be filtered out no matter how we used our land cover product. Finally, more substantial forms of land cover change (e.g., deforestation) should only affect our results in regions where different vegetation communities naturally display different phenological dynamics in response to the same broad bioclimatic controls. In this case, pixels undergoing land cover change are expected to fit phenocycles intermediate between the typical phenocycles of the before-change and after-change land cover classes, introducing some noise into our maps and statistical tests but not invalidating interpretation of their coherent patterns or significant results. We now explain this reasoning in the methods section (L560-576).

Additionally, per Reviewer 3’s request, we have now added covariates to the asynchrony drivers analysis to account for the average extent of land use and land cover change and the mean frequency of fire within each pixel’s neighborhood. Our results (Fig. S30, Table S6) reaffirm that land cover change is not making a major contribution to the patterns of LSP asynchrony we have described. Because of this, we are only more confident that although land cover change inevitably produces some noise in our maps and analyses it does not underpin any of our major findings.

Minor:

Line 150 In the Amazon, light and water availability exhibit a strong inverse relationship: during

the wet season, cloud cover reduces light availability, though water is abundant due to frequent rainfall; conversely, in the dry season, sunlight is plentiful, but water availability is limited. Both forests and non-forest ecosystems in the Amazon are regulated by a combination of light and water. It is difficult to isolate and represent their individual effects independently.

This is very true. Both factors inevitably play a role in regulating plant growth in these systems, so we should not have implied that the ecological processes controlling phenology here are simple enough to be represented by the independent effects of either one factor or the other. We feel that the interpretation we have suggested is reasonable in light of the literature we have cited, but we have rewritten these sentences to clarify that this is conjectural, that tropical phenological controls are complex and still poorly understood, and that more work is needed to determine how well reality aligns for the theoretical expectation we referenced (L148-157).

Fig. S5 why does Australia show such low R^2 ?

This is a good question: Australia has high sensitivity to ENSO and other climate oscillation modes, and because of this its drier biomes have exceptionally large interannual variation in ecosystem productivity (Broich, 2014; Poulter, 2014). We have now added an explicit reference to Australia as a key example of interannual variability (L172), which links to a newly added supplemental figure (Fig. S3).

Fig. S16, the white-to-black color legend is challenging to interpret. Consider switching to a contrasting color scheme, such as red-to-blue, to improve visual clarity and make patterns more discernible.

We did not feel that a diverging color palette was an appropriate choice for a non-diverging variable (R^2), but we have chosen a more complex and perceptually uniform sequential color palette to allow better discrimination between mapped values.

Referee #3:

The authors have created a global map of land surface phenology (LSP) using MODIS imagery that reveals LSP patterns. The map shows that similar climates have similar patterns, but local factors such as topography and vegetation can lead to differences in LSP diversity, and machine learning can be used to explain these patterns. This research has important implications for understanding how geographic separation can lead to differences in ecological and evolutionary processes.

The paper has been significantly improved in terms of clarity and reproducibility compared to the first submission. The authors have also updated the datasets, for example, using the most recent MODIS collection, extending the time period, and improving the gap-filling of NIRv to extend the spatial scope of the analysis. The authors have done a tremendous amount of work to address my concerns and introduce a new analysis using ground data, which is very valuable.

Thank you! We are glad that you find the work compelling and that you were pleased with the extensive changes we made during the first round of review. We are so grateful that you urged us to crosswalk our remote sensing analyses with ground data. That provided the key test of the Asynchrony of Seasons Hypothesis (ASH) that was previously missing, and it gave birth to a whole new section of the manuscript that we feel has added substantial depth and importance.

I still have one important point. I am still not convinced that the asynchrony hypothesis can be analyzed with MODIS, especially with the currently selected drivers. I really appreciate the effort of the authors to include the species level analysis, that is great. The new Figure 5 and the section "Remote sensing predicts species-level phenological asynchrony and genetic divergence" are very informative.

However, at the MODIS scale, there are aspects related to landscape management, disturbance, deforestation, land use that can be quite important in determining the LSP diversity signal. The authors use spatial structural entropy as a predictor, but I would suggest going deeper into the potential drivers of LSP by analysing for example disturbance patterns within the region (using for example the Global Forest Watch dataset), MODIS fire or burned area datasets, land cover change and other structural metrics.

This is a great point. It is entirely plausible that management, land use change, and other forms of disturbance could increase spatial variation in the timing of LSP in some regions, and thus elevate our metric of LSP asynchrony. Natural forms of disturbance (e.g., natural fire regimes), if they cause spatially variable phenological timing, would constitute a source of naturally occurring allochry by allopatry, but because our previous model lacked such variables it would be unable to characterize this. Anthropogenic disturbance, if it causes spatially variable timing in LSP, could cause a more serious omitted variable problem, as it would lead our model to attribute artificial LSP asynchrony to natural gradients that are not actually influencing it. Land cover change was only partially accounted for in our previous draft, by the exclusion from our LSP asynchrony map of all pixels that are agricultural or have ever transitioned to or from agriculture.

We have now reworked our LSP asynchrony drivers analysis to incorporate two new neighborhood metrics of disturbance, as recommended. The Global Forest Watch data you mentioned (Hansen et al. 2013) is actually only applicable to forest (defined as $\geq 25\%$ tree cover), so does not apply to the full range of biomes covered by our analysis. However, recent work by the same group has integrated that tree cover change dataset into a global, harmonized map of

2019 land use and land cover change (LULCC; Hansen et al. 2022). We reclassified this dataset into a binary categorization in which 1s represent pixels with LULCC (including non-fire-driven tree cover loss since 2000, whether followed by regrowth or not; built-up land; and cropland), and then calculated the proportion of each of our ~5 km analysis pixels that registers LULCC within this dataset's Landsat-resolution (30 m) pixels. For our random forest model we calculated the neighborhood mean of that LULCC fraction map (within a 100 km radial neighborhood, as with our vegetation entropy metric).

We also used the MODIS burned area dataset that you referenced (Giglio et al. 2021) to create a globally consistent map of burn frequency, which we then summarized as the neighborhood mean burn frequency within a 100 km radial neighborhood and included in the random forest model. This does not differentiate between anthropogenic and natural fire, but it nonetheless allows the random forest to describe places where spatial variability in fire regimes appears to contribute to spatial variability in phenology. This dataset complements the Hansen dataset well because the Hansen tree cover loss categories exclude fire-driven loss.

We have updated our model description (L198-203) and methods (L857-873) to describe the incorporation of these new variables, and we have updated all figures, tables, and statistics to reflect the new results. Aside from this, we have not made any major changes to the manuscript because our overarching findings and interpretation remain unchanged (we found that these new variables were only detected as dominant drivers in a few scattered locations; Fig. S30). We hope you will agree that the addition of these variables improves confidence in the key findings we report from this component of the study.

Minor points

Lines 44-50. In general I agree with the statement, but it is unclear how LSP diversity can actually improve the capability of Earth system models (running at grid cell or column level assuming one or a mixture of PFTs). In my opinion, this argument should be better explained. Once we understand LSP, how can this be used to improve models? Should it only be used to benchmark modelled LSP or can we gain insights into landscape diversity and how this interacts with climate? At the moment it's quite unclear what the benefit of understanding LSP is for models. In fact, I would remove this point as for me the analysis is more interesting in terms of landscape ecology and understanding ecological processes rather than supporting modelling. This is an excellent point. In light of your comments and those of reviewer 2 we have rewritten the first paragraph of the introduction, removing the tangential references to ESMS and honing the focus on landscape phenology. We hope you agree that this serves as a more natural and variable context for our work.

Line 54 "Multivariate analysis of new physiologically based proxies" Proxies or what exactly? SIF and NIR_v are proxies for two very different things.

This is a great point. While SIF and NIR_v do indeed measure two very different things, they are related by way of their strong correlation with each other and with GPP. Because of this we use them as independent proxies of plant productivity and thus of ecosystem function. We have now clarified this briefly in the main text (L52-55) and in more detail in the methods (L505-509).

Methods:

Line 510: Why are longer SIF datasets not used?

This is a totally fair question. When we began this work nearly seven years ago this was one of the few available SIF datasets that was global, spatially contiguous, and at a resolution roughly comparable to the LSP gradients we aimed to study. Thus, we adopted it as an independent dataset against which to evaluate our main results. Because our main NIR_v dataset is robust, and because this SIF dataset still serves that supporting role despite its limited temporal extent, we have not seen an overwhelming scientific motivation for replacing it.

Figure S2: I think it would be important to show example time series for some specific areas, e.g. few pixels in the tropics, few in the Mediterranean (although I think Tonzi Ranch can be classified as Mediterranean), some boreal areas, so areas where harmonic regression may have problems due to data gaps.

This is an excellent suggestion. Beyond the Mediterranean sites in the California figure and the tropical lowland sites in the Amazon floodplain figure, we have now added three additional supplemental figures, comparing fitted phenocycles between divergent sites in a semi-arid region with high interannual variability (Australian outback; Fig. S3), between divergent sites in a tropical montane region with significant data loss because of cloud cover (the Colombian Andes; Fig. S4), and between sites with similar phenologies but different land cover, and thus different degrees of snow-driven NIR_v data dropout, in the boreal biome (field and forest sites in Saskatchewan, Canada; Fig. S5). We hope you agree that this increases confidence in the ability of our modeling framework to fit phenocycles across ecological contexts, including in the locations most likely to challenge our approach. Beyond this, by providing the data viewer app (linked within our GitHub repository) we enable readers to create and inspect these comparisons for any sites of interest.

References

- Broich, M. et al. Land surface phenological response to decadal climate variability across Australia using satellite remote sensing. *Biogeosciences* 11, 5181–5198 (2014).
- Giglio, L. et al. MODIS/Terra+Aqua Burned Area Monthly L3 Global 500m SIN Grid V061. 10.5067/MODIS/MCD64A1.061 (2021).
- Hansen, M. C., et al. High-Resolution Global Maps of 21st-Century Forest Cover Change. *Science*, 342: 850-53. (2013).
- Hansen, M. C., et al. Global land use extent and dispersion within natural land cover using Landsat data. *Environmental Research Letters* 17.3: 034050. (2022).
- Poulter, B. et al. Contribution of semi-arid ecosystems to interannual variability of the global carbon cycle. *Nature* 509, 600–603 (2014).

Response to reviewers, revision 3: Terasaki Hart et al.: Global phenology maps reveal the diversity, convergence, and asynchrony of ecosystem function

Below, we provide our detailed responses (in green) to each of the reviewer comments on our original submission (in black).

Referee #2:

The authors have done a substantial amount of work to address the previous reviewer's comments, and the manuscript has been significantly improved. I only have one major concern in this round.

In the revised manuscript, the authors state that they used NDVI derived from ground-based phenology cameras in the PhenoCam network and added a new Fig. S18. However, this appears to be a misunderstanding. The PhenoCam network does not provide NDVI; rather, it provides green chromatic coordinate (Gcc), which is derived from RGB imagery. Gcc is calculated as the ratio of the green digital number to the sum of the red, green, and blue digital numbers ($Gcc = G / [R + G + B]$). While Gcc is widely used as a proxy for canopy greenness and phenological transitions, it is fundamentally different from NDVI, which requires near-infrared (NIR) data that is not available from standard PhenoCam cameras.

It is surprising that the authors incorporated and analyzed this dataset—adding new figures in the process—yet confused Gcc with NDVI. This suggests a lack of clarity regarding the variables used. This point should be corrected in the manuscript.

We chose to use NDVI instead of GCC from the PhenoCam network because the two variables show differential performance in relation to ground-based human phenological observations (Richardson 2023), and the former is a much better comparator to our NDVI-derived MODIS index. PhenoCam cameras initially provided only GCC data, but the most common camera model used in the network (StarDot NetCam SC) is actually sensitive to infrared (IR), and an innovation in the earlier days of the network found that the software controlling the sliding IR filter could be leveraged to enable capture of contemporaneous NIR and RGB imagery and thus calculation of a “camera NDVI” metric (Petach et al. 2014). The PhenoCam version 3.0 dataset now includes this camera NDVI for ~75% of all camera sites, and this is the dataset we used. We have now clarified this (lines 769-771) and we have also switched to citing the paper describing version 3.0 (Young et al. 2025; this paper was not available for citation at the time we submitted our second draft because it was published on March 28, 2025).

In addition, I could not find a clear data description section in the Methods that introduces and details the datasets used in the study. I recommend including a dedicated subsection that explicitly describes the source, nature, and processing of all datasets, with citations, to improve transparency and reproducibility.

This is a good point. We use a wide variety of datasets in this work. The two land surface phenology remote sensing datasets are presented within their own section of the methods, but the other datasets are introduced across the various sections in which they first appear. Because these other datasets differ substantially in nature and purpose, we felt that trying to present them all in

one section would reduce readability. Thus, instead, we have now reworked former Table S4, which provided a synopsis of all of the datasets used as covariates in our phenological asynchrony drivers analysis. This table (now Supplementary Table 1) provides a synopsis of all datasets used in the study. We reference this table at appropriate points in the manuscript (lines 86-87, 597-598, 1224). We hope it will provide a helpful guide for anyone who reads at this level of detail.

References

- Petach, A. R. et al. Monitoring vegetation phenology using an infrared-enabled security camera. *Agricultural and forest meteorology*, 195, 143-151 (2014).
- Richardson, A. D. PhenoCam: An evolving, open-source tool to study the temporal and spatial variability of ecosystem-scale phenology. *Agricultural and Forest Meteorology*, 342, 109751 (2023).
- Young, A. M. et al. Tracking vegetation phenology across diverse biomes using Version 3.0 of the PhenoCam Dataset. *Earth Syst. Sci. Data Discuss.* [preprint], <https://doi.org/10.5194/essd-2025-120>, in review (2025).